# SENDAI: A Hierarchical Sparse-measurement, EfficieNt Data AssImilation Framework

**Xingyue Zhang** [* 1]   **Yuxuan Bao** [* 2]   **Mars Liyao Gao** [3]   **J. Nathan Kutz** [2 4]

## Abstract

Bridging the gap between data-rich training regimes and observation-sparse deployment conditions remains a central challenge in spatiotemporal field reconstruction, particularly when target domains exhibit distributional shifts, heterogeneous structure, and multi-scale dynamics absent from available training data. We present SENDAI, a hierarchical **S**parse-measurement, **E**fficie**N**t **D**ata **A**ss**I**milation Framework that reconstructs full spatial states from hyper sparse sensor observations by combining simulation-derived priors with learned discrepancy corrections. We demonstrate the performance on satellite remote sensing, reconstructing MODIS (Moderate Resolution Imaging Spectroradiometer) derived vegetation index fields across six globally distributed sites. Using seasonal periods as a proxy for domain shift, the framework consistently outperforms established baselines that require substantially denser observations—SENDAI achieves a maximum SSIM improvement of $185\%$ over traditional baselines and a $36\%$ improvement over recent high-frequency-based methods. These gains are particularly pronounced for landscapes with sharp boundaries and sub-seasonal dynamics; more importantly, the framework effectively preserves diagnostically relevant structures—such as field topologies, land cover discontinuities, and spatial gradients. By yielding corrections that are more structurally and spectrally separable, the reconstructed fields are better suited for downstream inference of indirectly observed variables.

*Equal contribution [1]School of Environmental and Forest Sciences, University of Washington, Seattle, USA [2]Department of Applied Mathematics, University of Washington, Seattle, USA [3]Paul G. Allen School of Computer Science & Engineering, University of Washington, Seattle, USA [4]Department of Electrical and Computer Engineering, University of Washington, Seattle, USA. Correspondence to: J. Nathan Kutz <kutz@uw.edu>, Yuxuan Bao <baoyx@uw.edu>.

*Proceedings of the 43$^{rd}$ International Conference on Machine Learning*, Seoul, South Korea. PMLR 306, 2026. Copyright 2026 by the author(s).

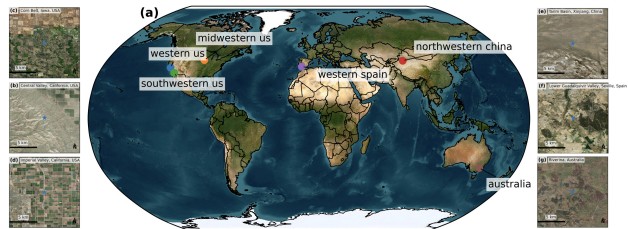

*Figure 1.* Study sites for NDVI reconstruction experiments. (a) Global distribution of the six study sites, spanning: (b, f) Mediterranean, (c) continental, (d, e) arid, and (g) subtropical climates.

The results therefore highlight a lightweight and operationally viable framework for sparse-measurement reconstruction that is applicable to physically grounded inference, resource-limited deployment, and real-time monitor and control.

## 1. Introduction

Reconstructing full spatiotemporal fields from sparse observations constitutes a fundamental challenge across Earth sciences, with applications spanning vegetation monitoring, hydrological modeling, and climate analysis (Weiss et al., 2020; Mohanty et al., 2017; Adrian et al., 2025; Jiang et al., 2025). Satellite remote sensing platforms such as MODIS provide unprecedented global coverage, yet cloud contamination, sensor gaps, and transmission constraints frequently yield incomplete spatial fields that compromise downstream analyses (Shen et al., 2015; Zhang et al., 2018). Traditional methods typically require substantial observational coverage to achieve high-fidelity reconstruction (Stock et al., 2020).

Contemporary deep learning approaches have demonstrated impressive reconstruction capabilities but typically demand GPU clusters, massively labeled training datasets, and substantial computational resources that preclude deployment in operational or resource-constrained settings (He et al., 2016; Morel et al., 2025; Meraner et al., 2020; Cresson, 2018; Zhang et al., 2018), while still exhibiting considerable data dependency (Sarafanov et al., 2020). This computational burden is particularly problematic for near-real-time agricultural monitoring, where latency constraints and bandwidth limitations necessitate lightweight yet accurate

reconstruction (Gao & Zhang, 2021; Denby & Lucia, 2020). Recent works in physics-informed AI (Kutz et al., 2025; Karniadakis et al., 2021; Raissi et al., 2019; Fan et al., 2025; Liu et al., 2024) and neural operators (Lu et al., 2021; Li et al., 2020; Roy et al., 2025) have shown how physical structure can be embedded into surrogates for improved generalization. However, many practical reconstruction problems are not naturally posed as a well-specified PDE with reliable priors at the spatial and temporal scales of interest, motivating methods that remain effective under weak or unknown governing structure. Moreover, the assumption of stationarity in the underlying field distribution is frequently violated in Earth observation contexts where phenological shifts, seasonal transitions, and land cover dynamics introduce substantial domain changes over time (Zeng et al., 2024; Truong et al., 2021; Cheng et al., 2025).

Additionally, existing approaches often learn latent representations that blend multiple contributing effects, yielding entangled corrections with limited physical interpretability (Dylewsky et al., 2019; Wang et al., 2022; Chen et al., 2022). This entanglement hinders inverse inference of indirectly observed fields (e.g., soil moisture, land surface temperature)—an essential capability in environmental monitoring and other Earth-science settings where target quantities are sparsely observed, intermittently available, or infeasible to measure directly at scale (Koronaki et al., 2025).

In this work, we present SENDAI (**S**parse-measurement, **E**fficie**N**t **D**ata **A**ss**I**milation), a hierarchical data assimilation framework that reconstructs full spatial states from severely sparse sensor observations by combining simulation-derived priors with learned discrepancy corrections. The architecture decomposes reconstruction into two complementary pathways: (i) a *low-frequency pathway* that leverages Takens' embedding theorem (Takens, 2006) through shallow recurrent decoder networks (SHRED) (Williams et al., 2024) to capture dominant spatiotemporal dynamics, with latent-space adversarial alignment bridging distribution shifts between simulation and ground truth; and (ii) a *high-frequency pathway* employing sequential frequency peeling with coordinate-based implicit neural representations (INRs) (Sitzmann et al., 2020; Tancik et al., 2020) to resolve fine-scale structure, sharp boundaries, and localized corrections that smooth decoders cannot capture. This hierarchical decomposition enables the framework to address heterogeneous spatiotemporal fields exhibiting domain changes, topological variations, and multi-scale structure (Barwey et al., 2025). Moreover, it overcomes the limitations of spectral bias and band-pass filtering commonly observed in neural networks (Rahaman et al., 2019).

The principal innovations of this work address three critical gaps in existing machine learning methods for spatiotemporal reconstruction:

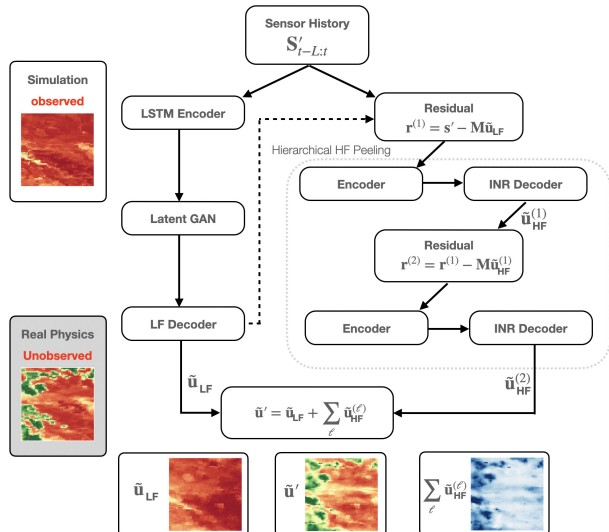

*Figure 2.* SENDAI architecture. The LF pathway learns dominant dynamics from simulation and aligns to ground truth via a latent GAN. The HF pathway employs sequential peeling layers, each consisting of a sensor residual encoder and a coordinate-based INR decoder, to extract spectrally-distinct corrections.

1. *Extreme sparsity reconstruction.* The framework achieves effective full-state reconstruction from only 64 sensors covering 1.56% of the spatial domain—substantially below the density thresholds required by conventional methods.

2. *Computational efficiency for operational deployment.* The lightweight architecture enables training and inference on standard hardware within minutes, making it suitable for resource-constrained operational settings where extensive computational infrastructure is unavailable (Erichson et al., 2020).

3. *Hierarchical frequency peeling for heterogeneous fields.* We introduce a novel sequential peeling strategy that decomposes high-frequency corrections into interpretable layers with explicit spectral constraints and frequency exclusion mechanisms. Combined with coordinate-based INR decoders, this approach produces spatially coherent reconstructions that preserve topological structure.

While demonstrated here on vegetation index, the framework could be generalized beyond Earth observation to broader remote sensing tasks with learnable structure and sparse measurement on heterogeneous spatiotemporal fields.

## 2. Preliminaries

We demonstrate SENDAI on satellite remote sensing, reconstructing MODIS-derived NDVI fields across six globally distributed sites spanning Mediterranean, continental, arid, and subtropical climates, and using seasonal periods as proxies for the sim2real domain shift.

## 2.1. Remote Sensing Data and Vegetation Index

We utilize imagery from the Moderate Resolution Imaging Spectroradiometer (MODIS) aboard NASA's Terra and Aqua satellites (Justice et al., 2002), which provide continuous global observations with complete coverage every one to two days. Our primary state variable is the Normalized Difference Vegetation Index (NDVI), computed as:

$$\text{NDVI} = \frac{\rho_{\text{NIR}} - \rho_{\text{Red}}}{\rho_{\text{NIR}} + \rho_{\text{Red}}} \tag{1}$$

where $\rho_{\text{NIR}}$ and $\rho_{\text{Red}}$ denote surface reflectance in the near-infrared (841–876 nm) and red (620–670 nm) spectral bands, respectively (Tucker, 1979; Huete et al., 2002). NDVI exploits the distinctive spectral signature of photosynthetically active vegetation, ranging from approximately $-0.1$ for water bodies and bare soil to $0.8$–$0.9$ for dense vegetation. Specifications of MODIS is in Appendix A.

## 2.2. Data Collection and Experimental Setup

**Data Acquisition and Processing.** All MODIS imagery was acquired through Google Earth Engine (Gorelick et al., 2017), merging Terra and Aqua observations to maximize temporal density. For each study site, we define a 15 km × 15 km region resampled to a standardized $64 \times 64$ pixel grid. Complete specifications are provided in Appendix B.

**Study Sites.** We evaluate our SENDAI framework across six globally distributed study sites spanning diverse climate zones, land cover types, and phenological regimes (Figure 1). This geographic diversity serves to demonstrate the generalizability of the approach across heterogeneous landscapes and explicitly tests model robustness to out-of-distribution conditions arising from differences in vegetation phenology, soil backgrounds, and atmospheric conditions. Table 5 summarizes the geographic, climatic, and phenological characteristics of each study site. Details of site biogeochemical characteristics are provided in Appendix C.

**Seasonal Split Strategy and Sensor Configuration.** A fundamental challenge in applying data assimilation frameworks to satellite remote sensing is the absence of a true physics-based simulation model analogous to those available for fluid dynamics or combustion systems. In our formulation, we repurpose the architecture by treating observations from one seasonal period as the "simulation" training data and observations from a different season as the "ground truth" reality to be reconstructed. The specific seasonal assignments and detailed sparse sensor configuration specifications are provided in Appendix D.

## 2.3. Baseline Methods

To contextualize SENDAI's performance, we compare against established geostatistical baselines and recent deep learning approaches, all operating under identical constraints: reconstruction from 64 sensors with full temporal context within the ground truth period. **SG+IDW** applies Savitzky-Golay filtering with Inverse Distance Weighting interpolation (Wong et al., 2004); **HANTS+IDW** fits harmonic models before IDW interpolation (Roerink et al., 2000); **Kriging** performs per-timestep Gaussian Process regression; and **MMGN** (Luo et al., 2024) uses a Gabor filter-based auto-decoder for continuous field reconstruction. We additionally evaluate three deep learning architectures: **FNO** (Li et al., 2020), **DeepONet** (Lu et al., 2021), and **Senseiver** (Santos et al., 2023), all trained on simulation and deployed on ground truth under the same protocol as SENDAI. None bridges the sim2real domain shift without additional adaptation; when augmented with SENDAI's HF pipeline, DeepONet and Senseiver approach SENDAI's performance, isolating SENDAI's multiscale data assimilation components as the active ingredients. Further implementation details are provided in Appendix E.

# 3. SENDAI Architecture

We develop a hierarchical data assimilation architecture that reconstructs full spatial fields from sparse sensor observations, bridging the gap between simplified simulation models and complex multi-physics ground truth systems. The approach builds upon the DA-SHRED framework and Cheap2Rich, its multiscale extension (Bao & Kutz, 2025a; Bao et al., 2026) while introducing two key methodological advances: (i) a *sequential frequency peeling* strategy that decomposes high-frequency corrections into interpretable layers, enabling cleaner spectral separation and improved stability; and (ii) a *coordinate-based implicit neural representation* (INR) for the high-frequency pathway that produces spatially coherent reconstructions. Critically, the full field of the ground truth system is never observed during training—only sparse sensor measurements are available from the target domain. The resulting architecture is lightweight and can be trained on standard hardware without requiring GPU clusters. Figure 2 illustrates the complete pipeline. Code is available at: `https://github.com/xswzaqnjimko/SENDAI_framework`.

## 3.1. Problem Formulation

We adopt the notation established in (Bao & Kutz, 2025a). Let $\mathbf{u}_k = \mathbf{u}(\mathbf{x}, t_k) \in \mathbb{R}^n$ denote the full state at time $t_k$, where $n = H \times W$ is the spatial dimension. Sparse sensor measurements are given by $\mathbf{s}_k = \mathbf{M}\mathbf{u}_k \in \mathbb{R}^p$ with $p \ll n$, where $\mathbf{M} \in \mathbb{R}^{p \times n}$ is the sampling operator. The simulation model, governed by $\dot{\mathbf{u}} = \mathcal{N}(\mathbf{u}, \mathbf{x}, t)$, provides full-state training data $\mathbf{X} = [\mathbf{u}_1, \ldots, \mathbf{u}_m]$. For the multi-physics ground truth system, governed by an unknown $\dot{\mathbf{u}}' = \mathcal{M}(\mathbf{u}', \mathbf{x}, t)$, only sensor measurements $\mathbf{S}' = [\mathbf{s}'_1, \ldots, \mathbf{s}'_m]$ are observed.

The reconstruction task seeks a mapping $\mathbf{X}' = F_\theta(\mathbf{S}')$ that estimates the unobserved full state of the ground truth system. Following the multi-scale decomposition (Bao et al., 2026; Ilersich & Nair, 2025), we write:

$$\tilde{\mathbf{u}}'(t) = \tilde{\mathbf{u}}_{\mathrm{LF}}(t) + \tilde{\mathbf{u}}_{\mathrm{HF}}(t), \tag{2}$$

where $\tilde{\mathbf{u}}_{\mathrm{LF}}$ captures dominant dynamics learned from simulation and adapted via latent-space alignment; $\tilde{\mathbf{u}}_{\mathrm{HF}}$ represents fine-scale corrections absent from the simplified model.

## 3.2. Low-Frequency (LF) Pathway

The low-frequency pathway follows the DA-SHRED methodology (Bao & Kutz, 2025a), consisting of a temporal encoder trained on simulation data and a latent-space alignment mechanism.

**Temporal Encoder.** Given sensor time-history $\mathbf{S}'_{t-L:t} = [\mathbf{s}'_{t-L+1}, \ldots, \mathbf{s}'_t] \in \mathbb{R}^{L \times p}$ with $L$ temporal lags, the encoder maps this sequence to a latent representation via a multi-layer LSTM:

$$\mathbf{z}_{\mathrm{LF}}(t) = \mathcal{E}_{\mathrm{LF}}(\mathbf{S}'_{t-L:t}; \theta_{\mathrm{enc}}) = \mathrm{LayerNorm}\left(\mathbf{h}_L^{(K)}\right), \tag{3}$$

where $\mathbf{h}_L^{(K)} \in \mathbb{R}^{d_z}$ is the final hidden state of the $K$-layer LSTM with hidden dimension $d_z$.

**Latent-Space Alignment.** To address the distribution mismatch between simulation-derived latents and those induced by ground truth sensor measurements, we employ a residual generator $\mathcal{G}$ that learns to align the latent distributions adversarially (Goodfellow et al., 2020):

$$\tilde{\mathbf{z}}_{\mathrm{LF}}(t) = \mathbf{z}_{\mathrm{LF}}(t) + \gamma \cdot \mathcal{G}(\mathbf{z}_{\mathrm{LF}}(t); \theta_{\mathcal{G}}), \tag{4}$$

where $\gamma$ is a learnable scale parameter initialized small to ensure stable training. The generator and discriminator $\mathcal{D}$ are trained with the standard adversarial objectives as described in Appendix G.2.

**LF Decoder with Spectral Constraint.** The aligned latent code is decoded and then low-pass filtered to enforce the low-frequency constraint:

$$\tilde{\mathbf{u}}_{\mathrm{LF}}(t) = \mathcal{P}_{k_c}\left(\mathcal{D}_{\mathrm{LF}}(\tilde{\mathbf{z}}_{\mathrm{LF}}(t); \theta_{\mathrm{dec}})\right) \in \mathbb{R}^n, \tag{5}$$

where $\mathcal{D}_{\mathrm{LF}}$ is a multi-layer MLP with ReLU activations and layer normalization, and $\mathcal{P}_{k_c}$ denotes a low-pass filter that retains only Fourier modes with wavenumber $k \leq k_c$:

$$\mathcal{P}_{k_c}(\mathbf{u}) = \mathcal{F}^{-1}\left(\mathbf{1}_{k \leq k_c} \cdot \mathcal{F}(\mathbf{u})\right), \tag{6}$$

where $\mathcal{F}$ and $\mathcal{F}^{-1}$ denote the discrete Fourier transform and its inverse. This explicit spectral constraint ensures a clean separation between LF and HF components.

## 3.3. Hierarchical High-Frequency Peeling (HFP)

A key limitation of single-stage high-frequency correction is the difficulty in separating distinct spectral modes that may have different physical origins. We introduce a *hierarchical peeling* structure that sequentially extracts frequency components layer by layer, enabling cleaner spectral separation and improved stability and interpretability.

**Sequential Residual Computation.** Let $\mathbf{u}^{(0)} = \tilde{\mathbf{u}}_{\mathrm{LF}}$ denote the low-frequency base reconstruction. For $\ell = 1, \ldots, N_{\mathrm{peel}}$ hierarchical layers, we compute:

$$\mathbf{r}^{(\ell)}(t) = \mathbf{s}'(t) - \mathbf{M}\mathbf{u}^{(\ell-1)}(t), \tag{7}$$

$$\tilde{\mathbf{u}}_{\mathrm{HF}}^{(\ell)}(t) = \mathcal{H}^{(\ell)}(\mathbf{r}^{(\ell)}(t); \theta_{\mathrm{HF}}^{(\ell)}), \tag{8}$$

$$\mathbf{u}^{(\ell)}(t) = \mathbf{u}^{(\ell-1)}(t) + \tilde{\mathbf{u}}_{\mathrm{HF}}^{(\ell)}(t), \tag{9}$$

where $\mathbf{r}^{(\ell)} \in \mathbb{R}^p$ is the sensor residual after all previous corrections, and $\mathcal{H}^{(\ell)}$ is the $\ell$-th high-frequency pathway.

The critical insight is that each layer $\mathcal{H}^{(\ell)}$ sees only the residual *after* all preceding layers have been applied, with gradients detached from previous layers during training. This prevents mode interference and encourages each layer to capture distinct spectral content.

**Frequency-Guided Sparsity with Exclusion.** To promote interpretable frequency decomposition, each peeling layer is trained with a bandlimited sparsity regularizer that additionally *excludes* frequencies discovered by previous layers. Let $\hat{\mathbf{u}}_{\mathrm{HF}}^{(\ell)}(\mathbf{k})$ denote the 2D Fourier coefficients of the $\ell$-th HF output. The sparsity loss combines three terms:

$$\mathcal{R}_{\mathrm{sparse}}^{(\ell)} = \underbrace{\frac{\|\hat{\mathbf{u}}^{(\ell)}\|_{1,\mathcal{B}}}{\|\hat{\mathbf{u}}^{(\ell)}\|_{2,\mathcal{B}} + \epsilon}}_{\text{in-band L1/L2}} + \beta_1 \mathcal{P}_{\bar{\mathcal{B}}}^{(\ell)} + \beta_2 \mathcal{P}_{\mathcal{E}}^{(\ell)}, \tag{10}$$

where the penalty terms are defined as:

$$\mathcal{P}_{\bar{\mathcal{B}}}^{(\ell)} = \frac{\|\hat{\mathbf{u}}^{(\ell)}\|_{2,\bar{\mathcal{B}}}^2}{\|\hat{\mathbf{u}}^{(\ell)}\|_2^2 + \epsilon} \quad \text{(out-of-band)}, \tag{11}$$

$$\mathcal{P}_{\mathcal{E}}^{(\ell)} = \frac{\|\hat{\mathbf{u}}^{(\ell)}\|_{2,\mathcal{E}^{(\ell)}}^2}{\|\hat{\mathbf{u}}^{(\ell)}\|_2^2 + \epsilon} \quad \text{(exclusion)}. \tag{12}$$

Here $\mathcal{B} = \{\mathbf{k} : \|\mathbf{k}\| \leq k_{\max}\}$ is the target frequency band, $\bar{\mathcal{B}}$ its complement, and $\mathcal{E}^{(\ell)} = \bigcup_{j < \ell} \mathcal{E}_j$ is the union of exclusion regions around frequencies discovered by layers $1, \ldots, \ell - 1$. The exclusion radius $r_{\mathrm{exc}}$ defining each $\mathcal{E}_j$ creates a buffer zone around discovered frequencies, preventing mode leakage where a subsequent layer partially recaptures the same spectral content with slight offset. The weights $\beta_1, \beta_2 \gg 1$ (typically $\beta_1 = \beta_2 = 100$) strongly penalize energy outside the allowed band and near previously captured modes.

**Adaptive Top-$k_\ell$ Mode Selection.** For each peeling layer $\ell$, we employ a top-$k_\ell$ sparsity term that encourages energy concentration in $k_\ell$ dominant modes:

$$\mathcal{R}_{\mathrm{topk}}^{(\ell)} = 1 - \frac{\sum_{i=1}^{k_\ell} |\hat{u}_{(i)}|^2}{\|\hat{\mathbf{u}}_{\mathrm{HF}}^{(\ell)}\|_2^2 + \epsilon}, \tag{13}$$

where $|\hat{u}_{(1)}| \geq \cdots \geq |\hat{u}_{(k_\ell)}|$ are the $k_\ell$ largest Fourier magnitudes. The key challenge is selecting $k_\ell$ adaptively to avoid peeling too much (capturing noise or minor modes) or too little (leaving correlated modes partially captured). We determine $k_\ell$ via spectral analysis of the layer's input residual $\mathbf{r}^{(\ell)}$ prior to training using two criteria:

*(i) Correlation clustering:* Modes with similar frequencies often arise from the same physical phenomenon. We group modes whose frequency radii fall within a bandwidth $\Delta k$:

$$k_\ell^{\text{cluster}} = |\{\mathbf{k} : |\|\mathbf{k}\| - \|\mathbf{k}^*\|| \leq \Delta k\}|, \qquad (14)$$

where $\mathbf{k}^* = \arg\max_{\mathbf{k}} |\hat{r}_{\mathbf{k}}^{(\ell)}|$ is the dominant frequency in the residual.

*(ii) Energy concentration threshold:* Select $k_\ell$ such that capturing these modes accounts for a target fraction $\rho$ (e.g., $\rho = 0.8$) of the residual's spectral energy:

$$k_\ell^{\text{energy}} = \min\left\{k : \frac{\sum_{i=1}^{k} |\hat{r}_{(i)}|^2}{\|\hat{\mathbf{r}}^{(\ell)}\|_2^2} \geq \rho\right\}. \qquad (15)$$

The final selection takes $k_\ell = \max(k_\ell^{\text{cluster}}, k_\ell^{\text{energy}})$, ensuring we capture coherent mode clusters with sufficient energy. For subsequent layers where the residual structure is less pronounced, we may set $k_\ell = \infty$ (equivalently, use bandlimited sparsity only without the top-$k$ constraint), allowing the layer to capture all remaining in-band energy.

### 3.4. Coordinate-Based Implicit Neural Representation

Direct MLP-based mapping from sparse sensor residuals to full spatial fields often produces "dotted" artifacts—localized peaks at sensor locations with poor interpolation elsewhere. This occurs because the network optimizes primarily for sensor-location accuracy without explicit spatial structure.

We address this limitation by replacing the direct MLP decoder with a *coordinate-based implicit neural representation* (INR) (Sitzmann et al., 2020; Tancik et al., 2020) that learns a continuous function over space. The key insight is that querying a shared decoder at *all* spatial coordinates, conditioned on a global latent encoding of the sensor residual, naturally produces smooth interpolation. Extended discussion in Appendix F.

**Architecture.** Each HF peeling layer $\mathcal{H}^{(\ell)}$ consists of two components:

*(i) Sensor Residual Encoder:* Maps the $p$-dimensional sensor residual to a compact latent code:

$$\mathbf{z}_{\text{HF}}^{(\ell)} = \mathcal{E}_{\text{HF}}^{(\ell)}(\mathbf{r}^{(\ell)}; \theta_{\text{enc}}^{(\ell)}) \in \mathbb{R}^{d_{\text{HF}}}, \qquad (16)$$

implemented as a multi-layer perceptron with layer normalization.

*(ii) Coordinate-Based Decoder:* For each spatial coordinate $(x, y) \in [0, 1]^2$ (normalized), the decoder computes:

$$u_{\text{HF}}^{(\ell)}(x, y) = \gamma^{(\ell)} \cdot \mathcal{D}_{\text{INR}}^{(\ell)}([\text{PE}(x, y); \mathbf{z}_{\text{HF}}^{(\ell)}]), \qquad (17)$$

where $\text{PE}(\cdot)$ is a Fourier positional encoding, $[\cdot ; \cdot]$ denotes concatenation, and $\gamma^{(\ell)}$ is a learnable scale parameter.

**Fourier Positional Encoding.** Following (Tancik et al., 2020), we encode spatial coordinates using sinusoidal features at $L$ frequency bands $\{\sigma_j\}_{j=1}^{L}$, typically log-spaced from 1 to $\sigma_{\max}$:

$$\text{PE}(x, y) = [x, y, \sin(2\pi\sigma_1 x), \cos(2\pi\sigma_1 x), \ldots,$$
$$\sin(2\pi\sigma_L y), \cos(2\pi\sigma_L y)]. \qquad (18)$$

This encoding enables the MLP to learn high-frequency spatial patterns that would otherwise be difficult due to the spectral bias of neural networks toward low frequencies (Rahaman et al., 2019).

**Spatial Smoothness Regularization.** To further encourage spatially coherent outputs, we incorporate a regularizer on the spatial structure of the HF field. The choice of regularizer should be informed by prior knowledge of the expected HF characteristics:

For systems where the high-frequency correction is expected to exhibit *smooth* spatial structure (e.g., gentle gradients or diffusive processes), we employ a Laplacian regularizer that penalizes curvature:

$$\mathcal{R}_{\text{lap}}^{(\ell)} = \frac{1}{|\Omega|} \sum_{(i,j)\in\Omega} \left|\nabla^2 u_{\text{HF}}^{(\ell)}(i, j)\right|^2, \qquad (19)$$

where $\nabla^2 u = u_{i+1,j} + u_{i-1,j} + u_{i,j+1} + u_{i,j-1} - 4u_{i,j}$ is the discrete Laplacian. This allows sharp but smooth features while suppressing spurious high-frequency oscillations.

Conversely, for systems where the HF correction contains *sharp discontinuities* or localized features (e.g., shock fronts, edges, or concentrated anomalies), the Laplacian regularizer would inappropriately penalize physically meaningful curvature. In such cases, a gradient-based total variation regularizer $\mathcal{R}_{\text{TV}}^{(\ell)} = \sum_{i,j} |\nabla u_{\text{HF}}^{(\ell)}(i, j)|$ preserves edges while promoting piecewise smoothness, or an edge-preserving bilateral formulation using Huber loss may be employed. The appropriate regularizer should be selected based on domain knowledge of the underlying physics; we provide detailed formulations in Appendix G.5.

**Full HF Layer Loss.** The training objective for layer $\ell$ is:

$$\mathcal{L}^{(\ell)} = \mathcal{L}_{\text{sensor}}^{(\ell)} + \lambda_{\text{sp}}\mathcal{R}_{\text{sparse}}^{(\ell)} + \lambda_{\text{topk}}\mathcal{R}_{\text{topk}}^{(\ell)}$$
$$+ \lambda_{\text{sm}}\mathcal{R}_{\text{smooth}}^{(\ell)} + \lambda_{\text{mag}}\mathcal{L}_{\text{mag}}^{(\ell)}, \qquad (20)$$

where $\mathcal{L}_{\text{sensor}}^{(\ell)} = \|\mathbf{M}\tilde{\mathbf{u}}_{\text{HF}}^{(\ell)} - \mathbf{r}^{(\ell)}\|_2^2$ is the sensor matching loss, $\mathcal{R}_{\text{smooth}}^{(\ell)}$ is the spatial regularizer, and $\mathcal{L}_{\text{mag}}^{(\ell)} =$

$[\max(0, \|\tilde{\mathbf{u}}_{\mathrm{HF}}^{(\ell)}\|_1 - \tau)]^2$ is a magnitude constraint preventing the HF component from dominating. When $k_\ell = \infty$, we set $\lambda_{\mathrm{topk}} = 0$.

### 3.5. Training Pipeline

The complete training procedure proceeds in three stages. *Stage 1* (SHRED on Simulation): Train the base SHRED model (LSTM encoder + decoder) on simulation data with either sparse sensor histories or full-state supervision to learn the dominant dynamics and establish a smooth spatial decoder. *Stage 2* (Latent Alignment): Train the latent transformation $\mathcal{G}$ and discriminator $\mathcal{D}$ via adversarial learning to align simulation and ground truth latent distributions. *Stage 3* (Hierarchical HF Peeling): For each layer $\ell = 1, \ldots, N_{\mathrm{peel}}$, freeze all parameters except $\theta_{\mathrm{HF}}^{(\ell)}$, determine $k_\ell$ adaptively via spectral analysis (Eqs. 14–15), train with a warmup schedule where $\lambda_{\mathrm{sp}} = 0$ for the first $E_{\mathrm{warm}}$ epochs before ramping linearly to the target value, identify dominant frequencies after training and add them to the exclusion set $\mathcal{E}^{(\ell)}$ for subsequent layers, then fine-tune with reduced sparsity weight $\lambda_{\mathrm{sp}}' = 0.1\lambda_{\mathrm{sp}}$.

**Inference** Given sensor history $\mathbf{S}_{t-L:t}'$ and current measurements $\mathbf{s}'(t)$, inference proceeds as follows. The LF pathway computes $\mathbf{z}_{\mathrm{LF}} = \mathcal{E}_{\mathrm{LF}}(\mathbf{S}_{t-L:t}')$, applies the learned alignment $\tilde{\mathbf{z}}_{\mathrm{LF}} = \mathbf{z}_{\mathrm{LF}} + \gamma \cdot \mathcal{G}(\mathbf{z}_{\mathrm{LF}})$, and decodes $\mathbf{u}^{(0)} = \mathcal{D}_{\mathrm{LF}}(\tilde{\mathbf{z}}_{\mathrm{LF}})$. The HF pathway then iteratively refines the reconstruction: for $\ell = 1, \ldots, N_{\mathrm{peel}}$, compute residual $\mathbf{r}^{(\ell)} = \mathbf{s}'(t) - \mathbf{M}\mathbf{u}^{(\ell-1)}$ and update $\mathbf{u}^{(\ell)} = \mathbf{u}^{(\ell-1)} + \mathcal{H}^{(\ell)}(\mathbf{r}^{(\ell)})$. The final reconstruction is $\tilde{\mathbf{u}}'(t) = \mathbf{u}^{(N_{\mathrm{peel}})}$.

Further architectural details, hyperparameter configurations, and model complexity analysis are provided in Appendix G.

## 4. Results

We evaluate the SENDAI framework across six geographically diverse study sites, assessing reconstruction of heterogeneous spatiotemporal NDVI fields from sparse sensor measurements. The evaluation encompasses two architectural variants: (i) the simplified SENDAI Jr. pipeline for sites with primarily low-frequency discrepancies, and (ii) the full SENDAI hierarchical multiscale architecture for sites requiring high-frequency correction. All experiments employ 64 sensors (~1.5% of pixels). We adopt the Structural Similarity Index Measure (SSIM) as our primary performance metric, as it captures the preservation of spatial patterns and topological structure that RMSE alone cannot assess—a smoothed reconstruction may achieve reasonable RMSE while completely destroying field boundaries and land cover discontinuities.

Beyond NDVI, we demonstrate multi-domain generalizability by additionally evaluating SENDAI on land surface

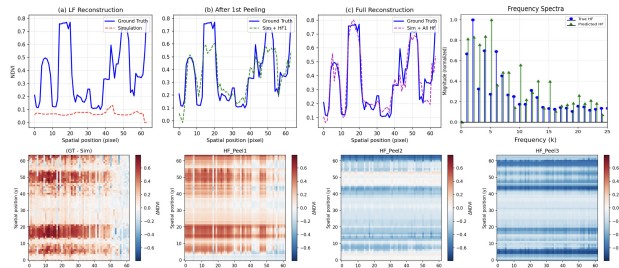

*Figure 3.* Hierarchical frequency peeling on a 1D NDVI slice from the Tarim Basin site. Top row presents single time-point reconstruction; bottom row presents spatiotemporal HF decomposition.

temperature, surface moisture, and seismic waveform reconstruction (Section 4.6).

### 4.1. Synthetic Analysis

Before evaluating full 2D NDVI reconstruction, we validate the HFP methodology on controlled systems where ground truth decomposition is known analytically. We present two complementary experiments: (i) a synthetic traveling wave system with analytically specified frequencies, and (ii) a 1D slice extracted from real MODIS NDVI data where simulation and ground truth exhibits clear spectral discrepancy.

**Traveling Wave System.** We construct a spatiotemporal field composed of three distinct traveling waves.

$$u(x,t) = \sum_{n=1}^{3} A_n \sin(k_n x - \omega_n t) \tag{21}$$

Figure 12 compares two HF correction strategies: joint discovery versus hierarchical peeling. Both achieve comparable reconstruction accuracy in terms of RMSE, but the joint approach produces a relatively noisy frequency spectrum with energy spread across multiple modes beyond the targets (panel a, rightmost), whereas hierarchical peeling yields clean outputs for each mode (panel b, rightmost). This spectral purity translates to improved fine-scale reconstruction, particularly in regions where the two modes interfere constructively or destructively. The decomposition also enables interpretable analysis of individual frequency contributions—critical for applications where different spectral components have distinct physical origins. Additionally, the hierarchical strategy also exhibits more stable training dynamics and supports modular refinement. Full experimental details and extended analysis is provided in Appendix H.

**NDVI 1D Validation.** To validate hierarchical peeling on remote sensing data with inherent noise, outliers, and nonstationary dynamics, we extract a 1D slice from the Tarim Basin site (Figure 13).

Despite the HF residual exhibits widespread energy, hierarchical peeling successfully captures the majority of the modes (Figure 3) while producing interpretable components that correspond to distinct spatialtemporal scales of the

landscape. The first layer ($HF_1$) captures coherent spatio-temporal variability consistent with climate-driven phenological offsets (temperature/episodic weather forcing) modulated by elevation, while the second layer ($HF_2$) isolates a largely time-invariant spatial gradient suggestive of persistent hydrological controls (e.g. water availability) that differ between spring simulations and summer–autumn observations. A third temporally stable component ($HF_3$) plausibly reflects edaphic heterogeneity (e.g. texture, salinity, nutrient level) that the low-frequency model cannot resolve.

Success on data without clean sinusoidal patterns demonstrates that hierarchical peeling is robust to realistic data imperfections. Extended analysis is provided in Appendix H.

## 4.2. Baseline Methods Performance

We establish baseline performance using three geostatistical methods for spatiotemporal reconstruction—Savitzky-Golay filtering with IDW (SG+IDW), Harmonic Analysis of Time Series with IDW (HANTS+IDW), and Kriging—as well as MMGN, a recent implicit neural representation approach. Figure 4 illustrates baseline reconstruction quality on the Tarim Basin site, which presents challenging heterogeneous spatial structure due to sharp mountain-basin boundaries. A critical observation emerges: baseline methods fundamentally fail to preserve the topological structure of spatial patterns despite achieving moderate RMSE values. These deficiencies underscore the importance of SSIM (structural similarity index measure) for evaluating structural preservation. MMGN, despite its strong performance on climate and oceanographic datasets, exhibits comparable limitations under our extreme sparsity regime (64 sensors, 1.56% coverage) and heterogeneous fine-scale landscapes.

We additionally evaluate the three deep learning baselines on the Tarim Basin site. FNO suffers from both input-format mismatch under sparse sensing and sim2real domain shift, achieving SSIM no higher than 0.345 even with masked training. DeepONet and Senseiver, adapted to the sparse sensing setting and trained on simulation, achieve SSIM 0.401 and 0.408 respectively. When augmented with SENDAI's hierarchical multi-scale architecture, DeepONet-SENDAI reaches SSIM 0.477 and Senseiver-SENDAI reaches 0.467 (Table 2, Figure 5)—approaching but still below SENDAI's full pipeline (see Section 4.4). These results confirm that SENDAI's contributions—latent-space alignment and hierarchical frequency peeling—provide key mechanisms that enable robust multi-scale reconstruction under domain shift from hyper-sparse sensor measurements, and are complementary to, rather than dependent on, the particular choice of spatial reconstruction unit. Full experimental details and discussions are provided in Appendix E.5. Table 1 compares the architectural capabilities of SENDAI against these deep learning baselines. While each addresses

*Table 1.* Capability comparison of SENDAI and some related methods for sparse-measurement spatiotemporal field reconstruction under domain shift. ✓ = supported; ✗ = not supported.

| Method | Sparse spatial sensing | Temporal seq. encoding | Sim2real alignment | Multiscale architecture |
|---|---|---|---|---|
| SENDAI | ✓ | ✓ | ✓ | ✓ |
| SENDAI Jr. | ✓ | ✓ | ✓ | ✗ |
| SHRED-ROM | ✓ | ✓ | ✗ | ✗ |
| Senseiver | ✓ | ✗ | ✗ | ✗ |
| FNO[†] | ✓ | ✗ | ✗ | ✗ |
| DeepONet[§] | ✓ | ✗ | ✗ | ✗ |

[†] Standard FNO requires full spatial fields; extensions such as RecFNO address spatial sparsity but lack the remaining capabilities.

[§] Standard DeepONet maps explicit PDE parameters or full-field inputs to solution fields; a sensor-input adaptation was benchmarked by (Tomasetto et al., 2025) but still lacks temporal encoding, domain alignment, and multiscale decomposition.

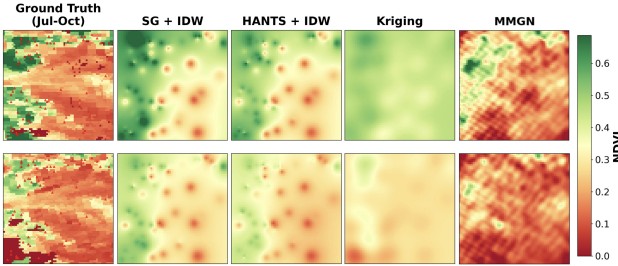

*Figure 4.* Baseline reconstruction comparison for the Tarim Basin site. Red markers indicate sensor locations.

spatial reconstruction, only SENDAI supports all four capabilities jointly: hyper-sparse spatial sensing, temporal sequence encoding, sim2real alignment, and multiscale decomposition.

## 4.3. SENDAI Jr. Reconstruction Performance

For sites where seasonal domain shift manifests predominantly as low-frequency distributional changes, the two-stage alignment alone from SENDAI (SENDAI Jr.) provides effective reconstruction. Implementation details are in Appendix G.2. Table 3 summarizes the performance.

SENDAI Jr. achieves substantial SSIM improvements over all baselines: +120% for Central Valley, +185% for Corn Belt, and +98% for Guadalquivir Valley. These improvements demonstrate that SENDAI Jr. successfully preserves spatial topology—field boundaries, vegetation gradients, and land cover patterns—that interpolation-based methods systematically destroy. RMSE improvements confirm quantitative accuracy alongside structural fidelity.

*Table 2.* Deep learning baseline comparison on the Tarim Basin site (mean ± std, 3 runs). Adapted variants are trained on simulation and deployed on GT; -SENDAI variants are augmented with SENDAI's hierarchical multi-scale architecture.

| Configuration | RMSE | SSIM |
|---|---|---|
| Adapted DeepONet | $0.170 \pm 0.003$ | $0.401 \pm 0.003$ |
| DeepONet-SENDAI | $0.132 \pm 0.002$ | $0.477 \pm 0.005$ |
| Adapted Senseiver | $0.090 \pm 0.001$ | $0.408 \pm 0.004$ |
| Senseiver-SENDAI | $0.130 \pm 0.005$ | $0.467 \pm 0.021$ |
| **SENDAI (full pipeline)** | $\mathbf{0.125} \pm 0.011$ | $\mathbf{0.489} \pm 0.021$ |

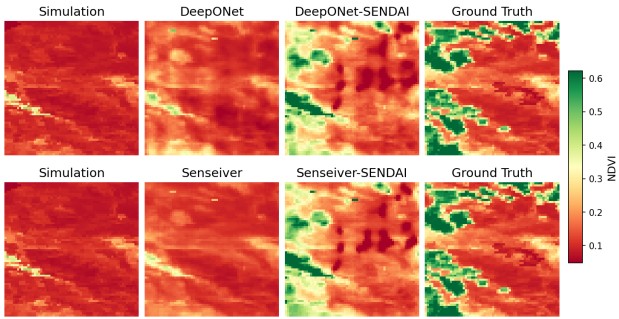

*Figure 5.* Reconstruction snapshots from DeepONet (top) and Senseiver (bottom) on the Tarim Basin site. Both augmented models (DeepONet-SENDAI and Senseiver-SENDAI) capture the broad vegetation patterns, though with visible spatial artifacts that are substantially reduced in SENDAI's outputs (Figure 7). Full comparison figures are provided in Appendix E.5.

Figure 6b presents reconstruction for the Central Valley site. SENDAI Jr. preserves the essential topological structure including the spatial arrangement of high- and low-NDVI regions. Site-specific details are provided in Appendix I.

### 4.4. Full SENDAI Performance

For sites exhibiting complex phenological dynamics and pronounced spatial heterogeneity, we deploy SENDAI with full hierarchical multiscale architecture with multiple peeling layers. Table 4 presents detailed result comparison.

The full SENDAI hierarchical architecture achieves the highest SSIM across all three sites: 0.4668 (Imperial Valley), 0.4777 (Tarim Basin), and 0.3354 (Riverina). These represent substantial improvements over both baselines and SENDAI Jr., with SSIM gains of 15.5% (Imperial Valley), 36.3% (Tarim Basin), and 21.5% (Riverina) from SENDAI Jr. to the full SENDAI pipeline. The Tarim Basin site exhibits the most dramatic improvement, where sharp mountain-basin boundaries require high-frequency corrections that smooth decoders cannot capture.

Figure 7 presents the full hierarchical reconstruction result for the Tarim Basin site, with each row corresponding to an equally spaced temporal frame within the respective simulation and ground truth periods. While all baseline methods fail to resolve sharp boundaries, SENDAI success-

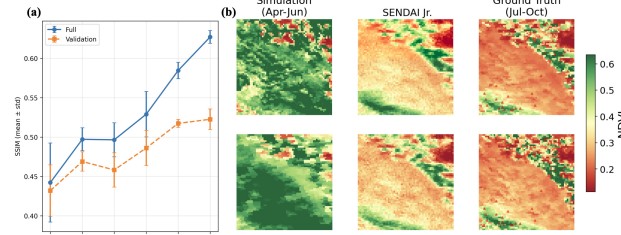

*Figure 6.* (a) Sensor sensitivity results (from five independent runs) on the Tarim Basin site for the full SENDAI framework. (b) SENDAI Jr. reconstruction for the Central Valley site.

*Table 3.* SENDAI Jr. reconstruction performance compared with baselines. The first row quantifies the phenological domain shift. Relative improvement from best baseline to SENDAI Jr. is indicated. Best results per site are **bolded**. All of the experiments below are averaged from five independent runs.

| | | **Central Valley** | **Corn Belt** | **Guadalquivir** |
|---|---|---|---|---|
| Sim. vs. GT | RMSE | 0.1965 | 0.4501 | 0.2387 |
| | SSIM | 0.4751 | 0.0312 | 0.2464 |
| SG + IDW | RMSE | $0.1447 \pm 0.0029$ | $0.1712 \pm 0.0011$ | $\mathbf{0.1444} \pm 0.0023$ |
| | SSIM | $0.2612 \pm 0.0098$ | $0.1588 \pm 0.0107$ | $0.1849 \pm 0.0084$ |
| HANTS + IDW | RMSE | $0.1451 \pm 0.0029$ | $0.1866 \pm 0.0006$ | $0.1496 \pm 0.0022$ |
| | SSIM | $0.2504 \pm 0.0096$ | $0.1498 \pm 0.0107$ | $0.1755 \pm 0.0083$ |
| Kriging | RMSE | $0.1634 \pm 0.0027$ | $0.1596 \pm 0.0020$ | $0.1481 \pm 0.0008$ |
| | SSIM | $0.0922 \pm 0.0132$ | $0.0312 \pm 0.0016$ | $0.0878 \pm 0.0028$ |
| **SENDAI Jr.** | RMSE | $\mathbf{0.1068} \pm 0.0015$ | $\mathbf{0.1103} \pm 0.0075$ | $0.1474 \pm 0.0014$ |
| | SSIM | $\mathbf{0.5747} \pm 0.0478$ | $\mathbf{0.4530} \pm 0.0251$ | $\mathbf{0.3655} \pm 0.0070$ |
| Improvement | | +120.0% | +185.3% | +97.7% |

fully reconstructs these landscape features. The learned HF component exhibits coherent spatial structure aligned with boundaries, confirming that peeling layers discover physically meaningful corrections.

Qualitative examination (Figures 31–36 in Appendix J) reveals consistent pattern reconstructions across sites, compared with baselines. The SENDAI Jr. base reconstruction recovers mesoscale spatial patterns, while the learned HF correction exhibits site-dependent characteristics: coherent structure correlated with field boundaries for Imperial Valley, sharp landscape boundaries for Tarim Basin, and more diffusive structure for Riverina consistent with gradual spatial transitions. Reconstruction quality remains stable across temporal frames, indicating generalizable representations. Site-specific details are provided in Appendix J.

### 4.5. Robustness, Ablation & Computational Efficiency

SENDAI's reconstruction quality is robust to key design choices. Ablation studies on the Tarim Basin site confirm stability across sensor number, temporal lag, maximum target frequency, sensor noise levels, and spatially non-uniform sensor placement. Sensor number sensitivity is shown in Figure 6a; full ablation details are provided in Appendix K.

SENDAI is remarkably efficient computationally. Unlike contemporary deep learning approaches requiring GPU clus-

*Table 4.* SENDAI reconstruction performance compared with baselines. Relative improvements from SENDAI Jr. are indicated. Best results per site are **bolded**. All of the experiments below are averaged from five independent runs.

| | | Imperial Valley | Tarim Basin | Riverina |
|---|---|---|---|---|
| Sim. vs. GT | RMSE | 0.2157 | 0.2077 | 0.2778 |
| | SSIM | 0.3488 | 0.3448 | 0.1115 |
| SG + IDW | RMSE | $0.1599 \pm 0.0054$ | $0.1794 \pm 0.0011$ | $0.1603 \pm 0.0009$ |
| | SSIM | $0.1123 \pm 0.0146$ | $0.1308 \pm 0.0320$ | $0.1359 \pm 0.0053$ |
| HANTS + IDW | RMSE | $0.1603 \pm 0.0054$ | $0.1806 \pm 0.0010$ | $0.1669 \pm 0.0009$ |
| | SSIM | $0.1049 \pm 0.0145$ | $0.1214 \pm 0.0315$ | $0.1245 \pm 0.0064$ |
| Kriging | RMSE | $0.1591 \pm 0.0024$ | $0.2163 \pm 0.0053$ | **0.1495** $\pm 0.0004$ |
| | SSIM | $0.0916 \pm 0.0190$ | $0.0449 \pm 0.0174$ | $0.0272 \pm 0.0025$ |
| MMGN | RMSE | $0.1786 \pm 0.0076$ | $0.1783 \pm 0.0108$ | $0.3233 \pm 0.0104$ |
| | SSIM | $0.0778 \pm 0.0270$ | $0.1226 \pm 0.0448$ | $0.0798 \pm 0.0116$ |
| SENDAI Jr. | RMSE | $0.1708 \pm 0.0029$ | $0.1827 \pm 0.0046$ | $0.1537 \pm 0.0192$ |
| | SSIM | $0.4041 \pm 0.0248$ | $0.3505 \pm 0.0269$ | $0.2761 \pm 0.0427$ |
| SENDAI Jr.+HFP | RMSE | $0.1588 \pm 0.0184$ | $0.1466 \pm 0.0075$ | $0.2526 \pm 0.0361$ |
| | SSIM | $0.4411 \pm 0.0499$ | $0.4257 \pm 0.0159$ | $0.3158 \pm 0.0222$ |
| **SENDAI** | RMSE | **0.1486** $\pm 0.0063$ | **0.1208** $\pm 0.0109$ | $0.1823 \pm 0.0151$ |
| | SSIM | **0.4668** $\pm 0.0390$ | **0.4777** $\pm 0.0205$ | **0.3354** $\pm 0.0202$ |
| Improvement | HFP | +9.2% | +21.5% | +14.4% |
| from SENDAI Jr. | HFP+INR | +15.5% | +36.3% | +21.5% |

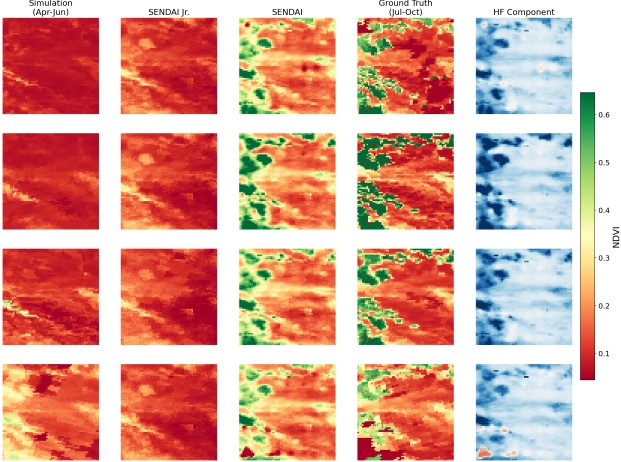

*Figure 7.* Full SENDAI hierarchical multiscale DA-SHRED reconstruction for the Tarim Basin site.

ters (Meraner et al., 2020; Cresson, 2018), SENDAI operates on standard CPU hardware with training times measured in minutes per site. This efficiency, deriving from the shallow architecture and exploitation of Takens' embedding theorem, enables full-state reconstruction from low-dimensional sensor histories without requiring dense spatial supervision. Detailed operational scenarios and quantitative analysis are provided in Appendix L.

### 4.6. Multi-Domain Generalizability

To demonstrate that SENDAI generalizes beyond NDVI, we apply the framework—with identical model architecture, hyperparameters, and training pipeline—to two additional remote sensing variables: land surface temperature (LST) and Land Surface Water Index (LSWI); and one fundamentally different physical domain: seismic waveform reconstruction. Across remote sensing variables, a consistent

pattern emerges: the architectural requirement is governed by landscape complexity rather than variable-specific properties. The seismic experiment—686 globally distributed broadband stations recording a M 7.1 earthquake—tests a regime where the LF pathway contributes negligibly, yet the HF pathway autonomously takes over without architectural modification, confirming that SENDAI's hierarchical design adapts to the data. Full details are provided in Appendix M.

## 5. Conclusions and Future Directions

The experimental results show that SENDAI robustly reconstructs heterogeneous spatiotemporal NDVI fields across diverse geographic and climatic settings, adapting from simulation to ground-truth observations across seasonal boundaries while recovering mesoscale patterns and fine-scale structure via hierarchical peeling. Performance depends on landscape complexity: SENDAI Jr. suffices for smoother fields with persistent spatial structure, whereas sites with sharp boundaries and sub-seasonal dynamics benefit from the full SENDAI framework with hierarchical structure.

The methodological contributions extend beyond NDVI reconstruction. SENDAI provides a general template for sparse-observation reconstruction of heterogeneous spatiotemporal fields: establish spatial priors from data-rich reference sources, adapt to sparse-measurement targets via latent-space alignment, and employ hierarchical frequency peeling when fine-scale corrections are needed. As demonstrated in Section 4.6, this template transfers directly to land surface temperature, surface moisture, and seismic waveform reconstruction, and may extend further to soil moisture (Mohanty et al., 2017), snow dynamics (Lievens et al., 2019), flood mapping (Schumann & Moller, 2015), and bandwidth-constrained settings (e.g., deep-space monitoring (De Cola et al., 2011)) wherever coherent fields, reference data, and sparse target observations are available. Additional discussion is provided in Appendix N.

Several limitations and future directions remain. First, SENDAI assumes stationary spatial structure between simulation and ground truth periods, which may break under land cover change, management shifts, or disturbances; extending the framework to non-stationary settings could be an important next step. Second, sensor placement could be improved through information-theoretic design (Bao & Kutz, 2025b; Santos et al., 2023), potentially reducing sensor requirements. Finally, in many operational settings, observations are not only sparse in space but also in time, confined to short temporal windows due to satellite revisit constraints, post-event data availability, or communication blackouts, etc. Extending the framework to reconstruct full spatiotemporal trajectories from such limited observation windows is a promising direction.

## Impact Statement

In this paper we investigate the applications of Machine Learning to the frontier of remote sensing. The SENDAI framework enables accurate heterogeneous spatiotemporal reconstruction and domain adaptation from severely sparse sensor observations, with potential benefits for environmental monitoring, space exploration, disaster response, and climate analysis, particularly in resource-constrained areas where dense observations are expensive or unavailable.

## Acknowledgments

This work was supported in part by the US National Science Foundation (NSF) AI Institute for Dynamical Systems (dynamicsai.org), grant 2112085. JNK further acknowledges support from the Air Force Office of Scientific Research (FA9550-24-1-0141).

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

## A. MODIS Platform Details

The Moderate Resolution Imaging Spectroradiometer (MODIS) is a key instrument aboard NASA's Terra and Aqua satellites, launched in 1999 and 2002 respectively, providing continuous global observations for over two decades (Justice et al., 2002). MODIS acquires data in 36 spectral bands ranging from 0.405 to 14.385 $\mu$m, with spatial resolutions of 250 m, 500 m, and 1000 m depending on the band. The instrument's 2330 km viewing swath width enables complete global coverage every one to two days, making it uniquely suited for monitoring dynamic land surface processes at regional to continental scales.

The Terra satellite follows a descending sun-synchronous orbit with a 10:30 AM local equatorial crossing time, while Aqua follows an ascending orbit with a 1:30 PM crossing. This complementary configuration provides up to two observations per day for any given location, significantly increasing the probability of obtaining cloud-free imagery. In this study, we utilize the Collection 6.1 daily surface reflectance products (MOD09GA from Terra and MYD09GA from Aqua), which provide atmospherically corrected surface reflectance values processed using the Second Simulation of the Satellite Signal in the Solar Spectrum (6S) radiative transfer model.

The Normalized Difference Vegetation Index (NDVI) exploits the distinctive spectral signature of photosynthetically active vegetation, which strongly absorbs red light for photosynthesis while reflecting near-infrared (NIR) radiation due to leaf cellular structure (Tucker, 1979). NDVI values typically range from approximately $-0.1$ for water bodies and bare soil to 0.8–0.9 for dense, healthy vegetation, responding sensitively to changes in chlorophyll content, leaf area, and vegetation fraction (Huete et al., 2002).

## B. Data Processing Pipeline Details

All MODIS imagery was acquired through Google Earth Engine (GEE) (Gorelick et al., 2017), a cloud-based platform providing direct access to petabyte-scale geospatial archives. For each study site, imagery from both Terra and Aqua sensors was merged to maximize temporal density, yielding up to two potential observations per day.

Cloud contamination is addressed using the `state_1km` quality assurance band, retaining only pixels flagged as "clear" (bit pattern 00 in bits 0–1). Images with less than 70% valid pixel coverage are excluded to ensure spatial coherence. For each study site, we define a 15 km $\times$ 15 km square region centered on representative coordinates, resampled to a standardized $64 \times 64$ pixel grid at fine resolution. This fixed grid dimension facilitates consistent architecture across study sites and enables direct comparison of model performance.

To ensure consistent temporal density while accounting for variable cloud cover, we employ an equally-spaced sampling strategy targeting 70–90 valid images per approximately 90-day observation period. In practice, persistent cloud cover at several sites (particularly tropical and monsoon-affected regions) limited acquisition to 40–50 valid images per period, testing the framework's robustness under reduced temporal sampling.

For each study site, the data generation pipeline produces simulation data (from one seasonal period) and ground truth data (from a different seasonal period). The simulation data serves to train the base SHRED architecture, learning a latent representation of the state space structure. The ground truth data provides sensor-only observations for the SENDAI, where the model learns to adapt its latent space to the distributional shift between seasons while reconstructing full-state fields from sparse measurements.

## C. Study Site Descriptions

### C.1. Western United States: Central Valley, California

The Central Valley represents one of the world's most productive agricultural regions, characterized by Mediterranean climate (Köppen Csa) with hot, dry summers and mild, wet winters. The landscape is dominated by irrigated permanent crops (orchards, vineyards) and annual field crops. The simulation period (April–June) captures spring green-up and early crop development, while the ground truth period (July–October) spans peak summer productivity through early senescence. The strong phenological signal and relatively cloud-free conditions make this site suitable for the standard SENDAI Jr. architecture.

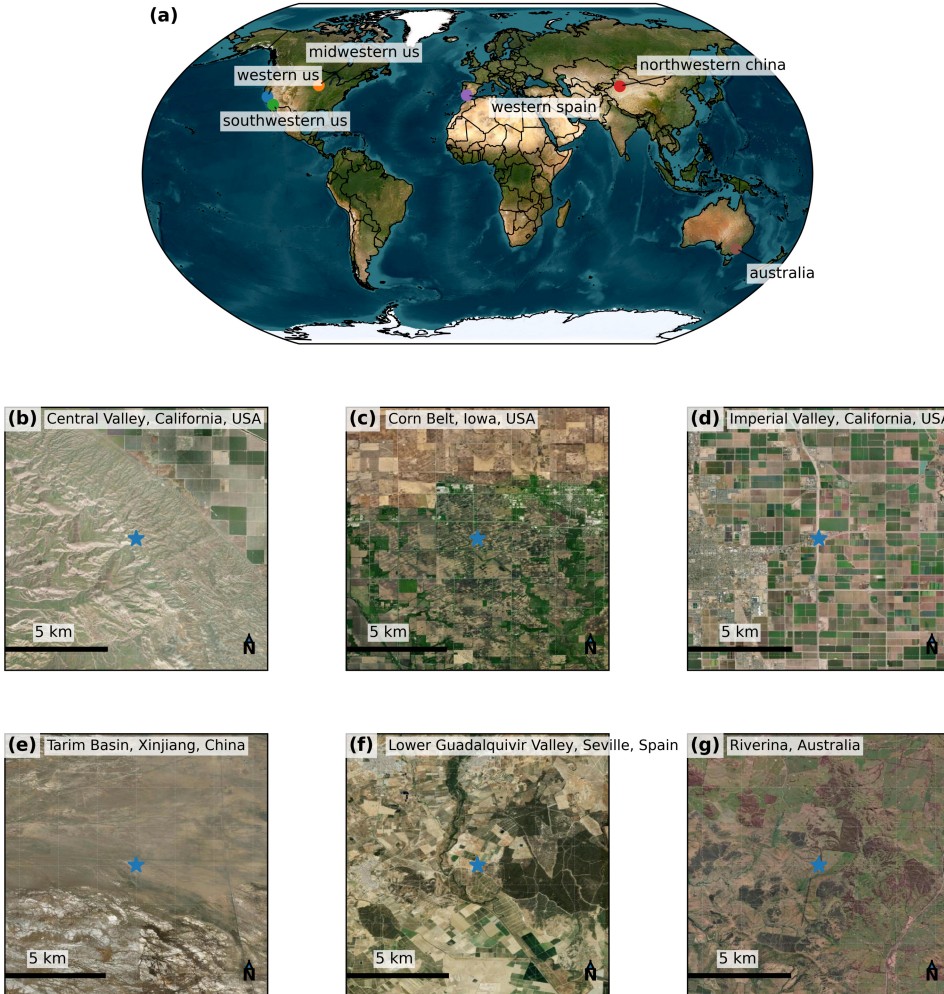

*Figure 8.* Study sites for NDVI reconstruction experiments. (a) Global distribution of the eight study areas spanning North America, South America, Europe, Asia, and Australia. (b–h) Local 32 km × 32 km windows centered at each site; stars denote site centers. The basemap imagery shown is a rolling mosaic for geographic context only. Administrative boundaries are from Natural Earth; imagery tiles are from Esri World Imagery or Sentinel-2 cloudless composites.

## C.2. Midwestern United States: Iowa Corn Belt

The Iowa Corn Belt exemplifies temperate continental agriculture (Köppen Dfa) with pronounced seasonal temperature variation. Land cover is dominated by corn–soybean rotations under rainfed conditions. The April–June simulation period captures emergence and vegetative growth, while July–October encompasses reproductive stages through harvest. Despite the dramatic phenological transition, the relatively predictable cropping calendar and absence of irrigation artifacts allow reconstruction with standard SENDAI Jr.

## C.3. Southwestern United States: Imperial Valley, California

Imperial Valley presents an extreme hot desert environment (Köppen BWh) where agriculture is entirely dependent on irrigation from the Colorado River. The sharp contrast between verdant irrigated fields and surrounding bare desert creates pronounced spatial heterogeneity. Multiple cropping cycles per year and variable irrigation schedules introduce high-frequency temporal dynamics requiring the full SENDAI architecture to capture sub-seasonal variability.

*Table 5.* Study site characteristics and observation periods. Sites are grouped by the architecture variant employed: SENDAI Jr. (simpler seasonal transitions) or SENDAI (complex phenological dynamics requiring high-frequency correction). All data are from 2023. Climate classifications follow the Köppen system: Csa = hot-summer Mediterranean (dry summers, wetter winters); Dfa = hot-summer humid continental (no dry season, cold winters); BWh = hot desert (extremely arid, very hot); BWk = cold desert (arid but cooler, with colder winters); Cfa = humid subtropical (no dry season, hot humid summers, mild winters).

| Region ID | Location Name | Center (Lon, Lat) | Climate | Land Cover | Sim. | GT | Model |
|-----------|---------------|-------------------|---------|------------|------|-----|-------|
| *North America* | | | | | | | |
| western_us | Central Valley, CA | $(-120.5, 36.5)$ | Csa | Irrigated cropland | Apr–Jun | Jul–Oct | SENDAI Jr. |
| midwestern_us | Corn Belt, IA | $(-93.5, 42.0)$ | Dfa | Rainfed cropland | Apr–Jun | Jul–Oct | SENDAI Jr. |
| southwestern_us | Imperial Valley, CA | $(-115.5, 32.8)$ | BWh | Irrigated cropland | Apr–Jun | Jul–Oct | SENDAI |
| *Europe* | | | | | | | |
| western_spain | Guadalquivir Valley | $(-6.25, 37.25)$ | Csa | Mixed agriculture | Feb–Apr | Sep–Dec | SENDAI Jr. |
| *Asia* | | | | | | | |
| northwestern_china | Tarim Basin | $(83.5, 41.5)$ | BWk | Oasis agriculture | Apr–Jun | Jul–Oct | SENDAI |
| *Southern Hemisphere* | | | | | | | |
| southeasthern_australia | Riverina | $(147.5, -35.5)$ | Cfa | Mixed cropping | Feb–Apr | Sep–Dec | SENDAI |

## C.4. Western Spain: Lower Guadalquivir Valley

The Guadalquivir Valley exhibits Mediterranean climate (Köppen Csa) with a distinctive reversed phenological calendar compared to northern hemisphere temperate regions—vegetation green-up occurs during mild, wet winters rather than spring. The February–April simulation period captures peak winter greenness and early spring drying, while September–December represents the onset of the growing season following summer drought. This phenological inversion provides a critical test of domain adaptation capability.

## C.5. Northwestern China: Tarim Basin

The Tarim Basin represents a hyper-arid cold desert environment (Köppen BWk) surrounded by the Tianshan and Kunlun mountain ranges. Agriculture is concentrated in narrow oasis strips fed by glacial meltwater, creating extreme spatial gradients between irrigated fields and surrounding desert. The April–June simulation period captures spring irrigation onset and crop establishment, while July–October encompasses peak productivity. The localized nature of vegetation and strong background contrasts require hierarchical frequency decomposition.

## C.6. Southeastern Australia: Riverina Region

The Riverina represents Australia's "food bowl", featuring humid subtropical climate (Köppen Cfa) transitional to semi-arid conditions. Mixed cropping systems include irrigated rice, wheat, and pasture. The Southern Hemisphere location provides reversed seasonality, with February–April capturing late summer/autumn conditions and September–December spanning spring green-up. This site tests generalization to different hemispheric phenological timing and drought-prone conditions.

## D. Seasonal Split Strategy and Sparse Sensor Configuration Details

Table 5 presents study site characteristics and observation periods, motivated by phenological domain shift. Together, they exhibit phenological coherence but heterogeneous spatiotemporal transitions in NDVI: vegetation dynamics vary not only across space but also exhibit distinct temporal autocorrelation structures between seasons (due to land cover heterogeneity, soil moisture gradients, and management practices, etc). For Southern Hemisphere sites and Mediterranean climates with reversed seasonality, the seasonal splits capture analogous phenological transitions adapted to local climate rhythms. The SENDAI framework must therefore learn to reconstruct fields that are simultaneously spatially heterogeneous and temporally non-stationary.

A central tenet of the SENDAI architecture is that full-state reconstruction can be achieved from sparse temporal observations at a limited number of spatial locations, leveraging Takens' embedding theorem and learned decoder representations (Williams et al., 2024; Bao & Kutz, 2025a). We employ $p = 64$ randomly placed sensors, representing 1.56% of the full state space ($64 \times 64 = 4096$ pixels)—comparable to or lower than typical ground-based monitoring network densities.

Sensor locations are randomly selected, excluding boundary pixels (2-pixel buffer). Following the SHRED formulation, sensor observations are organized into time-delay embeddings of length $L = 5$ lags, where at each time step $t$, the input to the encoder consists of the sensor history $\mathbf{S}_t = [\mathbf{s}_{t-L+1}, \ldots, \mathbf{s}_t]^\top \in \mathbb{R}^{L \times p}$, enabling the model to infer temporal derivatives and phenological trends critical for tracking dynamic vegetation changes.

## E. Baseline Methods Details

### E.1. SG + IDW Implementation

Savitzky-Golay filtering is applied with window length 7 and polynomial order 2, providing local temporal smoothing while preserving phenological trends. Inverse Distance Weighting uses power parameter $p = 2$ (standard inverse-square weighting). This baseline represents the computational floor—no training required, sub-second inference.

### E.2. HANTS + IDW Implementation

HANTS fits a truncated Fourier series with 3 harmonic terms (annual, semi-annual, and quarterly cycles) at each sensor location. The fitting procedure uses iterative refinement to reject cloud-contaminated observations. Spatial interpolation follows the same IDW protocol as SG+IDW.

### E.3. Kriging Implementation

Gaussian Process regression employs an RBF (squared exponential) kernel with automatic relevance determination. For Kriging-GT, lengthscale and variance hyperparameters are optimized via maximum likelihood on the ground truth sensor observations at each timestep. For Kriging-Sim, hyperparameters are pre-fitted on simulation full fields and held fixed during ground truth reconstruction, providing an intentionally favorable setting analogous to the use of simulation for encoder-decoder pretraining.

### E.4. MMGN Implementation

The Multiplicative and Modulated Gabor Network (MMGN) (Luo et al., 2024) represents a recent advance in implicit neural representations (INRs) for continuous field reconstruction from sparse observations. We implement MMGN following the original architecture and training protocol to provide a rigorous comparison with neural network-based approaches.

**Architecture.** MMGN employs an auto-decoder architecture that learns a continuous spatial representation conditioned on per-timestep latent codes. The decoder consists of multiplicative layers combining Gabor filters with bilinear fusion of coordinates and latent information:

$$\mathbf{h}^{(0)} = g_0(\mathbf{x}) \odot \mathcal{F}_0(\mathbf{0}, \mathbf{z}), \quad \mathbf{h}^{(\ell+1)} = g_{\ell+1}(\mathbf{x}) \odot \mathcal{F}_{\ell+1}(\mathbf{h}^{(\ell)}, \mathbf{z}), \tag{22}$$

where $g_\ell(\mathbf{x})$ are Gabor filters applied to spatial coordinates, $\mathcal{F}_\ell$ are bilinear fusion layers, and $\mathbf{z} \in \mathbb{R}^{d_z}$ is a learnable latent code for each time instance. The Gabor filters take the form:

$$g_\ell(\mathbf{x}) = \sin(\mathbf{W}_g\mathbf{x} + \mathbf{b}_g) \odot \exp\left(-\frac{\gamma}{2}\|\mathbf{x} - \boldsymbol{\mu}\|^2\right), \tag{23}$$

with learnable centers $\boldsymbol{\mu}$ and bandwidth parameters $\gamma$ sampled from a Gamma distribution. This multiplicative structure enables the network to represent the output as a linear combination of Gabor basis functions, providing shift-invariance properties beneficial for spatial field reconstruction.

**Adaptation to Our Setting.** The original MMGN was evaluated on climate simulation data (CESM2 global surface temperature, $192 \times 288$ resolution, 1024 timesteps) and satellite sea surface temperature (GHRSST, $901 \times 1001$ resolution, 360 timesteps) with sensor coverage ranging from 5% to 50%. Our MODIS NDVI reconstruction task presents four key differences: (i) substantially sparser observations (1.56% vs. 5–50%), (ii) smaller spatial extent ($64 \times 64$ pixels), (iii) significantly shorter temporal sequences ($\sim$70 timesteps vs. 360–1024), and (iv) heterogeneous agricultural landscapes with sharp boundaries rather than smoothly varying oceanographic or atmospheric fields.

We apply MMGN with identical sensor configurations (64 randomly placed sensors) and temporal splits as all other methods. The model learns separate latent codes for each timestep, with training performed on sensor observations only and evaluation on full-field reconstruction.

*Table 6.* FNO baseline on the Tarim Basin site (mean $\pm$ std, 3 runs).

| Experiment | Condition | RMSE | SSIM |
|---|---|---|---|
| i (vanilla) | Sim val | $0.080 \pm 0.000$ | $0.769 \pm 0.005$ |
| | GT sparse | $0.217 \pm 0.002$ | $0.125 \pm 0.010$ |
| ii (masked) | Sim val | $0.089 \pm 0.002$ | $0.544 \pm 0.008$ |
| | GT masked | $0.170 \pm 0.003$ | $0.345 \pm 0.010$ |
| SENDAI (full pipeline) | | $\mathbf{0.125} \pm 0.011$ | $\mathbf{0.489} \pm 0.021$ |

### E.5. Deep Learning Baselines

This subsection details the three deep learning baselines summarized in Section 2.3: FNO, DeepONet, and Senseiver. For each, we describe the adaptation to our sparse sensing setting, evaluate standalone and SENDAI-augmented configurations on the Tarim Basin site, and report results averaged over three independent runs. Table 9 reports parameter counts, training times, and per-sample inference latencies for these methods; SENDAI remains competitive in computational cost while achieving the highest reconstruction quality.

#### E.5.1. NEURAL OPERATOR BASELINES

As shown in Table 1, only SENDAI supports all four capabilities: hyper-sparse spatial sensing, temporal sequence encoding, sim2real alignment, and multiscale decomposition. Neural operator methods—including FNO (Li et al., 2020) and DeepONet (Lu et al., 2021)—are powerful frameworks for learning mappings between function spaces, but face structural mismatches with the SENDAI setting. Standard FNO requires full spatial fields as input, not hyper-sparse point measurements; neither FNO nor DeepONet provides a mechanism to bridge the distributional shift between simulation and ground truth; and neither supports a multiscale architecture. Adapting them to this setting would require modifications that would substantially alter their architectures, converging toward the temporal encoding and data assimilation design that SENDAI provides.

We evaluate both operators empirically on the Tarim Basin site to provide concrete context. All experiments still use 64 sensors (1.56% coverage) averaged over three independent runs.

**FNO baseline.** We consider two variants (Table 6). In Experiment i (*vanilla*), FNO is trained on full simulation fields and tested on 64 GT (ground truth) sensor values placed on a zero-filled grid—an input format never seen during training. This input distribution mismatch causes near-complete spatial collapse (SSIM 0.125), as the FNO cannot generalize from full-field inputs to sparse point observations. In Experiment ii (*masked*), FNO is trained on zero-filled sensor grids with an explicit binary mask channel, removing the input-format mismatch and isolating domain shift as the primary failure mode. The masked FNO achieves reasonable in-distribution performance (SSIM 0.544), confirming that FNO *can* learn sparse-to-full reconstruction. However, on GT sensor inputs in the identical format, SSIM drops to 0.345, suggesting that predictions are spatially coherent but reproduce simulation-period spatial structure, failing to capture seasonal vegetation gradients present in the GT (Figure 9). This degradation is attributable to the sim2real domain shift, which FNO has no mechanism to address.

**DeepONet baseline.** To test whether SENDAI's data assimilation framework can accommodate a different spatial decoder architecture, we replace the LSTM temporal unit and MLP decoder of SENDAI's LF pathway with a DeepONet (Lu et al., 2021) adapted to the sparse sensing setting: the branch net takes the same time-delay sensor input as SENDAI, and the trunk net maps spatial coordinates with Fourier positional encoding to learned basis functions. We evaluate two configurations (Table 7). The adapted DeepONet itself trained on simulation and deployed on GT achieves SSIM 0.401—comparable to the masked FNO and well below SENDAI—confirming that neural operators without domain adaptation cannot bridge the sim2real gap. However, when augmented with full SENDAI's hierarchical multi-scale architecture (*DeepONet-SENDAI*), the model reaches SSIM 0.477, closely matching SENDAI's SHRED-based pipeline (0.489). Figure 10 illustrates the progressive improvement qualitatively.

These results carry three implications. First, neither FNO nor DeepONet bridges the sim2real domain shift without additional adaptation, confirming that learned spatial representations alone are insufficient when training and deployment differ in distribution. Second, SENDAI's data assimilation components could transfer to a different spatial decoder, demonstrating

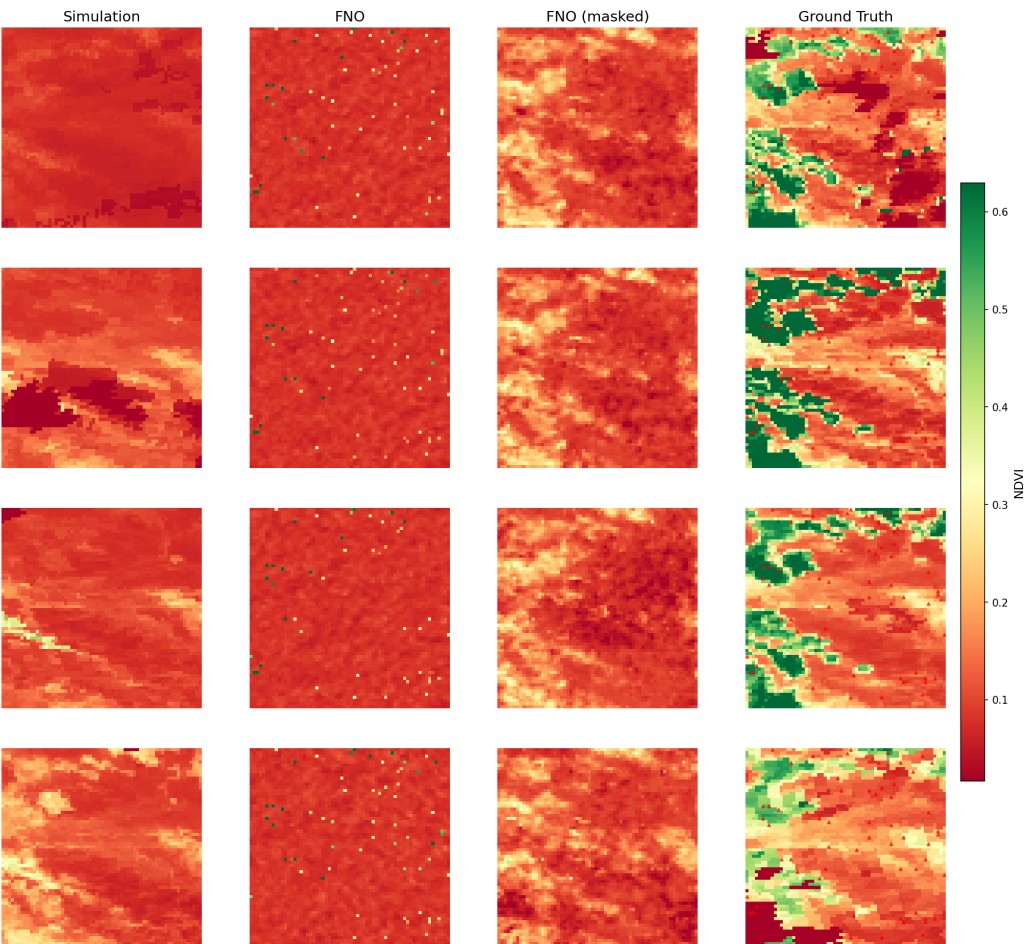

*Figure 9.* FNO reconstruction on the Tarim Basin site. Columns: Simulation, FNO (vanilla, Exp. i), FNO (masked, Exp. ii), Ground Truth.

that the framework is complementary to, rather than competing with, neural operator architectures. Third, while DeepONet-SENDAI achieves comparable SSIM, its reconstructions still exhibit visible spatial artifacts (vertical banding in Figure 10) that are substantially reduced in SENDAI's outputs (Figure 7).

### E.5.2. ATTENTION-BASED SPARSE SENSING BASELINE

**Senseiver baseline.** Distinct from the neural operator methods above, Senseiver (Santos et al., 2023) is an attention-based Perceiver IO architecture that uses cross-attention to encode sparse sensor observations into a fixed-size latent array and decode at arbitrary query locations—a set-to-field reconstruction paradigm rather than the function-space mappings of FNO or DeepONet. We adapt the official Senseiver repository to the SENDAI setting by integrating its core architecture into our training pipeline. Each sensor token contains its time-delay history concatenated with Fourier spatial positional encodings, thereby supplying analogous temporal and spatial information to that used by SENDAI's LSTM unit. As with DeepONet, we evaluate two configurations (Table 8): adapted *Senseiver*, trained on simulation and deployed directly on GT sensors, and *Senseiver-SENDAI*, which augments the Senseiver with SENDAI's multi-scale data assimilation pipeline.

The pure Senseiver achieves SSIM 0.408 on the held-out validation set—comparable to the adapted DeepONet (0.401) and the masked FNO on GT (0.345), and below SENDAI (0.489). Despite the ability of its attention mechanism to model long-range spatial dependencies, Senseiver shares the same core limitation as the other neural baselines: it largely reproduces the spatial structure of the simulation regime and does not transfer to the ground-truth seasonal regime without an explicit multi-scale discrepancy modeling mechanism. When augmented with SENDAI's DA pipeline (*Senseiver-SENDAI*), performance improves to SSIM 0.467, approaching SENDAI's SHRED-based result (0.489). Figure 11 illustrates the progression: the pure Senseiver produces smooth, simulation-like fields; Senseiver-SENDAI recovers the high-vegetation

*Table 7.* DeepONet baseline on the Tarim Basin site (mean $\pm$ std, 3 runs).

| Configuration | RMSE | SSIM |
|---|---|---|
| Adapted DeepONet | $0.170 \pm 0.003$ | $0.401 \pm 0.003$ |
| DeepONet-SENDAI | $0.132 \pm 0.002$ | $0.477 \pm 0.005$ |
| SENDAI (full pipeline) | $\mathbf{0.125} \pm 0.011$ | $\mathbf{0.489} \pm 0.021$ |

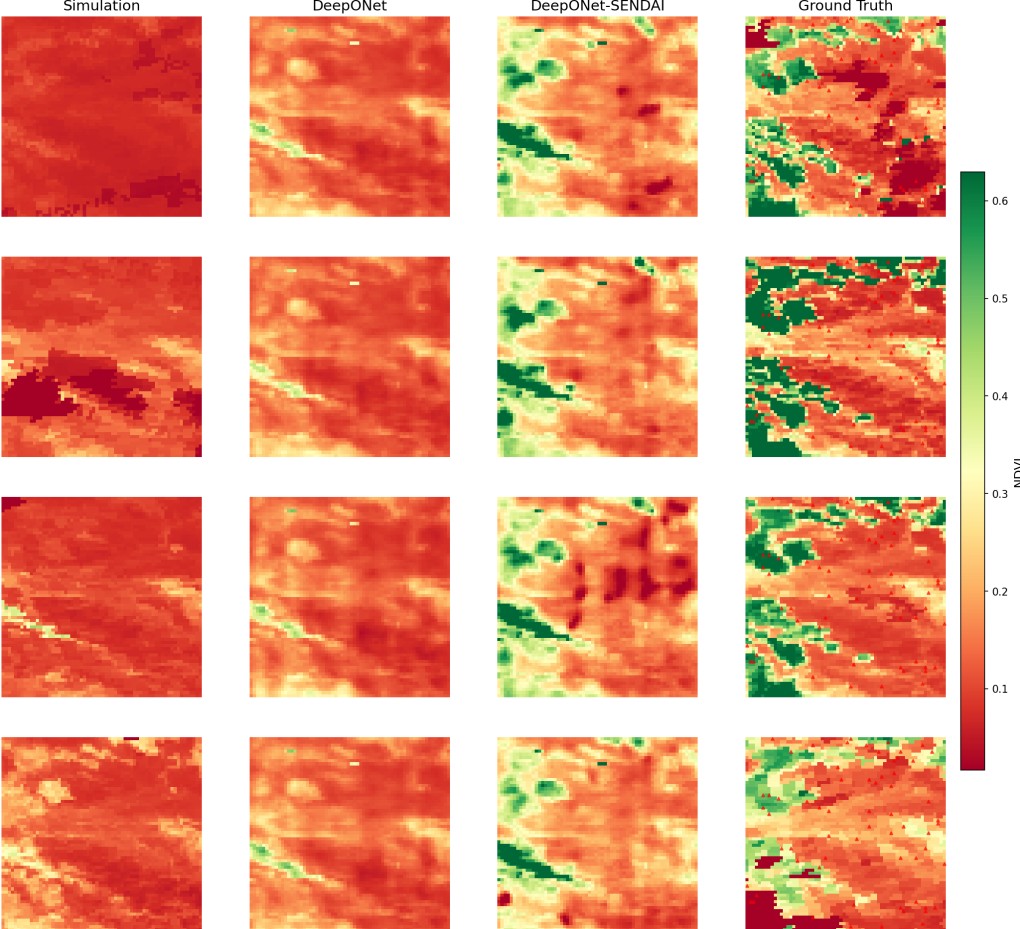

*Figure 10.* DeepONet reconstruction on the Tarim Basin site. Columns: Simulation, DeepONet (pure), DeepONet + SENDAI DA, Ground Truth.

patches present in the ground truth, though with some spatial artifacts compared to the SHRED-based SENDAI.

These results reinforce the conclusions from the neural operator experiments. Across the three distinct architectural paradigms for spatial reconstruction—FNO, DeepONet and Senseiver—none bridges the sim2real domain shift without SENDAI's multi-scale architecture. Conversely, both operator-based and attention-based architectures can be integrated into SENDAI's framework, with the current SHRED-based pipeline still providing consistent advantage. This confirms that SENDAI's contributions—latent-space alignment and hierarchical frequency peeling—are complementary to, rather than dependent on, the particular choice of spatial reconstruction unit. SENDAI provides key mechanisms that enable robust multi-scale reconstruction under domain shift from hyper-sparse sensor measurements.

*Table 8.* Senseiver baseline on the Tarim Basin site, held-out validation set (mean ± std, 3 runs).

| Configuration | RMSE | SSIM |
|---|---|---|
| Adapted Senseiver | 0.090 ± 0.001 | 0.408 ± 0.004 |
| Senseiver-SENDAI | 0.130 ± 0.005 | 0.467 ± 0.021 |
| SENDAI (full pipeline) | **0.125** ± 0.011 | **0.489** ± 0.021 |

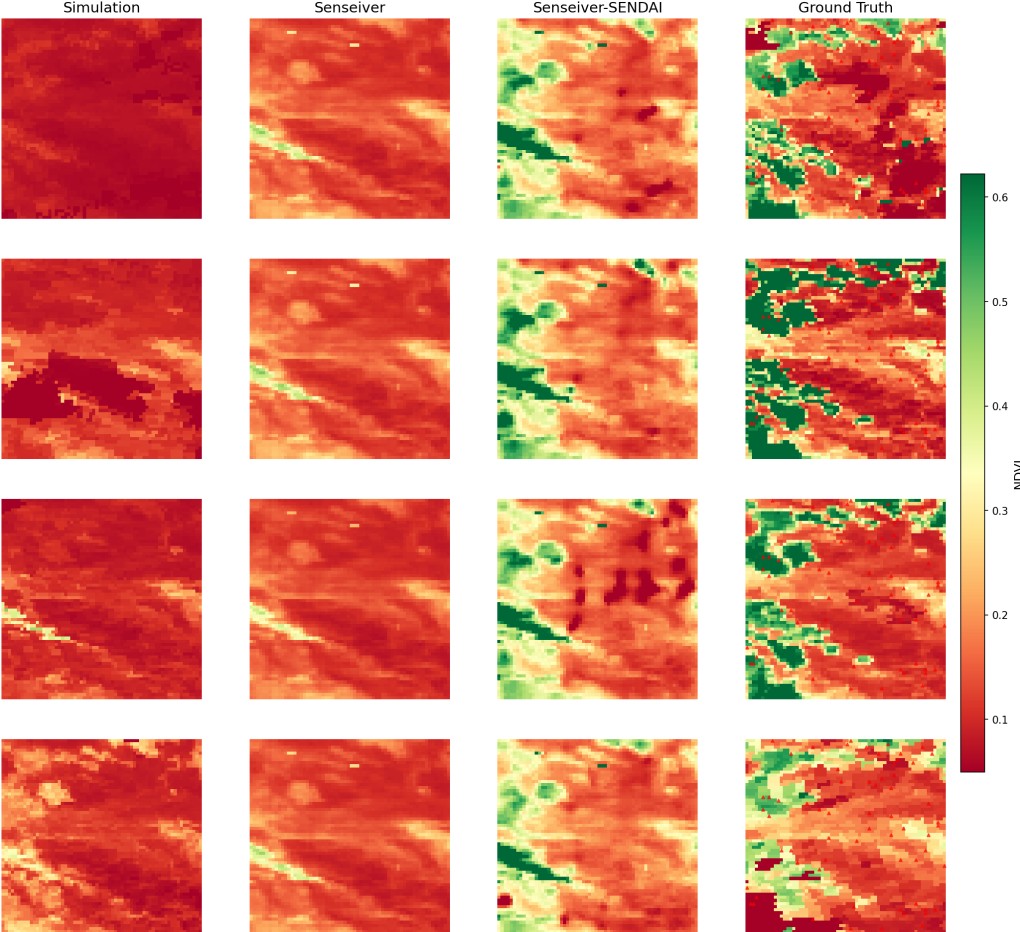

*Figure 11.* Senseiver reconstruction on the Tarim Basin site. Columns: Simulation, Senseiver (pure), Senseiver-SENDAI, Ground Truth.

*Table 9.* Computational cost on the Tarim Basin site. All methods run on CPU (Apple M4). Training time includes all stages; inference latency is per-sample.

| Method | Parameters | Training Time | Inference Latency |
|---|---|---|---|
| FNO (vanilla) | 2.1M | 2m 4s | 11.7 ms |
| FNO (masked) | 2.1M | 3m 10s | 11.8 ms |
| Adapted DeepONet | 365K | 4s | 0.2 ms |
| DeepONet-SENDAI | 699K | 33m 22s | 6.3 ms |
| Adapted Senseiver | 67.5M | 2m 1s | 12.1 ms |
| Senseiver-SENDAI | 67.5M | 46m 6s | 14.9 ms |
| **SENDAI** | 1.5M | 32m 12s | 12.0 ms |

## F. Implicit Neural Representations for Geospatial Data

The coordinate-based implicit neural representation (INR) employed in the high-frequency pathway represents an emerging paradigm for geospatial data with substantial unexplored potential (Sitzmann et al., 2020; Tancik et al., 2020). Unlike discrete gridded representations, INRs parameterize spatial fields as continuous functions, enabling queries at arbitrary coordinates and natural handling of irregular geometries.

Recent work has demonstrated INR applicability across Earth science domains: potential field geophysics (Smith et al., 2025), species distribution modeling (Cole et al., 2023), and climate data compression (Mostajeran et al., 2025). Our application to high-frequency correction in satellite imagery suggests additional utility: INRs can learn spatially coherent patterns from sparse sensor residuals, producing smooth interpolation without the localized artifacts characteristic of direct MLP regression.

The combination of INR decoders with recurrent encoders of sensor time-histories represents a hybrid architecture with broader applicability. The recurrent encoder captures temporal dynamics (phenological trends, seasonal patterns); the INR decoder produces spatially coherent instantaneous fields. This separation of temporal and spatial processing may prove advantageous across spatiotemporal reconstruction problems where temporal and spatial structure exhibit distinct characteristics.

## G. SENDAI Architecture Details

This appendix provides complete architectural specifications for the SENDAI components discussed in Section 3.

### G.1. Base SHRED Model

The base SHRED architecture consists of an LSTM temporal encoder and an MLP spatial decoder.

**LSTM Encoder.**   The encoder processes sensor time-histories $\mathbf{S}_{t-L:t} \in \mathbb{R}^{L \times p}$:

$$\mathbf{h}_\tau, \mathbf{c}_\tau = \text{LSTM}(\mathbf{s}_\tau, \mathbf{h}_{\tau-1}, \mathbf{c}_{\tau-1}), \quad \tau = t - L + 1, \ldots, t, \tag{24}$$

with $K$ stacked LSTM layers (typically $K = 2$), hidden dimension $d_z$, and dropout applied between layers during training. The latent representation is:

$$\mathbf{z} = \text{LayerNorm}(\mathbf{h}_t^{(K)}), \tag{25}$$

where $\mathbf{h}_t^{(K)}$ is the final hidden state of the top layer.

**MLP Decoder.**   The decoder maps latent codes to full spatial states:

$$\mathcal{D}_{\text{LF}}(\mathbf{z}) = \mathbf{W}_D \cdot \text{ReLU}(\text{LN}(\mathbf{W}_{D-1} \cdots \text{ReLU}(\text{LN}(\mathbf{W}_1 \mathbf{z} + \mathbf{b}_1)) \cdots)), \tag{26}$$

where LN denotes layer normalization and the hidden layer dimensions are specified in Table 10.

### G.2. DA-SHRED Latent Transform

The latent transformation module adapts simulation-trained representations to ground truth sensor data:

$$\mathcal{T}(\mathbf{z}) = \tanh\left(\mathbf{W}_2 \cdot \text{ReLU}(\mathbf{W}_1 \mathbf{z} + \mathbf{b}_1) + \mathbf{b}_2\right), \tag{27}$$

with the transformed latent given by Eq. (4). The $\tanh$ nonlinearity bounds the correction magnitude, and the learnable scale $\gamma$ is initialized to 0.1.

**GAN Discriminator.**   The discriminator $\mathcal{D} : \mathbb{R}^{d_z} \to [0, 1]$ is a 3-layer MLP with LeakyReLU activations (slope 0.2):

$$\mathcal{D}(\mathbf{z}) = \sigma\left(\mathbf{W}_3 \cdot \text{LReLU}(\mathbf{W}_2 \cdot \text{LReLU}(\mathbf{W}_1 \mathbf{z}))\right), \tag{28}$$

where $\sigma$ is the sigmoid function. Training uses the binary cross-entropy objectives:

$$\mathcal{L}_D = -\mathbb{E}_{\mathbf{z} \sim p_{\text{gt}}}[\log \mathcal{D}(\mathbf{z})] - \mathbb{E}_{\mathbf{z} \sim p_{\text{sim}}}[\log(1 - \mathcal{D}(\mathcal{G}(\mathbf{z})))], \tag{29}$$

$$\mathcal{L}_G = -\mathbb{E}_{\mathbf{z} \sim p_{\text{sim}}}[\log \mathcal{D}(\mathcal{G}(\mathbf{z}))]. \tag{30}$$

**SENDAI Jr. Pipeline.** For datasets where the sim2real gap is primarily low-frequency, a simplified two-stage pipeline suffices. Stage 1 trains the LSTM encoder and MLP decoder on simulation data with either sparse sensor histories or full-state supervision until convergence. Stage 2 then encodes both simulation and ground truth sensor data to obtain latent distributions and trains the GAN components (generator $\mathcal{G}$ and discriminator $\mathcal{D}$) on these latent codes to align the distributions. This variant relies solely on latent-space alignment to bridge the sim2real gap and is recommended as a baseline before deploying the full hierarchical architecture. The full pipeline should be used when spectral analysis of post-alignment residuals reveals significant high-frequency structure, evidenced by distinct peaks in the FFT magnitude spectrum of $\mathbf{s}'(t) - \mathbf{M}\tilde{\mathbf{u}}_{\mathrm{LF}}(t)$.

### G.3. Coordinate-Based INR for HF Correction

The implicit neural representation for high-frequency correction consists of an encoder and coordinate-based decoder.

**Sensor Residual Encoder.** Maps $p$-dimensional residuals to latent codes:

$$\mathcal{E}_{\mathrm{HF}}(\mathbf{r}) = \mathbf{W}_E^{(2)} \cdot \mathrm{ReLU}(\mathrm{LN}(\mathbf{W}_E^{(1)}\mathbf{r} + \mathbf{b}_E^{(1)})) + \mathbf{b}_E^{(2)}, \tag{31}$$

with output dimension $d_{\mathrm{HF}}$ (typically 64).

**Fourier Positional Encoding.** For coordinates $(x, y) \in [0, 1]^2$ and $L$ frequency bands with log-spaced frequencies $\sigma_j = 2^{(j-1)\log_2 \sigma_{\max}/(L-1)}$:

$$\mathrm{PE}(x, y) = [x, y, \sin(2\pi\sigma_1 x), \cos(2\pi\sigma_1 x), \sin(2\pi\sigma_1 y), \cos(2\pi\sigma_1 y), \ldots] \in \mathbb{R}^{2+4L}. \tag{32}$$

Typical values are $L = 16$ frequency bands with $\sigma_{\max} = 8.0$.

**Coordinate Decoder.** The decoder MLP takes the concatenation $[\mathrm{PE}(x, y); \mathbf{z}_{\mathrm{HF}}] \in \mathbb{R}^{2+4L+d_{\mathrm{HF}}}$:

$$\mathcal{D}_{\mathrm{INR}}([\mathrm{PE}; \mathbf{z}]) = \mathbf{W}_3 \cdot \mathrm{ReLU}(\mathrm{LN}(\mathbf{W}_2 \cdot \mathrm{ReLU}(\mathrm{LN}(\mathbf{W}_1[\mathrm{PE}; \mathbf{z}])))). \tag{33}$$

The final layer produces a scalar output for each queried coordinate.

**Batched Coordinate Queries.** At inference, all $n = H \times W$ grid coordinates are queried simultaneously. For memory efficiency with large grids, coordinates are processed in chunks:

$$u_{\mathrm{HF}}(i, j) = \gamma \cdot \mathcal{D}_{\mathrm{INR}}\left([\mathrm{PE}(i/H, j/W); \mathbf{z}_{\mathrm{HF}}]\right), \quad \forall (i, j) \in \{0, \ldots, H-1\} \times \{0, \ldots, W-1\}. \tag{34}$$

### G.4. 2D Frequency Sparsity Regularization

For 2D spatial fields, frequency sparsity is computed via the 2D real FFT.

**Frequency Grid.** Let $\hat{\mathbf{u}} = \mathrm{rfft2}(\mathbf{u})$ with shape $(H, W/2 + 1)$. The frequency coordinates are:

$$k_y \in \{0, 1, \ldots, H/2, -H/2 + 1, \ldots, -1\}, \tag{35}$$
$$k_x \in \{0, 1, \ldots, W/2\}. \tag{36}$$

The frequency radius is $\|\mathbf{k}\| = \sqrt{k_y^2 + k_x^2}$.

**Bandlimited Sparsity.** The in-band region $\mathcal{B} = \{\mathbf{k} : \|\mathbf{k}\| \leq k_{\max}\}$ and out-of-band $\bar{\mathcal{B}} = \{\mathbf{k} : \|\mathbf{k}\| > k_{\max}\}$:

$$\mathcal{R}_{\mathrm{band}}(\mathbf{u}) = \frac{\sum_{\mathbf{k} \in \mathcal{B}} |\hat{u}_{\mathbf{k}}|}{\sqrt{\sum_{\mathbf{k} \in \mathcal{B}} |\hat{u}_{\mathbf{k}}|^2 + \epsilon}} + \beta_1 \cdot \frac{\sum_{\mathbf{k} \in \bar{\mathcal{B}}} |\hat{u}_{\mathbf{k}}|^2}{\sum_{\mathbf{k}} |\hat{u}_{\mathbf{k}}|^2 + \epsilon}. \tag{37}$$

**Frequency Exclusion.** For layer $\ell$, let $\{(\bar{k}_y^{(j)}, \bar{k}_x^{(j)})\}_{j=1}^J$ be frequencies discovered by previous layers. The exclusion region with radius $r_{\text{exc}}$ is:

$$\mathcal{E}^{(\ell)} = \bigcup_{j=1}^J \left\{ \mathbf{k} : \sqrt{(k_y - \bar{k}_y^{(j)})^2 + (k_x - \bar{k}_x^{(j)})^2} < r_{\text{exc}} \right\}. \tag{38}$$

Due to conjugate symmetry of real signals, both $(k_y, k_x)$ and $(-k_y, k_x)$ are excluded. The exclusion penalty uses weight $\beta_2$:

$$\mathcal{P}_{\mathcal{E}}^{(\ell)} = \beta_2 \cdot \frac{\sum_{\mathbf{k} \in \mathcal{E}^{(\ell)}} |\hat{u}_{\mathbf{k}}|^2}{\sum_{\mathbf{k}} |\hat{u}_{\mathbf{k}}|^2 + \epsilon}. \tag{39}$$

After training each HF peeling layer, dominant frequencies are identified by computing the 2D FFT of the mean HF output over the training set, excluding the DC component, and extracting the top-$k_\ell$ unique frequency locations accounting for conjugate symmetry. These are then added to the exclusion set for subsequent layers.

**Top-$k_\ell$ Sparsity.** For peeling layer $\ell$ with adaptively selected $k_\ell$ modes:

$$\mathcal{R}_{\text{topk}}(\mathbf{u}; k_\ell) = 1 - \frac{\sum_{i=1}^{k_\ell} |\hat{u}_{(i)}|^2}{\sum_{\mathbf{k}} |\hat{u}_{\mathbf{k}}|^2 + \epsilon}, \tag{40}$$

where $|\hat{u}_{(i)}|$ is the $i$-th largest Fourier magnitude.

### G.5. Smoothness Regularization Options

Three smoothness regularization strategies are supported, with selection guided by prior knowledge of the expected HF field characteristics:

**Gradient (Total Variation).** For fields with sharp edges or discontinuities:

$$\mathcal{R}_{\text{grad}}(\mathbf{u}) = \frac{1}{HW} \sum_{i,j} \left[ (u_{i,j+1} - u_{i,j})^2 + (u_{i+1,j} - u_{i,j})^2 \right]. \tag{41}$$

This penalizes gradients uniformly, promoting piecewise constant solutions.

**Laplacian (Curvature).** For fields expected to be smooth with gentle variations:

$$\mathcal{R}_{\text{lap}}(\mathbf{u}) = \frac{1}{(H-2)(W-2)} \sum_{i,j} \left[ u_{i+1,j} + u_{i-1,j} + u_{i,j+1} + u_{i,j-1} - 4u_{i,j} \right]^2. \tag{42}$$

This penalizes curvature (second derivatives) rather than gradients, allowing linear ramps and sharp but smooth features while suppressing high-frequency oscillations.

**Bilateral (Edge-Preserving).** For fields with both smooth regions and sharp boundaries:

$$\mathcal{R}_{\text{bilateral}}(\mathbf{u}) = \sum_{i,j} \left[ \text{Huber}_\delta(u_{i,j+1} - u_{i,j}) + \text{Huber}_\delta(u_{i+1,j} - u_{i,j}) \right], \tag{43}$$

where $\text{Huber}_\delta(x) = \frac{1}{2}x^2$ if $|x| < \delta$, else $\delta(|x| - \frac{\delta}{2})$. The threshold $\delta$ controls the transition between quadratic (small gradients) and linear (large gradients) penalization, preserving edges while smoothing homogeneous regions.

### G.6. Hyperparameter Configuration

Table 10 summarizes the default hyperparameters.

### G.7. Computational Cost and Model Complexity

Table 11 summarizes the computational cost and model complexity.

*Table 10.* Default hyperparameters for the SENDAI architecture.

| Component | Parameter | Value |
|---|---|---|
| LSTM Encoder | Hidden dimension $d_z$ | 32 |
| | Number of layers $K$ | 2 |
| | Dropout rate | 0.1 |
| | Temporal lags $L$ | 5 |
| LF Decoder | Hidden layers | [256, 256] |
| | Activation | ReLU |
| GAN | Generator hidden | 64 |
| | Discriminator hidden | 64 |
| INR (per HF layer) | Latent dimension $d_{\text{HF}}$ | 64 |
| | Encoder hidden | [128, 128] |
| | Decoder hidden | [256, 256, 128] |
| | PE frequencies $L$ | 16 |
| | PE max frequency $\sigma_{\text{max}}$ | 8.0 |
| | Scale $\gamma$ init | 0.1 |
| HF Training | Warmup epochs $E_{\text{warm}}$ | 100 |
| | $\lambda_{\text{sp}}$ | 0.05 |
| | $\lambda_{\text{sm}}$ | 0.1 |
| | Fine-tune $\lambda'_{\text{sp}}$ | 0.005 |
| Sparsity Penalties | Out-of-band $\beta_1$ | 100 |
| | Exclusion $\beta_2$ | 100 |
| | Top-k weight $\lambda_{\text{topk}}$ | 10.0 |
| | Exclusion radius $r_{\text{exc}}$ | 2.0 |
| Adaptive $k_\ell$ | Bandwidth tolerance $\Delta k$ | 2.0 |
| | Energy threshold $\rho$ | 0.8 |
| Optimization | Learning rate | $10^{-4}$ |
| | Batch size | 16 |
| | Optimizer | AdamW |

*Table 11.* Computational cost and model complexity comparison.

| SENDAI Jr. | | | SENDAI | | |
|---|---|---|---|---|---|
| *Model Complexity* | | | *Model Complexity* | | |
| SHRED (LSTM + Decoder) | 1,149.0 K | | SHRED (LSTM + Decoder) | | 1,149.0 K |
| DA Transform | 4.2 K | | DA Transform | | 4.2 K |
| | | | HF Peeling Layers | | 334.5 K |
| **Total parameters** | **1,153.2 K** | | **Total parameters** | | **1,487.7 K** |
| *Training Time* | | | *Training Time* | | |
| Stage 1 (SHRED) | 2.76 sec | | Stage 1 (SHRED) | | 2.85 sec |
| Stage 2 (DA-SHRED + GAN) | 10.26 sec | | Stage 2 (DA-SHRED + GAN) | | 14.18 sec |
| | | | Stage 3 (Hierarchical HFP + INR) | | 25 min 50 sec |
| **Total** | **15.94 sec** | | **Total** | | **26 min 12 sec** |
| | | *Hardware* | | | |
| | | CPU: Apple M4, Memory: 24 GB | | | |

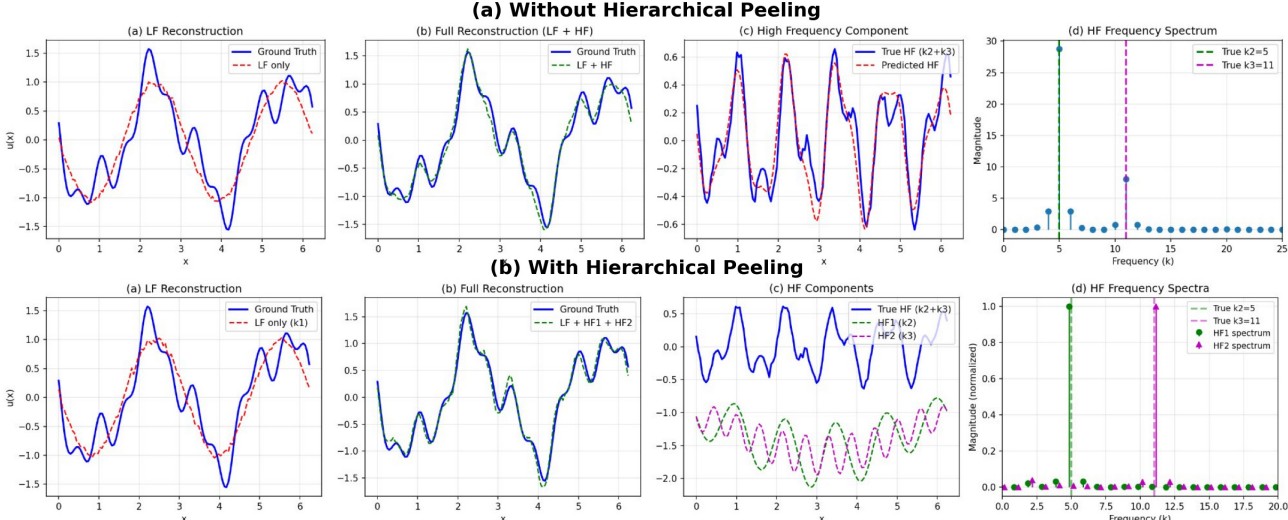

*Figure 12.* Comparison of joint and hierarchical frequency discovery on the three-mode traveling wave system for a single time-point reconstruction. **(a)** the frequency spectrum shows energy leakage to non-target modes. **(b)** modes are discovered sequentially, yielding spectrally clean outputs and improved fine-scale fidelity. In third panel, HF1 and HF2 show the individually learned components (unnormalized); their sum after scaling recovers the true HF.

## H. Synthetic Validation: Extended Details

This appendix provides comprehensive details for the synthetic validation experiments presented in Section 4.1.

### H.1. Traveling Wave System: Data Generation

The synthetic traveling wave system is generated on a spatial domain $x \in [0, 2\pi]$ with $N = 128$ grid points and temporal domain $t \in [0, 10]$ with $\Delta t = 0.05$ (200 timesteps). The full three-mode field is:

$$u(x, t) = \sin(2x - t) + 0.4\sin(5x - 3t) + 0.25\sin(11x - 7t), \tag{44}$$

where the different temporal frequencies $\omega_1 = 1$, $\omega_2 = 3$, $\omega_3 = 7$ ensure the three modes remain distinguishable as the system evolves. The simulation model contains only the first term, representing a simplified physics model that misses intermediate and fine-scale dynamics.

### H.2. NDVI Slice Extraction

For the NDVI slice experiment, we extract a 1D transect from the Tarim Basin site at $x = 25\%$ of the image width (column 16 of a $64 \times 64$ grid). This slice traverses a sharp mountain-basin boundary, capturing the heterogeneous vegetation structure characteristic of the landscape.

Both simulation data from the April–June period and ground truth from July–October are aligned to $T = 72$ timesteps for training. The HF residual exhibits:

- Range: $[-0.46, 0.86]$ NDVI units

- Standard deviation: 0.212

- Sharp discontinuities at mountain boundaries (pixels 10–20, 45–55)

- Temporal variability from phenology

### H.3. Sensor Configuration

Both experiments employ an extremely sparse configuration of only $p = 3$ sensors. For the traveling wave system, this provides approximately 2.3% spatial coverage of the 128-point grid. For the NDVI slice, sensors are randomly placed as well.

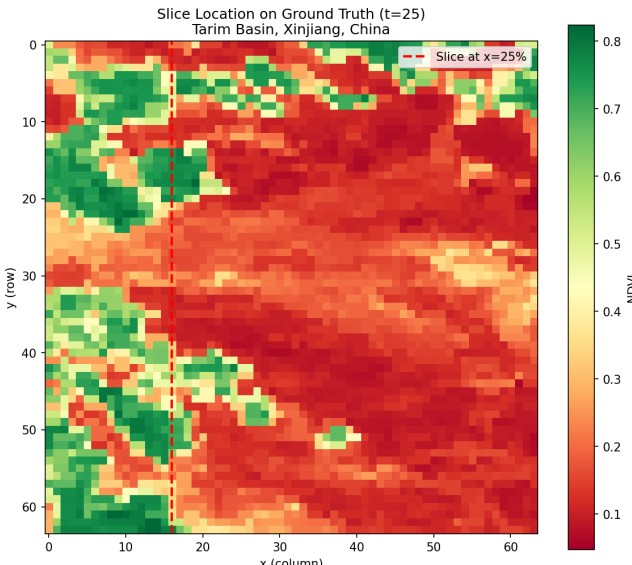

*Figure 13.* Slice location for hierarchical frequency peeling on a 1D NDVI slice from the Tarim Basin.

Despite this severe undersampling, both experiments successfully recover the missing frequency content, demonstrating the framework's ability to exploit temporal coherence for frequency recovery.

The LSTM encoder processes temporal histories of length $L = 20$ lags (traveling wave) or $L = 10$ lags (NDVI slice), exploiting temporal coherence to compensate for spatial sparsity.

For the NDVI experiment, INR decoders with Fourier positional encoding are used to produce spatially coherent outputs despite the sharp discontinuities in the target field.

### H.4. Joint vs. Hierarchical Frequency Discovery

We compare two strategies for HF correction on the traveling wave system:

**Joint Discovery.** A single HF pathway with bandlimited sparsity discovers both $k_2$ and $k_3$ simultaneously. While achieving 84.7% RMSE improvement, the frequency spectrum shows energy spread across modes $k \in \{3, 4, 5, 10, 11, 12\}$ rather than concentrated solely at the targets. The combined HF output represents an entangled mixture with energy leakage.

**Sequential Peeling.** The hierarchical approach achieves 85.1% RMSE improvement with clean spectrum: $HF_1$ captures $k = 5$ with $> 95\%$ of its energy at the target mode, and $HF_2$ captures $k = 11$ similarly. This spectral purity enables physical interpretation of individual frequency contributions, downstream analysis of specific spectral components, and modular addition of peeling layers without retraining.

### H.5. NDVI Transect Decomposition

The 1D NDVI slice from the Tarim Basin site illustrates how hierarchical frequency peeling decomposes a complex, noisy simulation–observation residual into interpretable components across scales.

**$HF_1$: Climate-Driven Seasonal Dynamics.** The first peeling layer captures the dominant residual variability, combining coherent temporal fluctuations with a broad spatial gradient along the transect. Its synchronized, episode-like oscillations are consistent with basin-wide meteorological forcing (e.g., temperature anomalies and intermittent precipitation events) acting across the slice, while the gradual late-season attenuation suggests a phenological convergence between spring-calibrated simulations and summer–autumn observations. The spatial amplitude structure further indicates modulation by the elevation gradient, supporting an interpretation of temperature-mediated phenological offsets that vary systematically across landscape zones.

**HF$_2$: Hydrological Persistence.**    The second peeling layer is characterized by stronger, more persistent spatial stratification with relatively minimal temporal evolution, indicating a quasi-stationary control on the residual. In the hyperarid Tarim Basin, such stable structure could be plausibly explained by differential water availability governed by proximity to snow-melt, groundwater access, and topographic accumulation of runoff.

**HF$_3$: Edaphic and Microsite Heterogeneity.**    The third peeling layer isolates finer-scale, temporally invariant spatial heterogeneity, pointing to localized controls that persist over the season. We attribute this component to edaphic and microsite factors such as soil texture, salinity, nutrient level, and geomorphic microhabitats (e.g., abandoned channels or alluvial features) that influence vegetation independently of broad climate and hydrological gradients. Predominantly negative anomalies are consistent with systematic overestimation by the low-frequency reconstruction in sub-pixel heterogeneous patches that cannot be resolved without an explicit multi-scale residual model.

### H.6. Advantages of Hierarchical Peeling

The experiments reveal several advantages of hierarchical peeling:

**Robustness to Non-Ideal Signals.**    The NDVI slice experiment demonstrates that hierarchical peeling succeeds even when the target signal doesn't have clean sinusoidal patterns. The landscape creates sharp discontinuities (effectively high bandwidth), yet the peeling layers correctly identify the majority of spectral modes without being corrupted by edge effects.

**Frequency Exclusion Mechanism.**    The exclusion penalty prevents mode leakage where a subsequent layer partially recaptures previously discovered content. In the NDVI experiment, HF$_2$ correctly discovers $k = 4$ rather than reinforcing the $k = 2$ mode already captured by HF$_1$, despite $k = 2$ having substantial residual energy.

**Stable Training Dynamics.**    By constraining each layer to discover a concentration of modes, the training procedure becomes more robust. The traveling wave experiment shows HF$_1$ converging to $k = 5$ before epoch 200, while HF$_2$ locks onto $k = 11$ shortly after. The NDVI experiment exhibits similar sequential convergence despite the noisy target signal.

**Interpretability.**    The adaptive HF selection encourages each layer to focus on the most salient remaining structure, while the exclusion mechanism reduces redundancy across layers, yielding a nested decomposition that maps naturally onto scale-dependent control mechanisms.

### H.7. Implications for Full 2D Reconstruction

The synthetic validation confirms two key capabilities essential for the full SENDAI framework:

**Extreme Sparsity Tolerance.**    Both experiments succeed with only $p = 3$ sensors ($\sim$2–5% coverage), demonstrating the framework's ability to exploit temporal coherence for frequency recovery. This directly supports the NDVI reconstruction task where 64 sensors cover only 1.56% of the spatial domain.

**Universality Across Signal Types.**    The success on both clean synthetic waves and noisy real NDVI data—with outliers, sharp discontinuities, and non-stationary dynamics—demonstrates that hierarchical peeling is not limited to idealized signals. The frequency exclusion mechanism and INR decoders together handle the heterogeneous spatiotemporal fields characteristic of real remote sensing applications.

These findings motivate the hierarchical architecture employed in the full SENDAI framework, where multiple peeling layers with coordinate-based INR decoders sequentially extract interpretable spectral corrections from the sim2real discrepancy.

## I. SENDAI Jr. Site-Specific Results

This appendix provides detailed analyses and qualitative reconstruction results for the three sites evaluated using the SENDAI Jr. pipeline. Performance is assessed using both RMSE and the Structural Similarity Index Measure (SSIM), with SSIM serving as the primary indicator of reconstruction quality. As explained in the main text, baseline methods fail to preserve the topological structure of spatial patterns despite achieving moderate RMSE values. IDW-based methods exhibit "bullseye" artifacts centered at sensor locations. Kriging produces overly smooth reconstructions that obscure sharp boundaries. SSIM

captures the preservation of spatial patterns, textures, and structural information that RMSE alone cannot assess—critically important for remote sensing applications where topological fidelity determines downstream analysis utility.

## I.1. Central Valley, California, USA

This irrigated cropland site achieves RMSE of 0.1068 and SSIM of 0.5747, representing a 120% SSIM improvement over the best-performing baseline (SG+IDW at 0.2612). This substantial structural improvement indicates that SENDAI Jr. successfully preserves field boundaries, irrigation patterns, and vegetation gradients that baseline methods systematically destroy.

Figure 25 shows baseline performance on the Central Valley site. While this site exhibits less extreme heterogeneity than the Tarim Basin, the baseline methods still produce characteristic artifacts: IDW-based methods show bullseye patterns around sensor locations (SSIM: 0.2612 and 0.2504), while Kriging over-smooths field boundaries resulting in the lowest SSIM (0.0922). The poor SSIM values of baseline methods—despite moderate RMSE—demonstrate the critical importance of structural similarity metrics for evaluating reconstruction quality.

Figure 26 presents qualitative reconstruction results across four equally-spaced temporal frames within the ground truth observation period. The reconstructed fields preserve the essential spatial heterogeneity of the ground truth, including field boundaries and vegetation gradients, despite access to only 64 point measurements per frame. The high SSIM value (0.5747) quantitatively confirms this visual assessment of structural preservation.

## I.2. Corn Belt, Iowa, USA

Despite the pronounced phenological transition from vegetative growth (April–June) to reproductive and senescence stages (July–October), SENDAI Jr. achieves RMSE of 0.1103 and SSIM of 0.4530 at this site. The SSIM improvement is particularly striking: a 185% increase over the best baseline (SG+IDW at 0.1588). This dramatic improvement reflects SENDAI Jr.'s ability to maintain spatial coherence across the severe phenological domain shift characteristic of temperate agricultural systems.

Figure 27 presents baseline performance on the Iowa site. The rainfed cropland exhibits more homogeneous vegetation patterns than irrigated sites, yet baseline methods still fail to capture the field-scale structure characteristic of corn-soybean rotations. Kriging achieves the lowest SSIM (0.0312), producing nearly featureless smooth fields that completely eliminate the spatial patterns present in the ground truth. This extreme structural degradation occurs despite Kriging achieving the best baseline RMSE (0.1596), highlighting the inadequacy of RMSE as a sole evaluation metric.

Figure 28 illustrates the SENDAI Jr. reconstruction, where the framework successfully adapts from spring emergence conditions to mid-season reproductive stages despite the substantial phenological shift. The reconstruction captures the characteristic high-NDVI patterns of mature corn and soybean crops, preserving the field-scale spatial structure that enables crop type discrimination and yield estimation.

## I.3. Guadalquivir Valley, Spain

This Mediterranean site, characterized by reversed phenological timing relative to temperate Northern Hemisphere regions, achieves RMSE of 0.1474 and SSIM of 0.3655. The SSIM represents a 98% improvement over the best baseline (SG+IDW at 0.1849). The framework successfully adapts from winter-spring simulation conditions (February–April) to autumn ground truth observations (September–December), demonstrating robustness to non-standard seasonal calendars.

Figure 29 shows baseline performance on this site. The mixture of irrigated agriculture and natural vegetation creates heterogeneous patterns that baseline methods fail to faithfully reproduce. All baseline methods achieve SSIM below 0.19, indicating severe structural degradation. Kriging again shows the largest disconnect between RMSE and SSIM, achieving moderate RMSE (0.1481) but poor SSIM (0.0878).

Figure 30 demonstrates SENDAI Jr. reconstruction for the Guadalquivir Valley, where the reversed Mediterranean phenological calendar requires adaptation from winter-spring greenness to autumn ground truth conditions. The model recovers coherent spatial patterns including agricultural field boundaries and regional vegetation gradients, as reflected in the substantially improved SSIM.

### I.4. Summary of SENDAI Jr. Performance

Across all three sites, SENDAI Jr. demonstrates consistent superiority over baseline methods in both RMSE and SSIM metrics, with particularly pronounced advantages in structural preservation:

- **Central Valley**: 120% SSIM improvement, preserving irrigated field boundaries

- **Iowa Corn Belt**: 185% SSIM improvement, maintaining field-scale crop patterns

- **Guadalquivir Valley**: 98% SSIM improvement, capturing Mediterranean agricultural structure

The large gap between baseline RMSE and SSIM performance—particularly for Kriging—demonstrates that traditional error metrics inadequately capture reconstruction quality for heterogeneous landscapes. SENDAI Jr.'s success in improving both metrics simultaneously indicates that the framework learns physically meaningful spatial priors rather than simply minimizing point-wise error.

## J. SENDAI Site-Specific Results

This appendix provides detailed analyses and qualitative reconstruction results for the three sites evaluated using the full SENDAI hierarchical multiscale DA-SHRED architecture with INR. These sites exhibit complex phenological dynamics, sub-seasonal variability, or pronounced spatial heterogeneity that require high-frequency correction beyond what SENDAI Jr. can provide. SSIM serves as our primary performance indicator, capturing the preservation of sharp boundaries, fine-scale features, and topological structure.

### J.1. Imperial Valley, California, USA

This dry and hot irrigated agriculture site exhibits consistent improvement through the hierarchical pipeline. SENDAI Jr. achieves SSIM of 0.4041, which increases to 0.4411 with HF peeling (+9.2%) and further to 0.4668 with the full SENDAI pipeline (+15.5% total improvement from SENDAI Jr.). RMSE improves correspondingly from 0.1708 to 0.1486.

The baseline methods achieve uniformly poor SSIM values: SG+IDW (0.1123), HANTS+IDW (0.1049), and Kriging (0.0916). This represents a $4\times$ to $5\times$ improvement in structural similarity for the full SENDAI pipeline over baselines, demonstrating the framework's ability to preserve the rectilinear irrigation infrastructure that defines this landscape.

Figure 31 presents baseline reconstruction results for Imperial Valley. The contrast between irrigated fields and surrounding desert creates sharp NDVI boundaries that are lost in baseline reconstructions, manifesting as circular bullseye artifacts (IDW methods) or smooth gradients (Kriging). The low SSIM values quantify this structural failure.

Figure 32 shows the full hierarchical reconstruction pipeline. The HF pathway captures fine-scale field boundaries characteristic of the rectilinear irrigation infrastructure, with the learned HF component exhibiting coherent spatial structure aligned with field edges rather than random noise.

### J.2. Tarim Basin, China

This continental site presents the most pronounced contrast between SENDAI Jr. and Sendai performance. The SENDAI Jr. achieves SSIM of 0.3505, which increases to 0.4257 with HF peeling (+21.5%) and to 0.4777 with the full pipeline (+36.3% total improvement). This is the largest SSIM improvement across all six sites, reflecting the critical importance of hierarchical high-frequency correction for landscapes with sharp boundaries.

The hierarchical pipeline also achieves substantial RMSE improvement: from 0.1827 (SENDAI Jr.) to 0.1208 (SENDAI), a 33.9% reduction. This demonstrates that the high-frequency corrections are not merely cosmetic but represent genuine improvements in reconstruction accuracy.

Baseline methods fail catastrophically on this site in terms of structural preservation: Kriging achieves the worst SSIM (0.0449), producing nearly featureless smooth fields despite moderate RMSE. The sharp mountain-basin boundaries that define this landscape are completely unresolved by all baseline approaches.

Figure 33 illustrates the failure of baseline methods on this challenging site. The oasis-desert transition creates extreme spatial gradients that interpolation-based methods cannot capture. IDW methods produce pronounced bullseye artifacts,

while Kriging eliminates all boundary information.

Figure 34 presents the full hierarchical reconstruction pipeline. The HF component captures the sharp oasis-desert boundaries that the smooth LF decoder fails to resolve. The learned corrections exhibit spatially coherent structure aligned with the landscape's topological features, confirming that the peeling layers discover physically meaningful high-frequency content rather than noise.

### J.3. Riverina, Australia

This mixed cropping region exhibits SSIM improvement from 0.2761 (SENDAI Jr.) to 0.3158 (SENDAI Jr.+HFP, +14.4%) and 0.3354 (SENDAI, +21.5%). While the absolute SSIM values are lower than the other sites, this reflects the inherently more diffuse spatial structure of this landscape rather than reconstruction failure.

Baseline methods again show poor structural preservation: Kriging achieves SSIM of only 0.0272, the lowest across all six sites. Even SG+IDW and HANTS+IDW achieve only 0.1359 and 0.1245 respectively. The full SENDAI pipeline achieves $2.5\times$ higher SSIM than the best baseline.

Figure 35 shows baseline performance on the Riverina site. While the spatial heterogeneity is less extreme than at other sites, baseline methods still fail to capture the field-scale structure of this mixed cropping landscape.

Figure 36 presents the hierarchical reconstruction. The HF component shows more diffuse structure consistent with the gradual spatial transitions in this mixed cropping region, reflecting the site's inherent landscape characteristics. The Southern Hemisphere location provides a test of generalization to reversed seasonal timing.

### J.4. Summary of SENDAI Performance

The hierarchical SENDAI framework demonstrates consistent improvements over both baselines and SENDAI Jr. across all three challenging sites:

- **Imperial Valley**: 15.5% SSIM improvement from SENDAI Jr., $4.2\times$ improvement over best baseline

- **Tarim Basin**: 36.3% SSIM improvement from SENDAI Jr., $3.7\times$ improvement over best baseline

- **Riverina**: 21.5% SSIM improvement from SENDAI Jr., $2.5\times$ improvement over best baseline

## K. Ablation and Robustness Analysis

### K.1. Sensor Number Sensitivity Analysis

We conducted a sensitivity analysis to evaluate the effect of sensor density on reconstruction quality. Experiments were performed with sensor counts ranging from 8 to 256, with five independent trials per configuration.

Figure 6a shows SSIM performance as a function of sensor count. Both the full dataset and validation set exhibit a clear positive trend: reconstruction quality improves with increasing sensor density. For our main experiments, we selected 64 sensors as a conservative configuration. At this density, the model achieves mean SSIM values of approximately 0.53 (full) and 0.49 (validation), representing a reasonable trade-off between reconstruction accuracy and practical deployment constraints. While higher sensor counts yield improved performance, the marginal gains must be weighed against increased instrumentation costs and maintenance requirements in field applications.

### K.2. Temporal lag

Figure 14 sweeps $L \in \{2, 5, 10, 20\}$. SSIM improves monotonically with increasing lag, consistent with Takens' embedding theorem: longer sensor histories provide a more faithful embedding of the underlying dynamics. RMSE shows a similar improving trend up to $L=10$. We adopt $L=5$ as the default throughout the paper as a conservative choice.

### K.3. Maximum target frequency

Figure 15 sweeps $k_{\max} \in \{4, 8, 12, 16, 20, 24\}$. Both RMSE and SSIM remain relatively stable across the range, with slight degradation at the highest $k_{\max}$ as the expanded band admits modes that the sparse sensor coverage cannot reliably

Ablation: Effect of Temporal Lag on SENDAI (Tarim Basin)

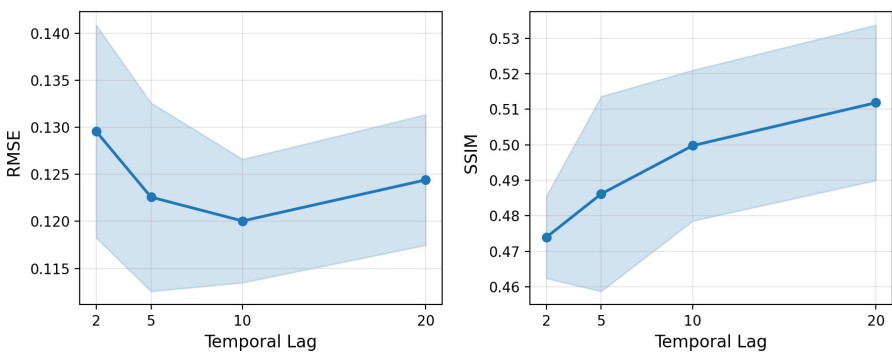

Figure 14. Effect of temporal lag $L$ on full SENDAI reconstruction (Tarim Basin). Shaded regions represent $\pm 1$ std.

Ablation: Effect of HF Max Target Frequency on SENDAI (Tarim Basin)

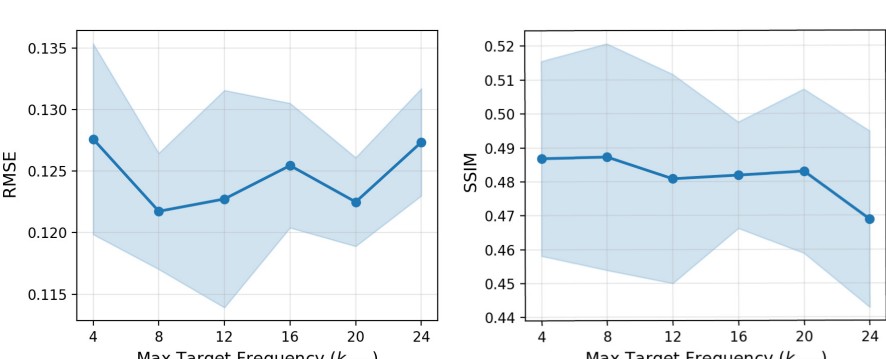

Figure 15. Effect of maximum target frequency $k_{\max}$ on full SENDAI reconstruction (Tarim Basin). Shaded regions represent $\pm 1$ std.

constrain. The overall insensitivity implies that SENDAI's spectral sparsity regularization and frequency exclusion zones are preventing the model from overfitting.

### K.4. Sensor noise robustness

Figure 16 trains on clean data and evaluates on sensors corrupted with additive Gaussian noise $\sigma \in \{0, 0.02, 0.05, 0.10, 0.15\}$ in raw NDVI units, thereby emulating deployment under degraded sensor quality. Both RMSE and SSIM remain stable across the noise levels tested. The robustness is expected given that the MODIS data already contains inherent sensor noise and atmospheric contamination, and the SHRED temporal unit naturally attenuates uncorrelated noise over the lag window. These results confirm that the hierarchical corrections are not artifacts of overfitting to sensor noises.

### K.5. Sensor Placement Robustness

To address the concern of performance degradation when sensor coverage is not random but becomes spatially skewed—as would occur under realistic settings—we present a controlled experiment on the Tarim Basin site, and additionally note that the seismic experiment in Section M.2 provides a complementary, real-world example of severely non-uniform sensor placement.

**Experimental design.** Cloud cover is spatially correlated and tends to preferentially obscure regions with dense vegetation, where higher evapotranspiration drives convective cloud formation. In the Tarim Basin, this would leave the mountain-basin boundary with reduced sensor coverage while the arid desert floor retains more observations. To simulate this scenario, we compare two sensor placement strategies, each using 64 sensors ($\sim 1.5\%$ coverage), averaged over three independent runs:

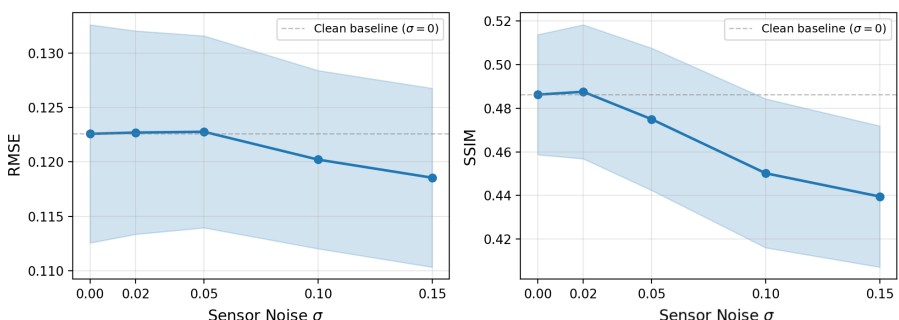

*Figure 16.* Sensor noise robustness of SENDAI (Tarim Basin). The model is trained on clean data and evaluated with Gaussian noise of varying $\sigma$ added to sensor observations. Dashed line indicates the clean baseline. Shaded regions indicate $\pm 1$ standard deviation over three independent runs.

*Table 12.* Sensor placement robustness on the Tarim Basin site (mean $\pm$ std, 3 runs). All experiments use 64 sensors ($\sim 1.5\%$ coverage).

| Placement Strategy | RMSE | SSIM |
|---|---|---|
| Uniform coverage | $0.125 \pm 0.007$ | $0.528 \pm 0.022$ |
| Density-biased clustered | $0.129 \pm 0.002$ | $0.536 \pm 0.008$ |

- **Uniform coverage**: sensors distributed uniformly across the full $64 \times 64$ domain.
- **Density-biased clustered**: 70% of sensors (45 of 64) concentrated in the bottom-right quadrant (25% of the domain, corresponding to the low-NDVI desert region), with the remaining 30% (19 sensors) scattered across the other 75% of the domain. This simulates a large-scale cloud system that preferentially obscures the vegetated mountain region while allowing sparse observations to persist in clear-sky gaps.

**Results.** Table 12 reports reconstruction metrics for both placement strategies. Under density-biased clustering, SENDAI achieves SSIM of $0.536 \pm 0.008$, compared with $0.528 \pm 0.022$ under uniform coverage. RMSE similarly remains stable ($0.129 \pm 0.002$ vs. $0.125 \pm 0.007$). The negligible difference indicates that SENDAI's reconstruction quality is robust to spatially non-uniform sensor coverage. This robustness derives from two architectural properties: (i) the SHRED temporal unit operates on sensor *histories* rather than instantaneous spatial snapshots, allowing temporal coherence to compensate for spatial gaps; and (ii) the coordinate-based INR decoder learns a continuous spatial function conditioned on a global latent, enabling smooth interpolation into unobserved regions rather than relying on local sensor proximity.

Figure 17 presents the sensor placement maps for the two strategies, and Figure 18 presents qualitative reconstruction results at two representative timesteps. Both placement strategies produce structurally coherent fields that preserve the mountain-basin boundary and mesoscale NDVI gradients. Under density-biased placement, the reconstruction in the sparsely observed region (top-left, mountain boundary) exhibits marginally reduced sharpness at fine scales, but the overall topological structure remains intact.

**Connection to the seismic experiment.** While the above experiment uses a controlled non-uniform placement on the NDVI dataset, the seismic waveform experiment in Section M.2 provides a stronger, naturally occurring example: the global seismographic network for event `usc000kn4n` concentrates $\sim 300$ of 686 stations in the United States, with the remaining stations distributed sparsely and irregularly across the globe (Figure 21). SENDAI reconstructs waveforms at held-out stations—including those in sparsely instrumented regions—without any modification to accommodate the non-uniform coverage. Together, these two experiments demonstrate that SENDAI is robust to sensor placement non-uniformity across both controlled and naturally occurring settings.

## L. Hardware Efficiency: Extended Discussion

This appendix provides detailed operational scenarios for the hardware efficiency paradigm introduced in Section 4.5.

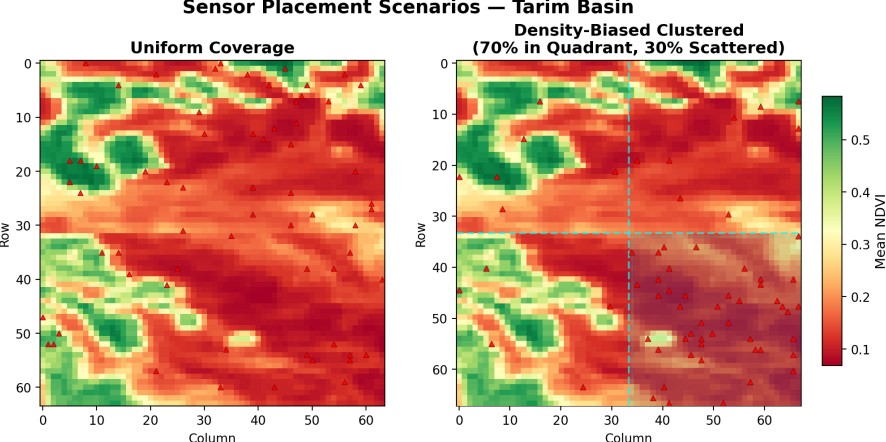

*Figure 17.* Sensor placement maps for the two strategies, overlaid on the mean NDVI field of the Tarim Basin ground-truth period.

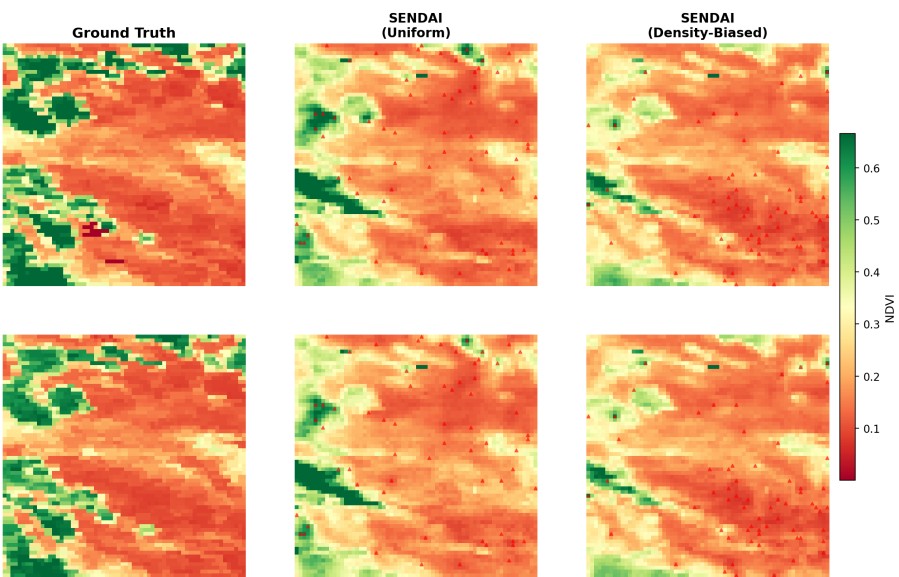

*Figure 18.* Reconstruction comparison under uniform coverage vs. density-biased sensor placement on the Tarim Basin site. Red markers indicate sensor locations. Under density-biased placement, 70% of sensors cluster in the bottom-right quadrant (desert), with 30% scattered elsewhere.

The demonstrated capacity to reconstruct full fields from 64 sensors (1.56% of pixels) suggests alternative paradigms for satellite data systems. Rather than transmitting complete imagery, systems could transmit sparse measurements alongside periodically updated model weights, achieving substantial data reduction while preserving spatial structure essential for downstream analysis. This paradigm addresses bandwidth constraints in resource-limited regions (De Cola et al., 2011), onboard storage limitations, and low-latency decision support requirements where provisional reconstructions from partial observations enable time-critical responses (Denby & Lucia, 2020; Giuffrida et al., 2020).

### L.1. Bandwidth Reduction Scenario

Consider the operational scenario: a satellite acquires a $64 \times 64$ pixel scene (4,096 values). Under our sparse sensing protocol, only 64 sensor measurements need be transmitted—a $64\times$ reduction before any conventional compression. If model weights are updated infrequently (e.g., seasonally), the marginal transmission cost per scene becomes negligible.

While reconstruction fidelity will never match lossless transmission of complete imagery, for applications where the reconstructed fields serve as inputs to downstream models (crop monitoring, anomaly detection, change analysis), the accuracy-bandwidth tradeoff may prove favorable.

## L.2. Deep-Space Exploration Applications

This paradigm holds particular significance for deep-space exploration missions, where communication bandwidth constrains scientific return (De Cola et al., 2011; Xie et al., 2021). Planetary surface monitoring—whether for Mars rover operations, lunar resource mapping, or asteroid characterization—faces extreme bandwidth limitations. A SENDAI-like architecture, pretrained on simulation data or initial mission observations, could enable substantial compression of subsequent observations while preserving the spatial structure essential for scientific interpretation.

## L.3. Low-Latency Decision Support Scenario

Consider a scenario where only a subset of spectral bands has completed atmospheric correction, or where preliminary telemetry provides sensor-location values before full-scene processing completes. The framework can generate an immediate reconstruction that, while provisional, may suffice for time-critical decisions. Full-fidelity imagery, when available, supersedes the provisional estimate.

This capability aligns with emerging architectures for on-board satellite processing (Denby & Lucia, 2020; Giuffrida et al., 2020), where computational constraints preclude comprehensive analysis but rapid triage decisions (e.g., assessing scenes to be prioritized for downlink) could leverage lightweight reconstruction from sparse measurements.

# M. Multi-Domain Generalizability

## M.1. Remote Sensing Variables: LST and LSWI

To address generalizability concern, we apply the SENDAI framework—with identical model architecture, hyperparameters, and training pipeline—to two additional remote sensing variables: land surface temperature (LST) and Land Surface Water Index (LSWI), a well-established surface moisture proxy. We additionally note that the seismic waveform experiment in Section M.2 provides a substantially more challenging generalizability test on a fundamentally different physical domain.

### M.1.1. LAND SURFACE TEMPERATURE

**Data and setup.** We reconstruct daytime land surface temperature from MODIS MOD11A1/MYD11A1 thermal imagery (1 km resolution) over the Central Valley, California—a same site used for NDVI in the main text, enabling direct comparison. The seasonal split is identical: April–June (simulation) to July–October (ground truth), with the domain shift manifesting as a $\sim$15–20°C increase in summer temperatures and altered spatial thermal gradients between different land cover types. All model settings are unchanged from the NDVI configuration; only the input data differs.

**Results.** Table 13 reports reconstruction results averaged over three independent runs. SENDAI Jr. achieves SSIM of $0.646 \pm 0.016$ on LST, comparable to its NDVI performance on the same site (SSIM $0.575 \pm 0.048$; Table 1, main text). The full hierarchical architecture provides marginal additional improvement, consistent with the NDVI finding that the Central Valley site—which exhibits predominantly low-frequency domain shift—is well served by SENDAI Jr. alone. Figure 19 presents the qualitative reconstruction results.

### M.1.2. LAND SURFACE WATER INDEX (SURFACE MOISTURE PROXY)

**Data and setup.** We reconstruct the Land Surface Water Index (LSWI), computed as $(\rho_{\mathrm{NIR}} - \rho_{\mathrm{SWIR}})/(\rho_{\mathrm{NIR}} + \rho_{\mathrm{SWIR}})$ from MODIS MOD09GA surface reflectance bands at 500 m resolution. LSWI is a well-established proxy for surface moisture content (Xiao et al., 2004) with values ranging from approximately $-0.5$ (dry bare soil) to $+0.5$ (saturated vegetation). We apply SENDAI to the Tarim Basin site—a same site used for NDVI in the main text—using the same seasonal split (April–June to July–October). This site exhibits extreme moisture contrast between the snowmelt-fed mountain vegetation and the surrounding arid desert, with sharp spatial boundaries that require the full SENDAI architecture.

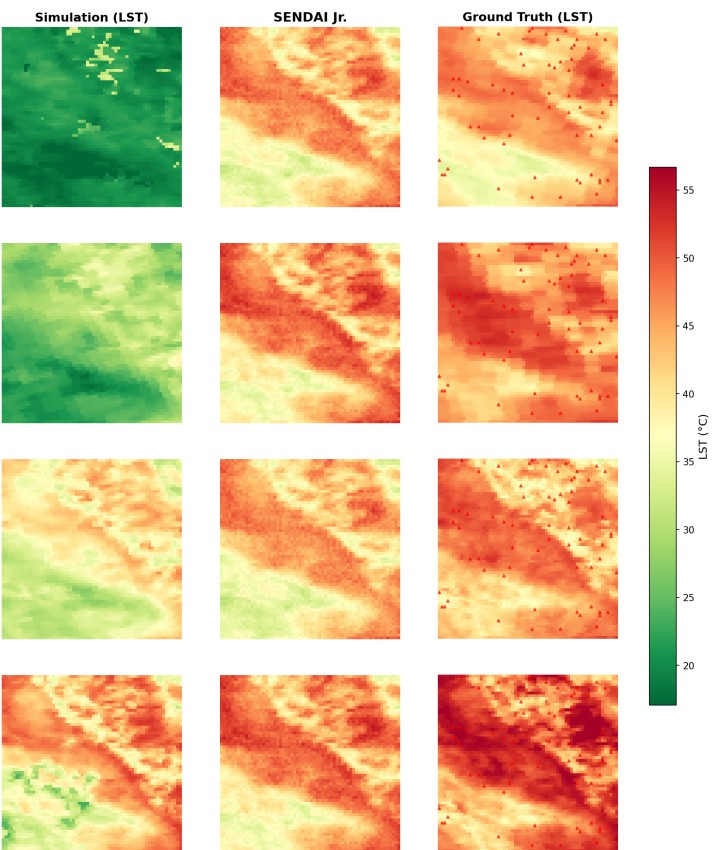

*Figure 19.* SENDAI reconstruction of land surface temperature over the Central Valley, CA. The simulation period (April–June, green/cool) exhibits uniformly lower temperatures than the ground-truth period (July–October, red/hot), yet SENDAI Jr. recovers the spatial thermal structure from 64 sensors.

**Results.** Table 13 reports LSWI reconstruction results. The full SENDAI pipeline achieves SSIM of $0.309 \pm 0.013$, with the HF peeling layers recovering the sharp moisture boundaries that the LF pathway alone cannot resolve. Figure 20 shows the qualitative result: the HF component (rightmost column) captures a coherent spatial correction aligned with the mountain-basin boundary, consistent with the physically meaningful structure observed in the NDVI HF correction for the same site.

### M.1.3. CROSS-VARIABLE CONSISTENCY

A consistent pattern emerges across variables. At the Central Valley site, where the sim-to-real domain shift is predominantly low-frequency, SENDAI Jr. suffices for both NDVI and LST—the LF pathway captures the spatial structure without requiring HF correction. At the Tarim Basin, where sharp spatial boundaries separate distinct land cover types, the full SENDAI multi-scale architecture with hierarchical peeling is needed for both NDVI and LSWI. This consistency confirms that the architectural requirement is governed by *landscape complexity* rather than variable-specific quantities.

**Connection to the seismic experiment.** While LST and LSWI demonstrate generalizability across earth observation variables that share similar spatiotemporal structure with NDVI, to demonstrate that SENDAI extends to fundamentally different physical domains, we also present the seismic waveform reconstruction experiment in Section M.2: seismic data exhibits broadband frequency content, sharp transients, spatially heterogeneous propagation, and severely non-uniform station coverage—properties that are markedly different from those of NDVI. That SENDAI reconstructs seismic waveforms without architectural modification provides substantially stronger evidence of generalizability.

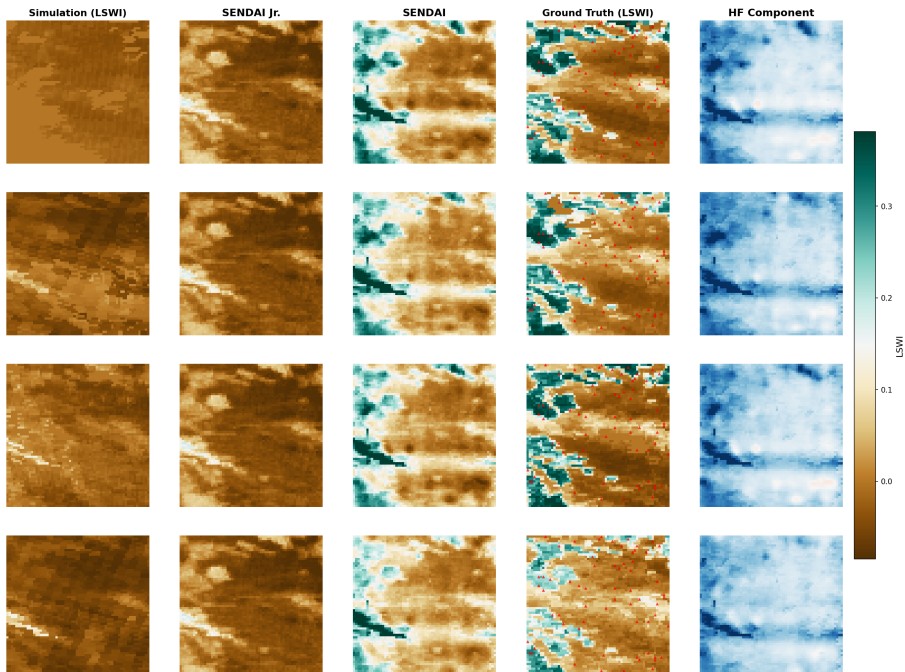

*Figure 20.* SENDAI reconstruction of LSWI (surface moisture proxy) over the Tarim Basin. The HF component captures the sharp mountain-basin moisture boundary that the LF pathway cannot resolve.

*Table 13.* Multi-variable reconstruction performance (mean ± std, 3 runs). All experiments use 64 sensors, identical architecture and hyperparameters. NDVI results from the main text are included for reference.

| Variable | Site | Pipeline | RMSE | SSIM |
|---|---|---|---|---|
| NDVI | Central Valley | SENDAI Jr. | $0.107 \pm 0.002$ | $0.575 \pm 0.048$ |
| LST (°C) | Central Valley | SENDAI Jr. | $3.80 \pm 0.18$ | $0.646 \pm 0.016$ |
| NDVI | Tarim Basin | SENDAI | $0.125 \pm 0.011$ | $0.489 \pm 0.021$ |
| LSWI | Tarim Basin | SENDAI | $0.087 \pm 0.004$ | $0.309 \pm 0.013$ |

## M.2. Seismic Waveform Reconstruction

The NDVI experiments in the main text operate in a regime where the simulation model is capable of providing a reasonable spatial reconstruction prior. We acknowledge that in settings where the data exhibits richer multiscale complexity—broadband frequency content, sharp transients, and spatially heterogeneous propagation—the LF pathway alone may reduce to a mean-amplitude estimate that captures little of the underlying physics. To demonstrate that the SENDAI architecture remains effective in such regimes, we apply it to seismic waveform reconstruction: a domain where the multiscale structure is also substantially complex.

### M.2.1. DATA AND SETUP

We consider the 2013 M 7.1 earthquake off the east coast of Honshu, Japan (event `usc000kn4n`), recorded at 686 globally distributed broadband stations (Hutko et al., 2017). The station network is highly non-uniform (Figure 21): ~300 stations cluster in the United States (epicentral distances 60°–90°, azimuths 315°–45°), with sparse global coverage elsewhere. We partition the stations into 480 observed for training (70%) and 206 for held-out testing (30%). The core SENDAI pipeline is applied without modification to the seismic waveforms.

### M.2.2. RESULTS

Figure 22 presents waveform reconstructions at representative observed and held-out stations. The LF pathway converges to near-zero mean prediction (flat traces, left column), as the high variability of seismic waveforms across stations—in

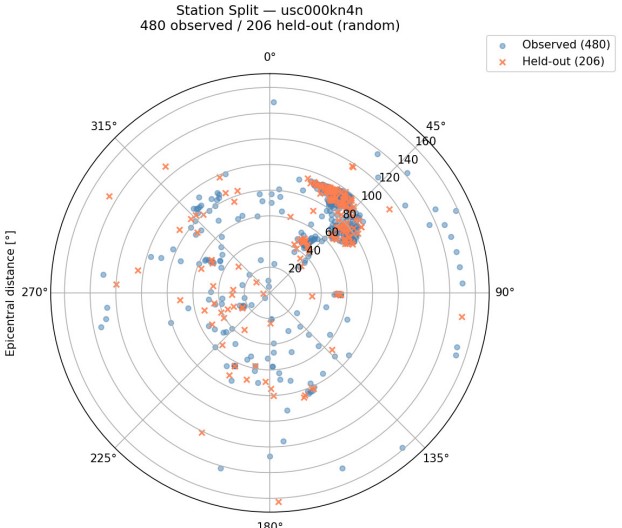

*Figure 21.* Station distribution for event `usc000kn4n` in polar coordinates. Blue: 480 observed stations; orange: 206 held-out stations.

arrival time, amplitude, and waveform character—cannot be adequately compressed into a shared low-dimensional latent representation. The remaining portions of the SENDAI architecture autonomously take over the entire reconstruction task. The full SENDAI output (right column, red) tracks the ground truth at observed stations, capturing both envelope structure and oscillatory detail. At held-out stations, the INR-based spatial interpolation yields structurally coherent waveforms with correct arrival times, envelope shapes, and dominant frequencies.

Figure 23 shows the frequency-resolved PSD ratio (predicted/truth) aggregated across stations. At observed stations, the median ratio remains close to unity within the dominant energy band. At held-out stations, the median falls below unity at higher frequencies and the spread increases, reflecting the reduced constraint available from spatially distant observed stations.

Figure 24 presents example wavelet scalograms for one observed and one held-out station. The reconstruction captures the dominant time–frequency structure at both locations, with the held-out station showing reduced power at higher frequencies consistent with the PSD analysis.

### M.2.3. DISCUSSION

The seismic experiment strengthens the conclusions drawn above by testing SENDAI in a setting that is both naturally skewed in sensor placement and fundamentally different from the NDVI regime. More importantly, in this experiment, the LF pathway contributes negligibly, as the high variability across stations cannot be compressed into a shared low-dimensional representation. Nevertheless, the HF pathway takes over the reconstruction task without architectural modification, confirming that SENDAI's hierarchical design does not assume a particular balance between the LF and HF pathways. The architecture adapts to the data: when a useful simulation prior is available (as in NDVI), the LF pathway exploits it and HF refines the residual; when it is not, HF still handles the task. This adaptability is a consequence of the modular, sensor-supervised multi-scale training procedure rather than any domain-specific tuning.

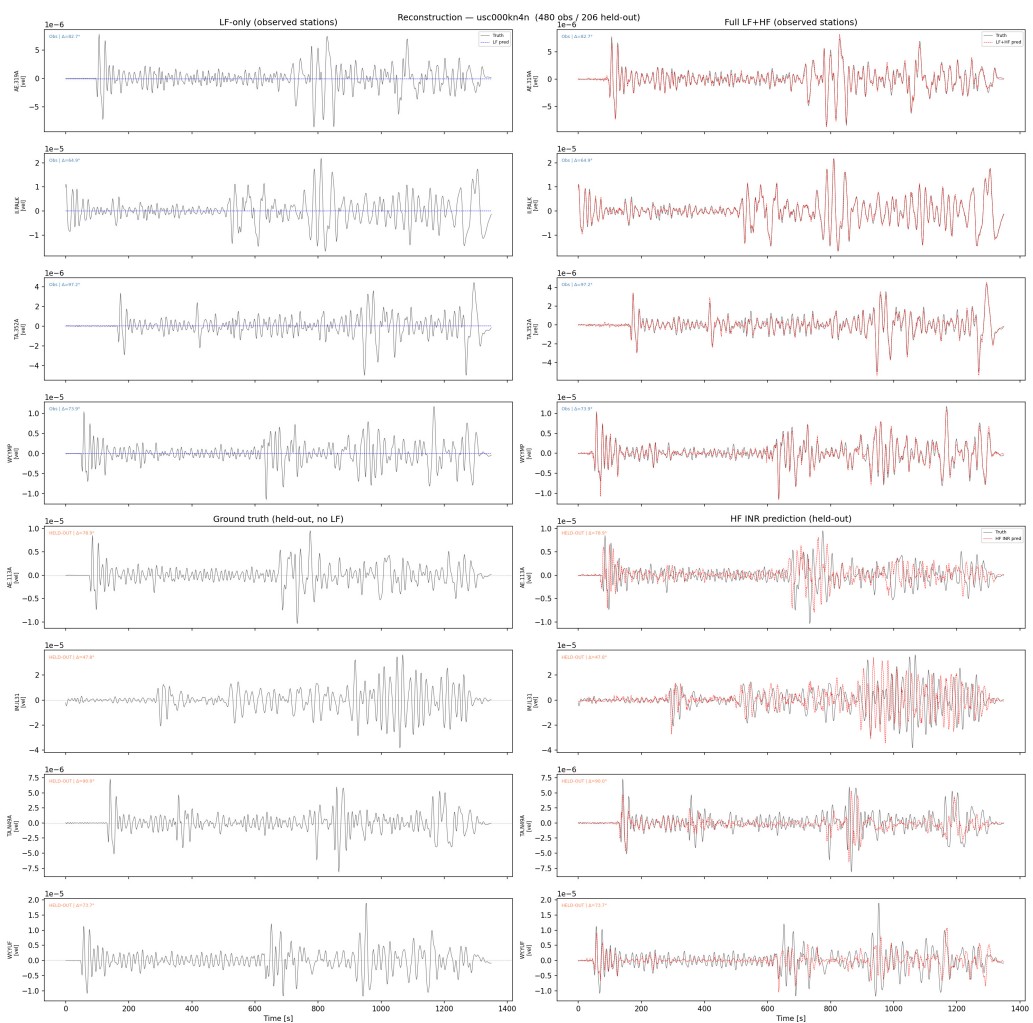

*Figure 22.* Waveform reconstruction. Top four rows present observed stations for LF-only (left, near-zero) and full SENDAI (right, red) vs. ground truth (grey). Bottom four rows present held-out stations.

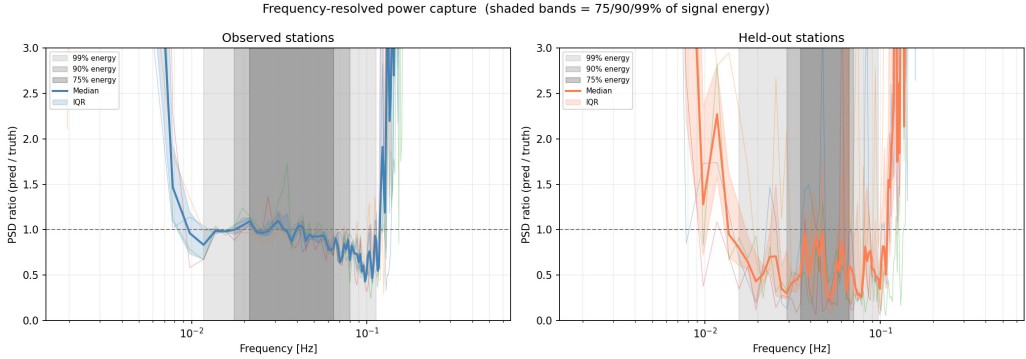

*Figure 23.* Frequency-resolved PSD ratio (pred/truth), aggregated across stations. Solid: median; shaded: IQR. Grey bands: 75/90/99% signal energy.

# N. Domain Adaptation: Extended Discussion

This appendix provides extended discussion of domain adaptation applications introduced in Section 5.

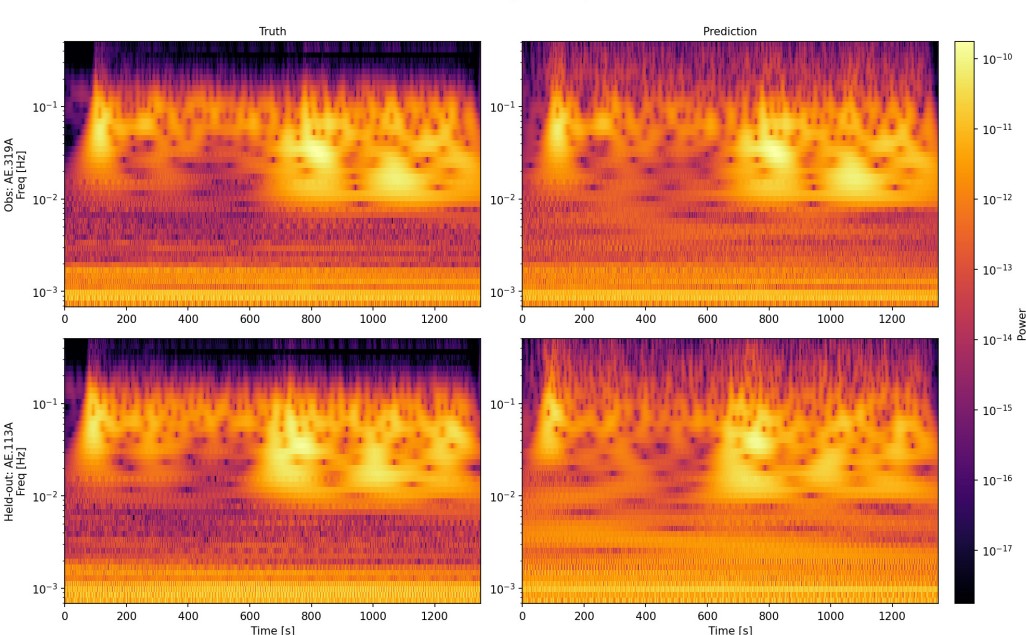

*Figure 24.* Wavelet scalograms (Morlet) for one observed (top) and one held-out (bottom) station. Time–frequency structure is reproduced at both locations.

### N.1. Application to Soil Moisture Retrieval

Consider soil moisture retrieval, where physics-based radiative transfer models provide simulation data but real-world observations exhibit systematic biases from surface roughness, vegetation attenuation, and instrument calibration (Mohanty et al., 2017). The SENDAI framework provides a principled approach to bridge such gaps: pretrain on simulation or reference-period data to establish spatial priors, then align to target conditions using sparse real observations from in-situ networks.

### N.2. Application to Land Surface Temperature

Similarly, land surface temperature products exhibit seasonal and diurnal biases that complicate multi-temporal analysis (Li et al., 2013). Physics-based energy balance models can provide simulation priors, while sparse thermal observations from clear-sky periods serve as target-domain data for latent-space alignment.

### N.3. Implications for Operational Monitoring

The implications for operational monitoring systems are substantial. Near-real-time agricultural monitoring—critical for yield forecasting, drought assessment, and food security early warning (Gao & Zhang, 2021; Weiss et al., 2020)—requires rapid processing of incoming satellite data streams. Current systems face a fundamental tension: sophisticated reconstruction methods improve data quality but introduce latency; rapid processing preserves timeliness but sacrifices accuracy. SENDAI's short inference times per scene suggest a viable path toward high-quality reconstruction within operational latency constraints.

Furthermore, the framework's ability to reconstruct from sparse observations has implications for monitoring in resource-constrained settings. Regions with limited ground-station coverage, intermittent connectivity, or bandwidth constraints—common in developing nations and remote areas—could potentially achieve comparable monitoring fidelity by transmitting only sparse sensor measurements alongside periodically updated model parameters, rather than full imagery.

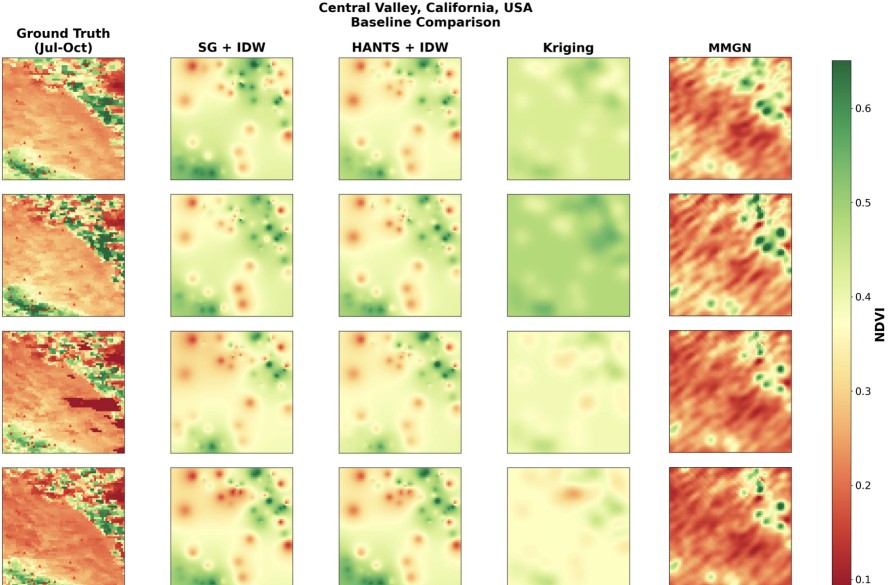

*Figure 25.* Baseline reconstruction comparison for Central Valley, California. Each row represents an equally-spaced temporal frame within the ground truth period. The irrigated cropland landscape exhibits field-scale heterogeneity that baseline interpolation methods fail to faithfully reproduce. Note the bullseye artifacts in IDW-based methods and over-smoothing in Kriging.

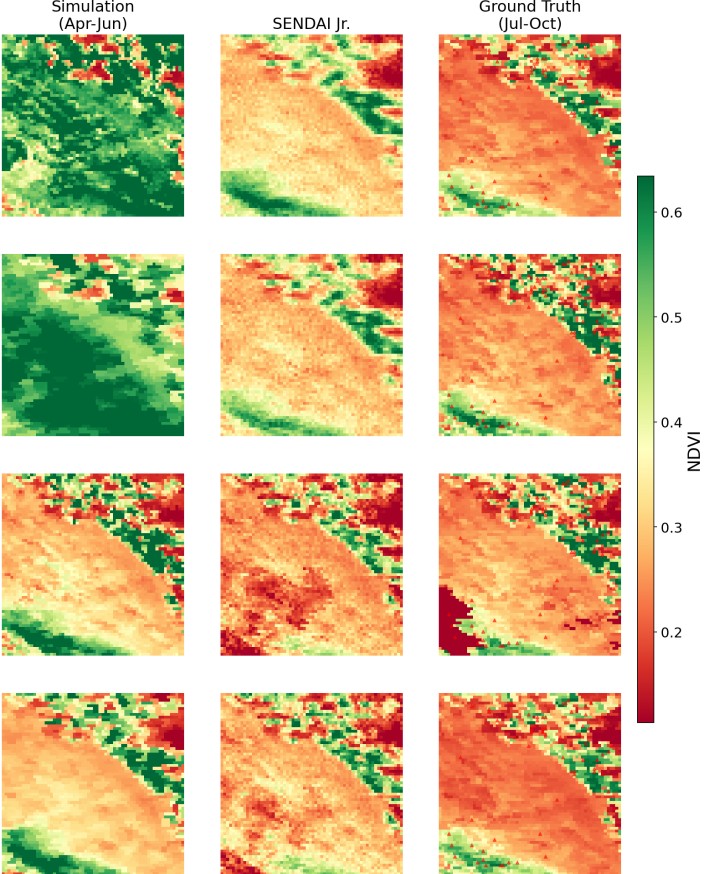

*Figure 26.* SENDAI Jr. reconstruction for Central Valley, California. Each row represents an equally-spaced temporal frame within the ground truth period (July–October). Columns show simulation-period reference (April–June), SENDAI Jr. reconstruction, and ground truth. Red markers indicate sensor locations. The SSIM of 0.5747 reflects the preservation of field-scale spatial structure.

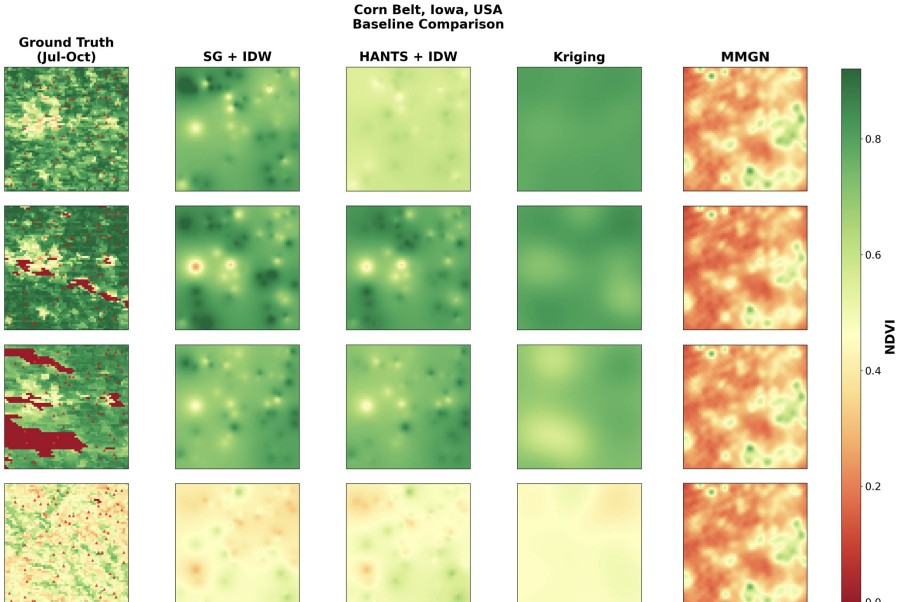

*Figure 27.* Baseline reconstruction comparison for the Iowa Corn Belt. Each row corresponds to an equally-spaced temporal frame during July–October. Despite the relatively homogeneous landscape, baseline methods produce interpolation artifacts that obscure the underlying field structure. Kriging achieves the best RMSE but worst SSIM, demonstrating the importance of structural metrics.

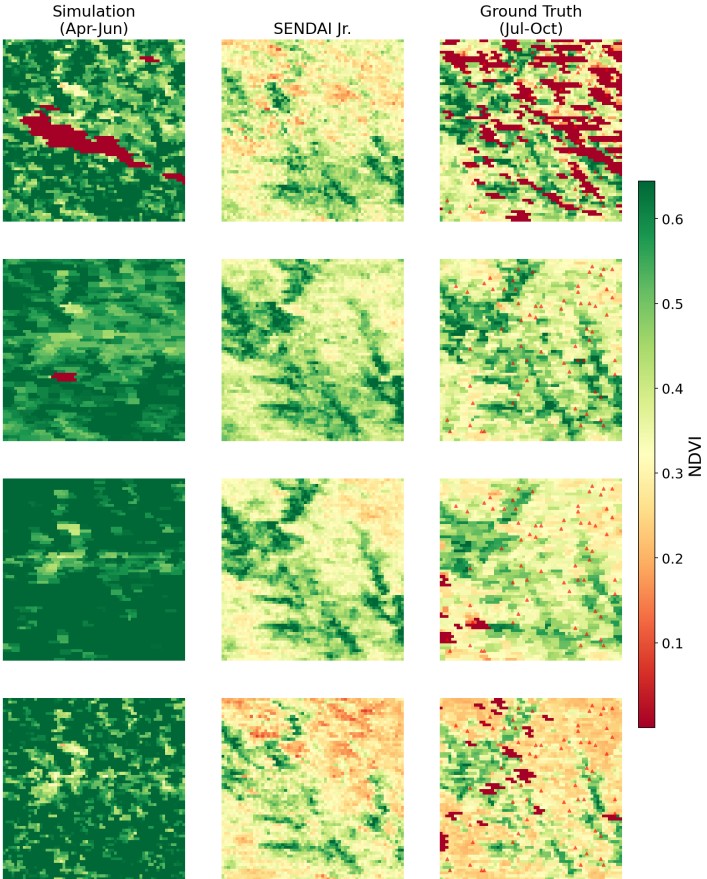

*Figure 28.* SENDAI Jr. reconstruction for the Iowa Corn Belt. Each row corresponds to an equally-spaced temporal frame during July–October. The framework captures the characteristic high-NDVI patterns of mature corn and soybean crops from sparse sensor observations, achieving SSIM of 0.4530.

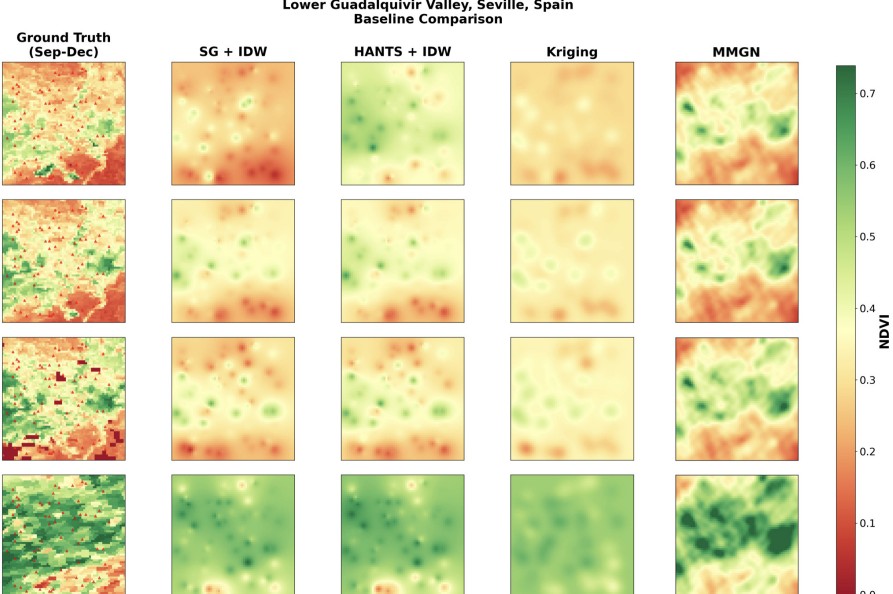

*Figure 29.* Baseline reconstruction comparison for the Guadalquivir Valley, Spain. Each row represents an equally-spaced temporal frame. The mixed agricultural landscape with varying field sizes challenges interpolation-based methods. All baselines achieve SSIM below 0.19, indicating severe structural degradation.

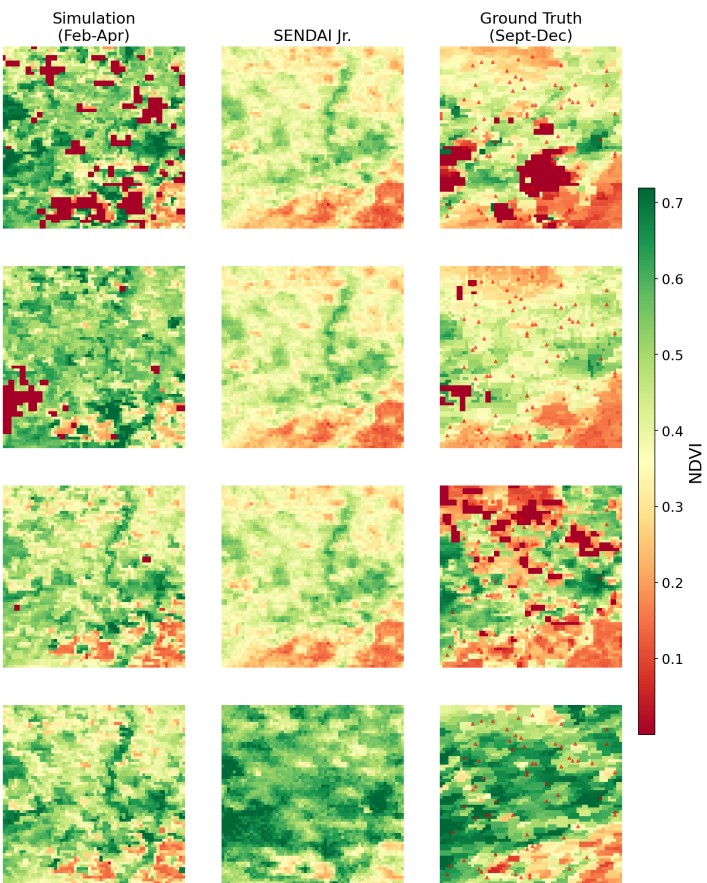

*Figure 30.* SENDAI Jr. reconstruction for the Guadalquivir Valley, Spain. Temporal frames span September–December ground truth observations, reconstructed from February–April simulation training. The model recovers coherent spatial patterns including agricultural field boundaries and regional vegetation gradients, achieving SSIM of 0.3655.

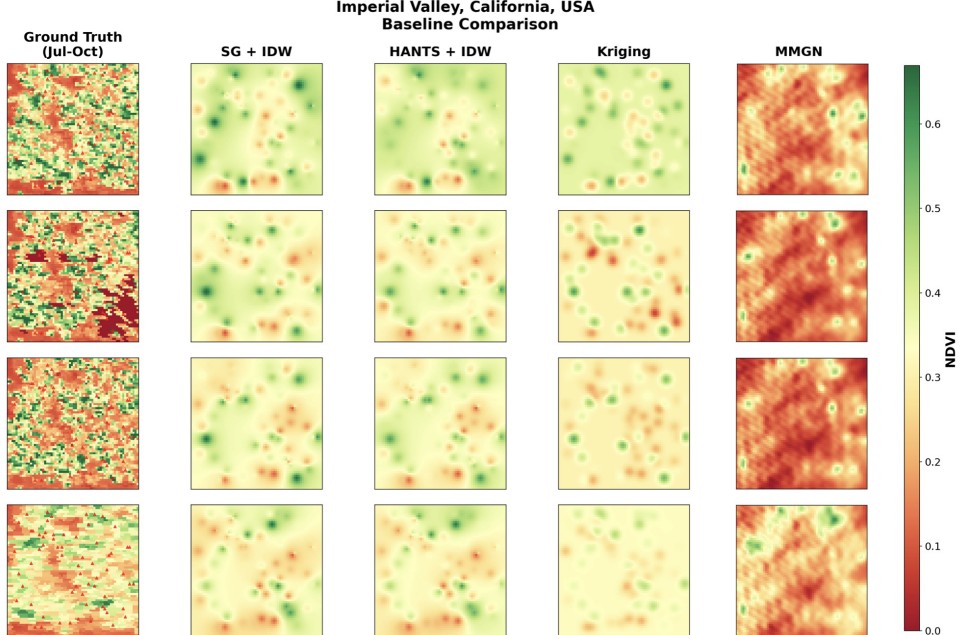

*Figure 31.* Baseline reconstruction comparison for Imperial Valley, California. Each row corresponds to an equally-spaced temporal frame. The rectilinear irrigation infrastructure creates sharp NDVI boundaries that baseline methods fail to preserve, producing either circular artifacts or smooth gradients. Baseline SSIM values range from 0.0916 to 0.1123.

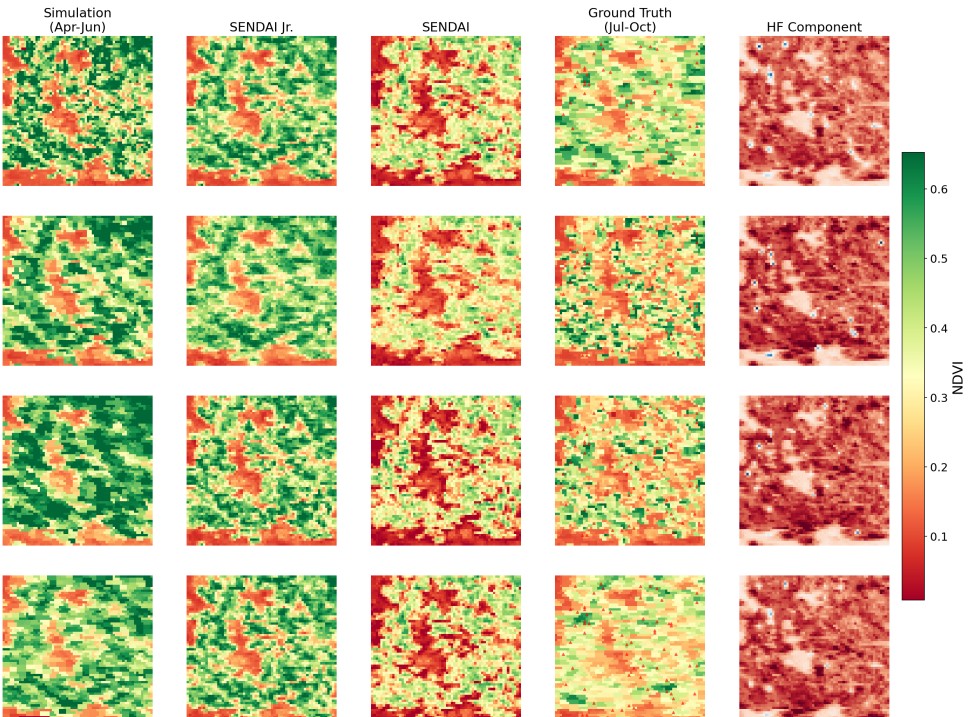

*Figure 32.* Full Sendai hierarchical multi-scale DA-SHRED reconstruction for Imperial Valley, California. Each row corresponds to an equally-spaced temporal frame during July–October. Columns show: simulation reference (April–June), SENDAI Jr. stage, full SENDAI hierarchical reconstruction, ground truth, and the learned HF correction component. SENDAI captures fine-scale field boundaries, achieving SSIM of 0.4668 in the full pipeline.

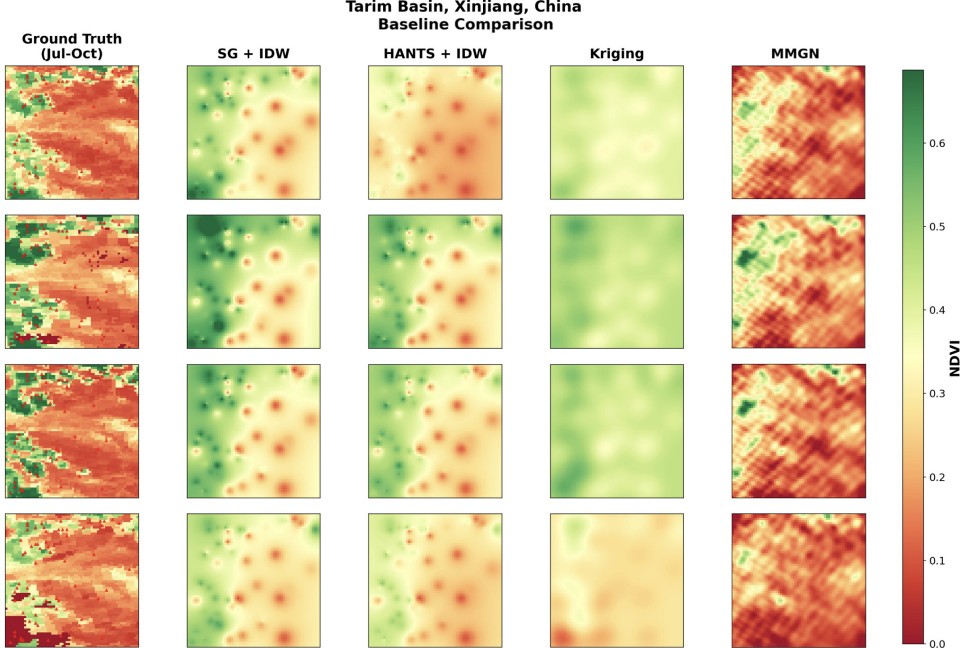

*Figure 33.* Baseline reconstruction comparison for the Tarim Basin, Xinjiang, China. Each row represents an equally-spaced temporal frame. The sharp mountain-basin boundaries that define this landscape are completely lost in all baseline reconstructions, demonstrating the fundamental limitations of interpolation-based approaches for heterogeneous terrain. Kriging achieves SSIM of only 0.0449.

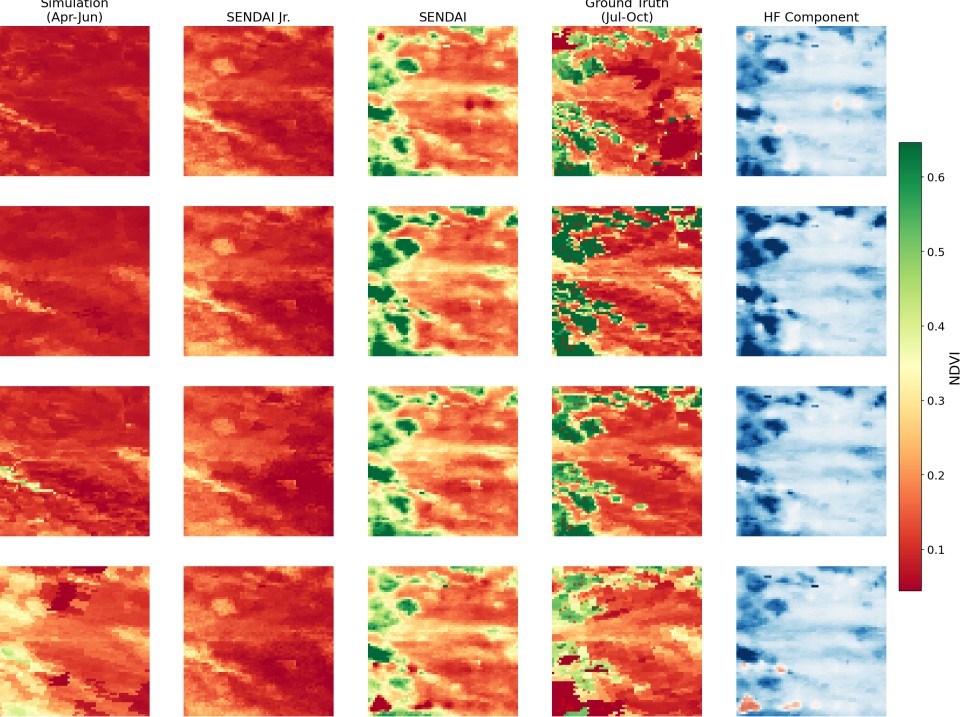

*Figure 34.* Full Sendai hierarchical multi-scale DA-SHRED reconstruction for the Tarim Basin, Xinjiang, China. Temporal frames span July–October ground truth observations. The HF component captures sharp oasis-desert boundaries that the smooth LF decoder fails to resolve, with the full pipeline achieving SSIM of 0.4777 and 33.9% RMSE improvement over SENDAI Jr.

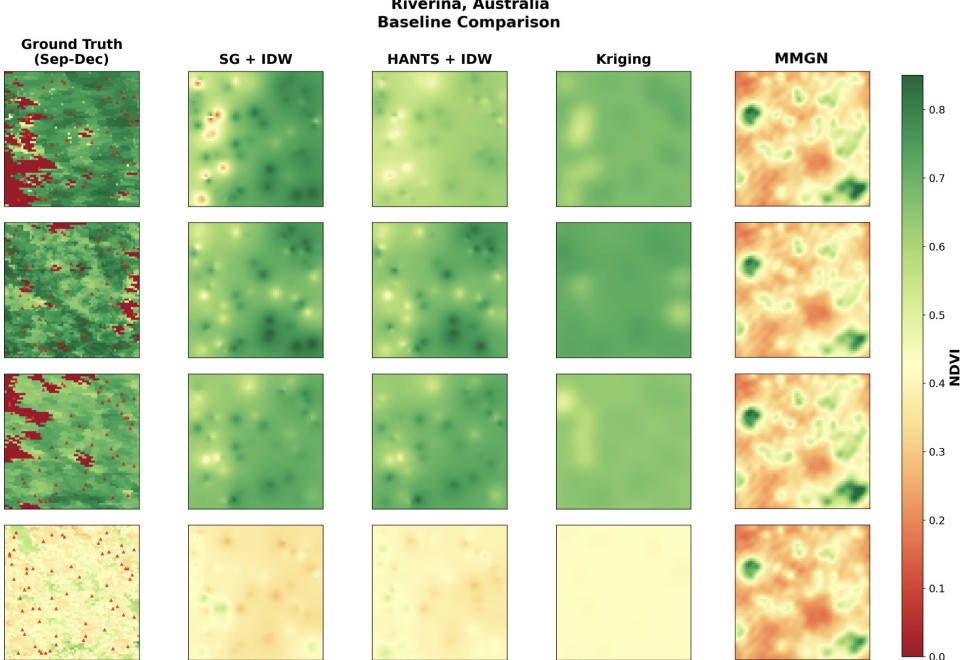

*Figure 35.* Baseline reconstruction comparison for Riverina, Australia. Each row represents an equally-spaced temporal frame. The mixed cropping landscape exhibits more gradual spatial transitions than other sites, but baseline methods still produce artifacts and over-smoothing. Baseline SSIM values range from 0.0272 to 0.1359.

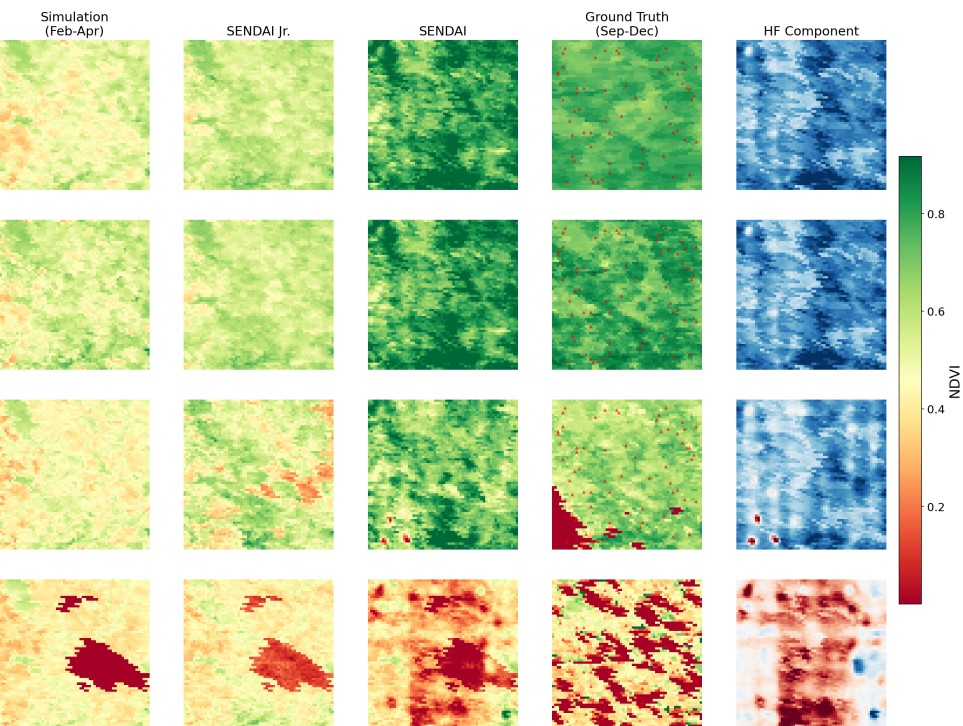

*Figure 36.* Full SENDAI hierarchical multi-scale DA-SHRED reconstruction for Riverina, Australia. Each row represents an equally-spaced temporal frame during the September–December ground truth period. The Southern Hemisphere site tests generalization to reversed seasonal timing, with the HF component showing more diffuse structure consistent with the gradual spatial transitions in this mixed cropping region. Full pipeline achieves SSIM of 0.3354.

