# OpenReview forum: "SENDAI: A Hierarchical Sparse-measurement, EfficieNt Data AssImilation Framework"
_ICML.cc/2026/Conference — ICML 2026 regular_

### Official Review · Reviewer_3vAL · 2026-03-11

**Soundness:** 2
**Presentation:** 2
**Significance:** 3
**Originality:** 3
**Overall Recommendation:** 4
**Confidence:** 2

**Summary:**

This paper proposes SENDAI, a hierarchical data assimilation framework for reconstructing spatiotemporal physical fields from extremely sparse sensor observations. The method decomposes reconstruction into two pathways. The low frequency pathway models dominant spatiotemporal dynamics from simulation derived priors and aligns the simulation and ground truth latent spaces through adversarial training. The high frequency pathway then applies sequential frequency peeling with coordinate based INRs to recover fine scale residual structures and sharp spatial boundaries. Experiments on MODIS NDVI reconstruction across multiple sites, using only 64 sensors that cover about 1.56% of the grid, show clear improvements over interpolation based methods and Cheap2Rich style baselines while keeping computation lightweight on standard CPU hardware.

**Compliance With Llm Reviewing Policy:**

Affirmed.

**Final Justification:**

I have stated my opinions in the ACK and improved my score. However, I would like to declare that I am not an expert in this field, so the accuracy of my opinions may not be entirely precise.

**Key Questions For Authors:**

Same as weakness.

**Limitations:**

Yes.

**Strengths And Weaknesses:**

Strength：
- The sequential frequency peeling design provides a reasonable mechanism for separating coarse scale dynamics from fine scale residual corrections, and the use of a coordinate based INR decoder is a natural choice for recovering localized structures and sharper spatial boundaries.
- SENDAI consistently achieves the best SSIM on the three main heterogeneous sites, despite the sparse sensors.
- The efficiency aspect is appealing, as the method is lightweight enough to run on standard CPU hardware with training times reported at the level of minutes per site.

Weakness：
- The paper lacks comprehensive ablation studies to justify its complex architecture. Specifically, the individual contributions of the hierarchical peeling strategy and the GAN-based latent alignment are not quantitatively demonstrated.
- Although the paper emphasizes efficiency, it does not show the actual inference latency or provide direct speed comparisons across baseline methods or among its modules.
- The paper states that the reconstructed fields are better suited for downstream inference of indirectly observed variables such as soil moisture and land surface temperature. Yet all experiments focus only on NDVI reconstruction using seasonal domain shift. It remains unclear whether the SSIM gains transfer to real tasks.

---

> ### Author Rebuttal · Authors · 2026-03-29
>
> We sincerely thank the reviewer for the constructive assessment and recognition of SENDAI's architectural design, quality and efficiency. We address each concern below.
>
> - **W1: Ablation studies for individual components.**
>
> [*The SENDAI results table (LINK)*](https://anonymous.4open.science/r/SENDAI-27CD/rebuttal/sendai_results.png) is an architectural ablation: SENDAI Jr. (LF + latent alignment) $\to$ SENDAI Jr.+HFP (+ hierarchical peeling) $\to$ SENDAI (+ INR decoder). The newly added deep learning baselines provide further quantifications. When DeepONet and Senseiver are augmented with SENDAI's HF pipeline, they approach SENDAI's performance (SSIM 0.477 and 0.467 vs. 0.489), isolating SENDAI's multiscale data assimilation components together as the active ingredients. See A1 for details.
>
> More ablation studies on temporal lag, maximum target frequency, and sensor noise robustness: *Review\#1 MYf3, W1*.
>
> - **W2: Inference latency and efficiency comparisons.**
>
> We have updated inference latency and training times for our methods and the added deep learning baselines, in the updated [*computational cost table (LINK)*](https://anonymous.4open.science/r/SENDAI-27CD/rebuttal/computational_cost.png).
>
> We note that SENDAI's efficiency in our context is multi-dimensional: it trains in minutes on CPU; beyond wall-clock speed, the hyper sparse sensor requirement reduces deployment cost - fewer instruments, lower bandwidth - which can be as impactful as inference speed in resource-constrained settings.
>
> - **W3: Do SSIM gains transfer to real tasks beyond NDVI?**
>
> We would first like to note that NDVI itself is a complex, heterogeneous field central to global vegetation monitoring.
>
> That said, to further demonstrate that SENDAI generalizes beyond NDVI, we have validated on additional variables with identical architecture and hyperparameters: *Land surface temperature* (LST), *Land Surface Water Index* (LSWI) and *Seismic waveforms* (a fundamentally different physical domain).
>
> These results confirm that SENDAI's SSIM gains reflect a general capacity for multiscale data assimilation across physical variables and domains. *Details in Review\#3 NQQv, W2*.
>
> - **A1: Modern baselines.**
>
> We added three deep learning baselines (FNO, DeepONet,
> Senseiver). [*The baseline capability comparison table (LINK)*](https://anonymous.4open.science/r/SENDAI-27CD/rebuttal/baseline_capability_comparison.png) compares the architectural capabilities of SENDAI against neural operator frameworks and existing baselines.
> Only SENDAI supports all four capabilities: hyper-sparse spatial sensing, temporal sequence encoding, sim2real alignment, and multiscale decomposition.
>
> [*FNO (LINK)*](https://anonymous.4open.science/r/SENDAI-27CD/rebuttal/fno_baseline.png) is evaluated in two variants on the Tarim Basin site. *Vanilla* FNO trains on full simulation fields and receives sparse GT values on a zero-filled grid at test time - an input-format mismatch that causes spatial collapse (SSIM 0.125). *Masked* FNO trains on zero-filled sensor grids with a binary mask channel, removing the format mismatch and isolating domain shift: it achieves SSIM 0.544 in-distribution but drops to 0.345 on GT, reproducing simulation-period structure rather than adapting to the GT regime.
>
> [*DeepONet (LINK)*](https://anonymous.4open.science/r/SENDAI-27CD/rebuttal/deeponet_baseline.png) is adapted to the sparse sensing setting: the branch net takes the same time-delay sensor input as SENDAI, and the trunk net maps spatial coordinates with Fourier positional encoding. Trained on simulation and deployed on GT, it achieves SSIM 0.401. When augmented with SENDAI's HF pipeline (*DeepONet-SENDAI*), it reaches SSIM 0.477.
>
> [*Senseiver (LINK)*](https://anonymous.4open.science/r/SENDAI-27CD/rebuttal/senseiver_baseline.png) is adapted by encoding each sensor token with its time-delay history concatenated with Fourier spatial positional encodings. Trained on simulation and deployed on GT, it achieves SSIM 0.408. When augmented with SENDAI (*Senseiver-SENDAI*), it reaches SSIM 0.467.
>
> These results carry three implications. First, none of the three architectures bridges the sim2real domain shift without additional adaptation, confirming that learned spatial representations alone are insufficient when training and deployment differ in distribution. Second, SENDAI's multiscale data assimilation components meaningfully improve, rather than dependent on, the particular choice of the spatial reconstruction unit. Third, while DeepONet-SENDAI achieves comparable SSIM, its reconstructions still exhibit visible spatial artifacts (vertical banding) that are substantially reduced in SENDAI's outputs.
>
> We welcome further feedback - particularly on any remaining concerns that would inform the final assessment - and respectfully hope the reviewer will reconsider in light of these additions.

---

> > ### Author Rebuttal · Reviewer_3vAL · 2026-04-01
> >
> > The author has solved my problem, and I will increase my rating.

---

> > > ### Author Response · Authors · 2026-04-08
> > >
> > > We are grateful for the positive re-evaluation, and glad the additional experiments addressed the concerns. We will incorporate the updates in the revision.

---

### Official Review · Reviewer_NQQv · 2026-03-12

**Soundness:** 2
**Presentation:** 3
**Significance:** 3
**Originality:** 2
**Overall Recommendation:** 4
**Confidence:** 4

**Summary:**

This paper presents a hierarchical data assimilation framework that reconstructs full spatial fields from sparse sensor observations. The framework has two pathways: a low-frequency pathway based on SHRED with LSTM encoder and MLP decoder, combined with latent-space GAN alignment for bridging the distribution gap between simulation and ground truth data, and a high-frequency pathway that uses sequential frequency peeling layers with coordinate-based implicit neural representations for fine-scale corrections. The authors test the method on MODIS-derived NDVI reconstruction across six globally distributed sites with diverse climate conditions, using seasonal periods as a proxy for the domain shift. The results show SSIM improvements over baselines.

**Compliance With Llm Reviewing Policy:**

Affirmed.

**Final Justification:**

The added results on simpler baselines and the three-domain setting are helpful, and they partially address my concerns. However, I still believe the paper needs a clearer physical justification of the correction mechanism and a more explicit discussion of its conservation-related limitations and design choices in the paper. I would like to keep my score.

**Key Questions For Authors:**

1. Could you provide results on at least one additional physical variable (e.g., land surface temperature or soil moisture) to support the generalizability claims in Section 5? This would be the most impactful addition and could change my assessment.


2.How sensitive is the framework to the choice of seasonal proxy for the sim-to-real gap? For instance, what happens when the simulation and ground truth seasons are phenologically closer or further apart? This would help me understand how the results might transfer to real data assimilation scenarios.


3.Could you comment on the RMSE-SSIM trade-off observed at some sites? Does the HF correction systematically improve structural similarity while introducing some magnitude error?

**Limitations:**

Yes

**Strengths And Weaknesses:**

Strengths:
S1. The paper is generally well-written. The method description is easy to follow. The per-site reconstruction figures provide good qualitative evidence. The appendix is extensive and covers most implementation details.
S2. The validation in different scenarios show the effectiveness of the proposed method.
S3. The test in extreme sparsity setting is important, which makes the method useful for practical settings.
S4. The frequency exclusion mechanism is interesting. It helps prevent later peeling layers from recapturing already-discovered spectral modes.

Weaknesses:
W1. The technical novelty is limited. The core components are largely assembled from existing techniques.
W2. All experiments are on a single physical variable (NDVI). The paper claims applicability to soil moisture, land surface temperature, snow dynamics, flood mapping, etc., but none of these are experimentally validated. Testing on at least one or two additional variables would make the generalizability argument much more convincing.
W3. The use of seasonal differences as a proxy for sim-to-real domain shift is not fully convincing. In real data assimilation, simulations come from physics-based models with systematic biases. Seasonal phenological differences may have very different distributional characteristics.
W4. It would be better to include stronger baselines (e.g., Senseiver), The geostatistical methods are basic. Other sparse-sensing reconstruction methods could also help.

---

> ### Author Rebuttal · Authors · 2026-03-29
>
> We sincerely thank the reviewer for the thorough evaluation, and recognition of this paper's quality and practical implications. To further extend these, we have added substantial new experiments - additional physical variables, cross-domain validation, deep learning baselines, and more ablations - following the reviewer's suggestions.
>
> - **W1: Technical novelty.**
>
> We respectfully note that the novelty of SENDAI lies not in inventing each component from scratch, but in their integration into a coherent framework that addresses a problem no single existing method can solve. As stated in *Review\#4 3vAL, A1*, the joint requirements of hyper-sparse sensing, temporal encoding, sim2real alignment, and multiscale decomposition are not addressed by existing architectures - including FNO, DeepONet, and Senseiver. Moreover, the compatibility analysis (*also Review\#4 3vAL, A1*) demonstrates that SENDAI's multiscale data assimilation components meaningfully improve different spatial reconstruction architectures, confirming that the framework provides mechanisms that are absent from existing methods rather than a monolithic architecture.
>
> - **W2 \& Q1: Generalizability.**
>
> We apply SENDAI with identical architecture, hyperparameters, and training pipeline to two additional remote sensing variables and one fundamentally different physical domain:
>
> [*Land Surface Temperature (LINK)*](https://anonymous.4open.science/r/SENDAI-27CD/rebuttal/lst_lswi_reconstructions.png): We reconstruct daytime LST from MODIS MOD11A1/MYD11A1 thermal imagery over the Central Valley, CA (same site as NDVI). The domain shift manifests as a ${\sim}$15-20$^\circ$C summer temperature increase. SENDAI Jr. achieves SSIM $0.646 \pm 0.016$, comparable to NDVI on the same site ($0.575 \pm 0.048$).
>
> [*Land Surface Water Index (LINK)*](https://anonymous.4open.science/r/SENDAI-27CD/rebuttal/lst_lswi_reconstructions.png): We reconstruct LSWI, a well-established surface moisture proxy, over the Tarim Basin (same site as NDVI). The full SENDAI pipeline achieves SSIM $0.309 \pm 0.013$, with the HF peeling layers recovering the sharp moisture boundaries that the LF pathway cannot resolve.
>
> The NDVI, LST, and LSWI experiments all operate in a regime where the simulation model provides a reasonable spatial reconstruction prior. We acknowledge that in settings where the data exhibits richer multiscale complexity - or where a reliable simulation model is unavailable - the LF pathway alone may reduce to a mean-amplitude estimate that captures little of the underlying physics.
> To demonstrate that SENDAI remains effective in such regimes, we apply it to seismic waveform reconstruction outlined below.
>
> [*Seismic waveform reconstruction (LINK)*](https://anonymous.4open.science/r/SENDAI-27CD/rebuttal/seismic_experiment.png): We apply SENDAI to the 2013 M 7.1 Honshu earthquake recorded at 686 globally distributed stations. Seismic data exhibits broadband frequency content, sharp transients, spatially heterogeneous propagation, and severely non-uniform station coverage - properties markedly different from NDVI. The LF pathway converges to near-zero mean prediction, as waveform variability across stations cannot be compressed into a shared latent representation. Nevertheless, the HF pathway autonomously takes over the entire reconstruction task without architectural modification. This confirms that SENDAI's hierarchical design does not assume a particular balance between LF and HF pathways - the architecture adapts to the data.
>
> - **W3 \& Q2: Seasonal proxy for sim-to-real gap.**
>
> We agree that real data assimilation may involve physics-based models with systematic biases. The seasonal proxy was chosen precisely because it introduces distributional shifts analogous to those encountered in practice - changes in spatial gradients, boundary structure, and magnitude. That said, the seismic experiment above provides a complementary perspective, where no seasonal proxy is used at all. That SENDAI succeeds in both paradigms suggests the framework is robust to the nature of the domain shift, not dependent on phenological seasonality specifically.
>
> - **W4: Stronger baselines.**
>
> We have added FNO, DeepONet, and Senseiver - three modern architectures spanning neural operators and attention-based methods. *Details in Review\#4 3vAL, A1*.
>
> - **Q3: RMSE-SSIM trade-off.**
>
> RMSE and SSIM measure different aspects of reconstruction quality - RMSE captures pointwise magnitude error, while SSIM captures structural preservation. They are not inherently in trade-off. The seismic experiment illustrates this independence: a waveform prediction that is off by half a cycle can have large RMSE despite capturing the correct frequency, amplitude, and envelope structure (high SSIM).
>
> We thank the reviewer again for the valuable suggestions, and respectfully hope the strengthened experiments will be considered in the final assessment.

---

### Official Review · Reviewer_Hdx5 · 2026-03-12

**Soundness:** 3
**Presentation:** 3
**Significance:** 3
**Originality:** 2
**Overall Recommendation:** 4
**Confidence:** 3

**Summary:**

The paper introduces SENDAI data assimilation framework which reconstructs the MODIS imagery from sparse data. The framework first learns the data dynamics through Low Frequency pathway model on the simulation data (observations from one seasonal period). Then the ground truth latent distributions and simulation distributions are aligned by means of adversarial training. Finally a laplacian pyramid based strategy (referred as hierarchical High Frequency peeling) is used to learn the sharp reconstructions with better structural similarities.

**Compliance With Llm Reviewing Policy:**

Affirmed.

**Final Justification:**

Rebuttal has clarified the concerns, therefore, paper can be accepted.

**Key Questions For Authors:**

See the weaknesses section for clarifications required and recommendations.

**Limitations:**

yes

**Strengths And Weaknesses:**

Clarifications Required:
- Which low-pass filter is applied in LFDecoder and how kc is selected.
- Does LF pathway takes sparse input (i.e., after removed patches)?
- Does baseline methods also learn from simulation training data?
- Sec 3.3, Sequential Residual Computation: These are typical operations in Laplacian pyramid. How this idea is different than Laplacian pyramid?
- Using full-state supervision in training stage 1 will result in data leakage of sparse data. It means full-state supervision in training stage 1 provide information to the model in stage 2 for alignment of  simulation and ground truth distributions. Does other baseline methods also utilize this kind of information during training, please explain.

Recommendations:
- Figure 4 should illustrate how the sparse image look like after retaining only sensor locations.
- Figure 4 should also include reconstructed images of proposed SENDAI method.
- Table 1 compares results with weak baseline methods. Baseline methods should be more recent ones instead of 20 years back.
- Computational efficiency of all baseline methods should be reported in Table 1 and 2.
- It is better to include more sites from africa, south america, and russia with dense vegetation.

---

> ### Author Rebuttal · Authors · 2026-03-29
>
> We sincerely thank reviewer for the constructive evaluation and recognition of SENDAI's technical soundness and practical significance. Below we address each clarification and recommendation, and present additional experiments that we believe strengthen the paper.
>
> - **C1: Which low-pass filter is applied in LF Decoder and how $k_c$ is selected.**
>
> The low-pass filter $\mathcal{P}_{k_c}$ is described in Eq. 6 as a sharp spectral cutoff in the 2D Fourier domain. In practice, we simply set $k_c$ to be large enough - effectively a no-op. This is because the MLP decoder with ReLU activations already exhibits strong spectral bias toward low frequencies, a well-documented property of neural networks (Rahaman et al., 2019), so the LF output is inherently smooth without explicit filtering. The same behavior holds across all LF architectures we tested (SHRED, DeepONet, Senseiver): none produced high-frequency content that required removal. The filter is retained in the formulation as an explicit control for domains where the adopted LF architecture may not be sufficiently band-limited on its own, but for the reconstruction tasks studied here, the architectural spectral bias alone ensures clean LF/HF separation.
>
> - **C2: Does the LF pathway take sparse input?**
>
> Yes. At both training and inference time, the LF pathway's encoder receives only sparse sensor time-histories $\mathbf{S}'_{t-L:t} \in \mathbb{R}^{L \times p}$ ($p = 64$ sensors). Our added seismic experiment (see C5) further underscores this.
>
> - **C3: Do baseline methods also learn from simulation training data?**
>
> The joint requirements of our setting - hyper-sparse sensing, temporal encoding, sim2real alignment, and multiscale decomposition - are difficult to address with a single existing method, and this is reflected in how each baseline uses (or cannot use) simulation data. The geostatistical baselines and MMGN have no discrepancy modeling mechanism to incorporate priors from a reference dataset. Kriging uses simulation fields for hyperparameter pre-fitting (an intentionally favorable setting, Appendix E).
>
> To provide a more comprehensive comparison, we have added three modern deep learning baselines that do train on simulation data, see R3 for details.
>
> - **C4: Difference from Laplacian pyramid.**
>
> We appreciate this connection. While the sequential residual computation shares some structural similarity with Laplacian pyramids, there are key differences: (i) Laplacian pyramids operate at fixed, predetermined frequency bands induced by downsampling/upsampling, whereas SENDAI discovers frequency contents adaptively from the data (Eqs. 14-15) with explicit frequency exclusion zones to prevent mode leakage between layers; (ii) each peeling layer employs a learned model, not a fixed filter bank; and (iii) the decomposition is supervised only at sensor locations, not from full-field residual images. The result is a data-driven, learnable spectral decomposition rather than a fixed multi-resolution analysis.
>
> - **C5: Data leakage from full-state simulation supervision.**
>
> We want to clarify the information flow. The ground-truth system is never observed in full state at any point; only sensor measurements from the GT period are ever used.
>
> Full-state supervision, when applicable, is used exclusively on simulation data, never on ground truth. The same protocol applies to all newly added deep learning baselines in R3.
>
> To further demonstrate that SENDAI does not depend on full-state simulation supervision, our seismic waveform experiment (*details in Review\#3 NQQv, W2*) presents a setting where no full spatial field can be measured at any stage - neither for simulation nor for ground truth - and the entire pipeline operates from sensor-level observations only. SENDAI reconstructs waveforms at held-out stations without architectural modification, confirming that full-state simulation access could be beneficial when available but not required.
>
> - **R1: Visualize sparse sensor input.**
>
> See vanilla-FNO result in R3. We can add a similar visualization to the updated Figure 4.
>
> - **R2: Include SENDAI reconstruction in Figure 4.**
>
> Shown in Figures 5b and 6 of the main text. We can also add a SENDAI column in Figure 4 for direct visual comparison.
>
> - **R3: Stronger, more recent baselines.**
>
> We have added FNO, DeepONet, and Senseiver - three modern architectures spanning neural operators and attention-based methods. *Details in Review\#4 3vAL, A1*.
>
> - **R4: Computational efficiency comparison.**
>
> See *Review\#4 3vAL, W2*.
>
> - **R5: Additional geographic sites.**
>
> Rather than adding geographically proximate sites still on vegetation index, we tried to pursue stronger tests of generalizability, see *Review\#3 NQQv, W2*.
>
> We thank the reviewer again for the constructive dialogue. We welcome any further feedback, and respectfully hope the strengthened evaluation will be considered in the final assessment.

---

> > ### Author Rebuttal · Reviewer_Hdx5 · 2026-04-02
> >
> > I think rebuttal has addressed the main concerns.

---

> > > ### Author Response · Authors · 2026-04-08
> > >
> > > We greatly appreciate the reviewer’s confirmation that the concerns have been fully resolved. Should the reviewer feel appropriate, we would welcome an updated score reflecting the assessment. We will incorporate the updates in the revision.

---

### Official Review · Reviewer_MYf3 · 2026-03-13

**Soundness:** 3
**Presentation:** 2
**Significance:** 2
**Originality:** 2
**Overall Recommendation:** 4
**Confidence:** 3

**Summary:**

This paper presents an approach to reconstruct spatial sensor data (particularly MODIS) from sparse observations.  The approach, SENDAI, accounts for dynamics such as seasonality, phonemic shifts, etc.  The approach decomposes the reconstruction into two components: (1) low-frequency path that captures dominant spatiotemporal dynamics; and (2) high-frequency path for fine scale structure and boundaries.

**Compliance With Llm Reviewing Policy:**

Affirmed.

**Final Justification:**

Updated based on rebuttal clarifications.

**Key Questions For Authors:**

- Is there a relationship between number of sensors needs and the frequency of spatial phenomena in the scene
- Is there sensitivity to sensor placement in time/space?
- How sensitive is the proposed approach to changes in hyperparameter settings? Which settings have the most influence and how can these be more reliably set?

**Limitations:**

Limitations are clearly discussed and future directions outlined.  Including discussion of sensor placement (however, in many real/practice examples, influence over these locations may not be possible).

**Strengths And Weaknesses:**

Strengths:
+ The proposed approach addresses an important challenge of filling in missing remote sensing data (such as due to cloud cover, gaps in coverage, etc) which often limit reliability and usability of satellite surveys
+ Overall concept well presented and intuitive

Weaknesses:
- Involves a number of key hyperparameters are required that seem difficult to set - e.g., bandwidth for frequency clustering, top-k, sensor location, etc
- In Figure 4, it appears that sensors are more or less evenly spatially distributed, however in cases of outages such as cloud cover or sensor gaps, this sort of coverage is not reliably possible.  How does performance degrade with wider and wider gaps between sensor locations?
- Results are shown averaged over multiple runs yet standard deviation not shared?

---

> ### Author Rebuttal · Authors · 2026-03-29
>
> We thank the reviewer for the assessment and recognition that SENDAI addresses an important challenge. We respectfully note that SENDAI's reconstruction is applying *temporal* sensor histories - it is this temporal coherence that enables full-state reconstruction from hyper-sparse observations and reduces sensitivity to sensor placement and hyperparameter choices. We believe some concerns from the reviewer may stem from evaluating SENDAI as a spatial interpolation method rather than a framework with joint requirements of hyper-sparse spatial sensing, temporal sequence encoding, sim2real alignment, and multiscale decomposition.
>
> That said, we have added comprehensive new experiments that we believe address each concern raised and demonstrate the robustness, generalizability, and significance of the framework.
>
> - **W1: Number of key hyperparameters.**
>
> All hyperparameters are held at the same values across all six NDVI sites and the two additional remote sensing variables (LST, LSWI) we add - none were tuned per site or per variable. We added [*ablation studies (LINK)*](https://anonymous.4open.science/r/SENDAI-27CD/rebuttal/ablations.png) on the Tarim Basin site:
>
> *Sensor number*: Figure 5a of manuscript.
>
> *Temporal lag*: $L \in \{2, 5, 10, 20\}$, SSIM improves monotonically with increasing lag, consistent with Takens' embedding theorem: longer sensor histories provide a more faithful embedding of the underlying dynamics. We adopt $L{=}5$ as a conservative default.
>
> *Maximum target frequency*: $k_{\max} \in \{4, 8, 12, 16, 20, 24\}$, RMSE and SSIM remain relatively stable across the range. The overall insensitivity implies that SENDAI's spectral sparsity regularization and frequency exclusion zones are preventing the model from overfitting.
>
> *Sensor noise robustness*: $\sigma \in \{0, 0.02, 0.05, 0.10, 0.15\}$, training on clean data and evaluating on sensors corrupted with Gaussian noise. Metrics remain stable, confirming that the hierarchical corrections are not artifacts of overfitting to sensor noise.
>
> - **W2: Sensor placement under non-uniform coverage.**
>
> We first would like to note that SENDAI does not rely critically on sensor placement by design: the SHRED temporal unit reconstructs spatial fields from sensor histories, so temporal coherence compensates for spatial gaps - this is a fundamentally different regime from snapshot-based spatial interpolation. That said, we added [*a controlled experiment (LINK)*](https://anonymous.4open.science/r/SENDAI-27CD/rebuttal/sensor_placement_comparison.png) on the Tarim Basin comparing uniform coverage with density-biased clustering: 70\% of sensors concentrated in one quadrant (low-NDVI desert), 30\% scattered elsewhere - simulating higher evapotranspiration driving convective cloud formation. SENDAI achieves comparable metrics in both settings.
>
> Additionally, our new seismic waveform experiment provides a naturally occurring example: the global seismographic network concentrates ${\sim}$300 of 686 stations in the US, with sparse coverage elsewhere. SENDAI reconstructs waveforms at held-out stations - including sparsely instrumented regions - without modification. *Details in Review\#3 NQQv, W2*.
>
> - **W3: Standard deviations.**
>
> We have updated Tables 1 and 2 to include $\pm$ std for main text - [*SENDAI Jr. results (LINK)*](https://anonymous.4open.science/r/SENDAI-27CD/rebuttal/sendai_jr_results.png) and [*SENDAI results (LINK)*](https://anonymous.4open.science/r/SENDAI-27CD/rebuttal/sendai_results.png). Results from newly added experiments also have standard deviations.
>
> - **Q1: Number of sensors vs. frequency of spatial phenomena.**
>
> We interpret frequency of spatial phenomena as the complexity of the landscape's spatial structure - sharp boundaries, heterogeneous land cover, and multi-scale dynamics. More complex landscapes require more spatial samples to constrain the fine-scale structures. The sensor number ablation in Figure 5a of main text illustrates this empirically: SSIM improves from 8 to 32 sensors as the spatial structure becomes resolvable, then saturates beyond 64 as additional sensors provide diminishing information.
>
> - **Q2: Sensitivity to sensor placement.**
>
> See W2 and W1 lag ablation.
>
> - **Q3: Hyperparameter sensitivity and reliable setting.**
>
> See W1, none of the hyperparameters require site-specific tuning.
>
> To further strengthen our work, we also present: Deep learning baselines (*results and compatibility analysis in Review\#4 3vAL, A1*); experiments on LST (Land Surface Temperature) and LSWI (Land Surface Water Index) with identical hyperparameters, and seismic waveform reconstruction on a fundamentally different physical domain (*details in Review\#3 NQQv, W2*).
>
> In light of these additions - ablations, sensor placement robustness, three deep learning baselines, additional variable and cross-domain validation - we believe the concerns have been thoroughly addressed. We respectfully hope the reviewer will reconsider the assessment.

---

> > ### Author Rebuttal · Reviewer_MYf3 · 2026-04-03
> >
> > Reading the rebuttal comments to me and the other reviewers, the authors have addressed much of my concerns. I encourage them to consider revising some of the introductory text to make some of these clarifications more clear from the start of the paper.  I will update my score.

---

> > > ### Author Response · Authors · 2026-04-08
> > >
> > > We sincerely thank the reviewer for the careful re-evaluation and for adjusting the score. We will incorporate the updates in the revision.

---

### Decision · Program_Chairs · 2026-04-30

**Decision:**

Accept (regular)

**Comment:**

This paper introduces SENDAI to reconstruct full spatial states from hyper-sparse sensor observations by combining simulation-derived priors with learned discrepancy corrections. The experimental results on MODIS-derived vegetation index fields demonstrate that SENDAI achieves better results than the baseline methods considered in this paper. Although some reviewers raised concerns regarding the novelty of the proposed method, its generalizability to other physical variables, the sensitivity of hyperparameters, and the inclusion of recent baselines, the authors addressed the majority of the raised concerns during the rebuttal period. Exception remains for Reviewer NQQv's comments regarding novelty and as well as the need for a clearer physical justification of the correction mechanism.